**communications** engineering

# Multichannel multicentroid motion-compensated single pixel imaging of a 2D arbitrarily moving rigid-body target
Chongwu Shao [1], Yue Cao [2], Shijian Li [3] ✉, Xu-Ri Yao [1,4] ✉ & Qing Zhao [1] ✉

Single-pixel imaging (SPI), distinguished by its cost-efficiency, exceptional spectral adaptability, and robust sub-Nyquist-Shannon sampling reconstruction capabilities, demonstrates transformative potential across imaging applications yet faces critical limitations in capturing arbitrarily moving targets. This work introduces a simple yet effective SPI architecture capable of simultaneous real-time tracking and high-fidelity imaging of objects undergoing unconstrained 2D planar motion, encompassing both periodic and non-periodic translational/rotational kinematics. Our methodology employs six strategically designed Fourier patterns with optimized spatial frequencies as localization markers, combined with multichannel centroid tracking, to achieve precise motion dynamics characterization. Furthermore, we develop a straightforward inverse motion-compensated reconstruction method that efficiently reconstructs images of objects subjected to composite motion. The proposed framework notably maintains reconstruction integrity even when individual detection channels experience temporary out of the field of view.

Single-pixel imaging (SPI) is an emerging computational imaging technology in which the object is modulated using a series of encoded structured patterns[1–3], and the resulting modulated light is subsequently integrated by a single-pixel detector[4,5]. SPI offers an alternative to imaging systems that rely on detectors that are either too expensive or too specialized to be feasibly produced in an array format[6,7], and has been successfully applied for terahertz imaging[8,9], single-photon imaging[10,11], and X-ray imaging[12,13]. Additionally, with SPI, multidimensional information can be acquired using a single-pixel detector[14–16] and allowing 3D imaging[3,17]. Furthermore, SPI enables the stable reconstruction of images sampled at rates below those required by the Nyquist-Shannon sampling theorem via compressive sensing (CS)[18,19]. Owing to this property, SPI has been widely validated in various optical applications, including lidar imaging[20,21], multispectral imaging[22,23], and phase imaging[24,25]. Recently, there has been considerable interest in the structure of patterns and the image reconstruction process in SPI. Hadamard[26,27] and Fourier[28,29] basis patterns have become increasingly popular because of their energy concentration and robustness[30–32].

However, the SPI process necessitates that the target remain stationary, as spatial information of the object is reconstructed using sequentially modulated patterns and their corresponding intensity signals. Relative motion may disrupt these correlations, leading to motion blur. A digital micromirror device (DMD) is the most commonly used spatial light modulator in SPI[33,34], but its maximum modulation rate of 22kHz typically results in imaging frame rates of less than 100fps. To address this problem, high-speed pattern projection methods have been proposed, for example, fast-switching light-emitting diodes (LEDs)[35–37], cyclic patterns on a spinning pattern[38,39], and DMD integrated with laser scanning[40]. Although these custom hardware systems have pattern projection rates of more than 1Mfps, the trade-off between spatial resolution and measurement time in SPI remains unresolved; the imaging frame rate is significantly lower than the modulation frequency of the device, approximately 1/1000 of it. At present, several approaches have been proposed to mitigate the above trade-off, including incorporating the spectral dimension to enhance sparsity and thereby increase the compression ratio[14]; employing customized patterns to enable efficient low-resolution previews together with convex optimization for high-resolution video reconstruction[41]; and dynamically adjusting the balance between temporal and spatial resolution according to frame complexity[42]. Additionally, real-time velocity and trajectory estimation is crucial in motion detection applications. In dynamic scenarios, this critical information must be available instantly. Unfortunately, existing high-speed

[1]Center for Quantum Technology Research and Key Laboratory of Advanced Optoelectronic Quantum Architecture and Measurements (MOE), Beijing Institute of Technology, Beijing, China. [2]Chinese Academy of Agricultural Mechanization Sciences Group Co. Ltd., Beijing, China. [3]School of Integrated Circuits and Electronics, Beijing Institute of Technology, Beijing, China. [4]State Key Laboratory of Chips and Systems for Advanced Light Field Display, Center for Interdisciplinary Science of Optical Quantum and NEMS Integration, Beijing Institute of Technology, Beijing, China. ✉e-mail: shijianli@bit.edu.cn; yaoxuri@bit.edu.cn; qzhaoyuping@bit.edu.cn

projection methods require post-processing for motion parameter extraction, resulting in unavoidable latency between measurements and actionable insight.

Recently, a new class of SPI configurations has been proposed that simultaneously acquires the target's trajectory information during the conventional SPI sampling process and then performs motion-compensated reconstruction, thus decoupling the modulation patterns and the intensity signals for rigid (non-deforming) moving objects to some extent. Consequently, this method allows for SPI of moving objects with arbitrary spatial resolution and sampling rate, without sacrificing the temporal resolution required for motion sensing of the target[43–45]. Initially, this method was implemented on imaging targets that undergo translational motion, which involves repeatedly embedding a limited number of location patterns throughout the imaging process to identify the target's location. During the reconstruction stage, the modulation patterns are adjusted in reverse to compensate for the motion, restoring the original shape of the target and the trajectory of the motion simultaneously[44–51].

Within a two-dimensional plane, motion can be decomposed into two primary forms: translation and rotation, while implementing compensation for rotational motion is significantly more complex. The Laguerre-Gaussian transform has been introduced to SPI to image rotating targets with prior knowledge of the target rotation center[52,53]. An SPI method primarily designed to handle rotating targets has been proposed that leverages the correlation information between the dynamic scene and each static pattern, but requires the dynamic scene to be repetitive or reproducible[54–56].

When targets exhibit both translational and rotational motion, a direct and effective strategy is to use well-established motion sensors or a CMOS camera to detect the motion in real time and calibrate it with the SPI system[57,58]. In a classical SPI architecture, an analogous approach is to reconstruct a low-resolution preview image to estimate the motion, which inevitably entails a trade-off between localization accuracy and temporal resolution, limiting adaptability and accuracy for high-speed targets[41,59,60]. When directly detecting a target's translation and rotation without an image under a classical SPI architecture, additional specialized patterns are needed to determine the target's rotation angle, leading to a decrease in imaging frame rate. Xiao et al. and Ji et al. employed geometric moment patterns to statistically estimate the principal axis of the target, in which three or more supplementary localization patterns need to be integrated into the positioning process, resulting in at least a twofold loss in temporal resolution[43,61].

In this paper, we propose a technique termed MC3-SPI (Multichannel Multicentroid Motion-compensated SPI) for tracking and imaging targets undergoing arbitrary two-dimensional motion (including both translation and rotation, periodic or non-periodic motion). We develop an optimized

Fourier localization method that uses spatial-frequency-optimized localization patterns and Sierra-Lite dithering with serpentine error diffusion to precisely determine centroids of targets with diverse sizes and morphologies. MC3-SPI estimates the target's kinematic state by leveraging the centroid positions from multiple channels, and enables real-time imaging of the target using the proposed inverse motion-compensated transformation (IMCT). In our implementation, multiple centroids are obtained in the RGB spectral channels. Experiments show that, unlike translational-only compensation schemes, MC3-SPI reconstructs color images of diverse moving objects and their motion parameters without requiring additional localization patterns. Furthermore, MC3-SPI can image an object that extends beyond the boundary of the field of view (FOV), even when the target does not fully enter the FOV during the entire measurement process. The MC3-SPI framework is compatible with general SPI systems and, when paired with high-speed modulation, supports higher temporal-resolution imaging.

## Results

### Absolute coordinate localization of the centroid

Each Fourier coefficient carries global information, so local changes appear across the spectrum. For a single rigid target, motion can be estimated by tracking only a few coefficients. Fourier-SPI recovers these coefficients with only a few measurements.

By the Spatial-Shift property of the Fourier transform, a translation $(\Delta x, \Delta y)$ induces phase shifts of $-2\pi f_x \Delta x$ and $-2\pi f_y \Delta y$ in the coefficients at $(f_x, 0)$ and $(0, f_y)$, respectively. This property yields only relative displacement. We extend this to absolute positioning by linearly mapping the measured phase directly onto the spatial coordinates.

We numerically evaluate the absolute localization error as a function of spatial frequency, using Fourier basis patterns $(f_x, f_y) = (f, 0)$ and $(0, f)$ for the $x$ and $y$ components, respectively.

As shown in Fig. 1a, we evaluate the absolute localization error as a function of target size and spatial frequency for Fourier basis patterns. Localization at frequencies $f > 1/M$ fails due to periodic ambiguity. Within $f \leq 1/M$, lower frequencies yield higher accuracy, while larger targets introduce an intrinsic phase bias that increases error. Thus, lower-frequency Fourier patterns provide the more accurate absolute localization.

Considering practical DMD-based SPI, we further simulate the binary implementation with Sierra-Lite dithering and a serpentine error-diffusion path in Fig. 1b. For typical target sizes of 80–120 pixels (approximately 1/9–1/4 of the field of view), the optimal Fourier localization pattern has a spatial frequency of $1/(2M)$ and yields a localization error of ~1/3 pixel. Consequently, the optimized high precision localization method can correctly detect subtle variations across a broader range of target sizes, for

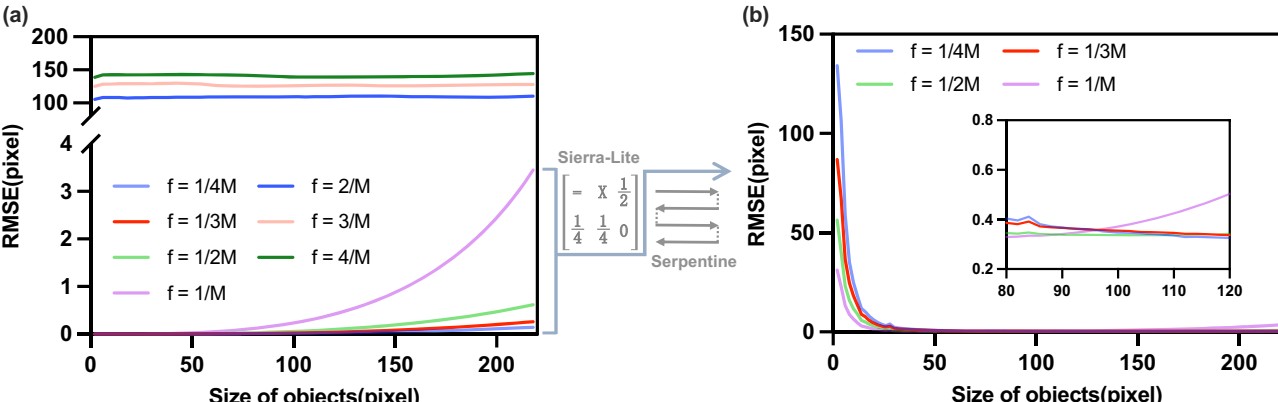

**Fig. 1 | Simulation of localization performance for Fourier location patterns at different spatial frequencies f.** The pattern size is 256 × 256 with no up-sampling, i.e., $M = 256$. A total of 500 MNIST digital targets are used, each randomly positioned 100 times for each target size. The mean RMSE (root-mean-square error) of the estimated centroid is computed and plotted versus target size. **a** Theoretical localization performance of Fourier basis patterns at different spatial frequencies f. **b** Empirical localization performance of Fourier location patterns at different spatial frequencies f under three-step phase shifting combined with Sierra-Lite dithering and a serpentine error-diffusion path.

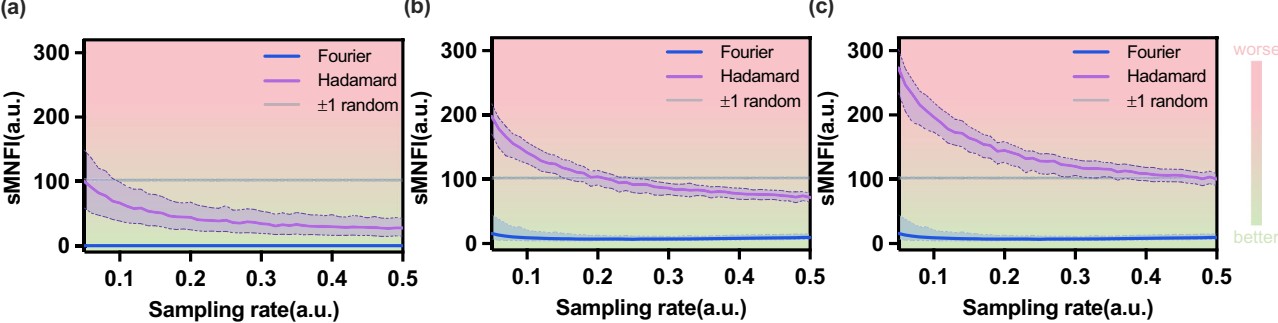

**Fig. 2 | Orthogonality analysis of Fourier and Hadamard basis patterns after motion-compensation at sampling rates between 5% and 50%.** The orthogonality of the basis patterns is quantified by the sMNFI (scaled mean normalized Frobenius inner-product magnitude) between each compensated basis pattern and the other uncompensated basis patterns. Statistical analysis employs the median together with the $p25 \sim p75$ interval (IQR, interquartile range). Values concentrated nearer to zero indicate stronger orthogonality. The random $\pm 1$ patterns (without motion-compensation) serve as a reference. Orthogonality analysis under **a** random translation compensation, **b** random rotation compensation, and **c** random translation and rotation compensation. The specific data distributions are shown in Supplementary Figs. 10 and 11.

example, accurately distinguishing the centroid positions of different constituent components in complex targets.

Supplementary Note 1 provides a more detailed discussion of the error analysis of the Fourier localization method and the rationale for selecting the spatial frequency $1/(2M)$.

## Orthogonality analysis of motion-compensated patterns

In SPI, motion compensation is commonly implemented by applying translations or rotations to the patterns in opposition to the target motion. Such transformations often alter the intrinsic properties of the patterns, thereby affecting sampling and reconstruction. Given that Fourier basis patterns and Hadamard basis patterns enable efficient sampling and reconstruction in SPI owing to their orthogonality, we conduct numerical simulations on the orthogonality of these bases under motion compensation.

In compressed sensing, the maximum normalized Frobenius inner product of a set of patterns is used to evaluate their cross-correlation and thereby ensure successful image reconstruction[62,63]. In this work, we aim only to quantify the loss of orthogonality of motion-compensated patterns and therefore adopt the mean normalized Frobenius inner product. A lower value also indicates weaker correlation and thus a smaller loss of orthogonality induced by motion compensation.

For each sampling rate, we randomly select 800 basis patterns to form a subset. Each chosen pattern is randomly rotated, translated, or both, and its sMNFI (scaled mean normalized Frobenius inner-product magnitude) with the other motion-uncompensated patterns in the subset is computed (Eq. 12 in Supplementary Note 7). This process is repeated 100 times per pattern, and the resulting values are statistically analyzed. Because the data are non-Gaussian, we report the median and the $p25 \sim p75$ interquartile range (IQR) as a robust uncertainty band, which represents the central 50% of the observations. We also include random $\pm 1$ patterns without motion compensation as a control, with results shown in Fig. 2.

Across sampling rates of 5~50%, Fourier patterns maintain significantly stronger orthogonality than Hadamard patterns after motion compensation. As shown in Fig. 2a, Fourier basis patterns are scarcely affected by translational compensation. As shown in Fig. 2b, Fourier basis patterns retain substantial orthogonality after rotation compensation. As shown in Fig. 2c, after joint translation-rotation compensation, the loss of orthogonality of Fourier basis patterns throughout the full sampling range remains at a very low level, which makes them well suited for motion-compensation imaging.

In contrast, the orthogonality of Hadamard basis patterns degrades substantially after motion compensation. In particular, after rotational compensation, their orthogonality at low sampling rates is even inferior to that of random ±1 patterns. In addition, the orthogonality of the Hadamard

basis patterns improves as the sampling rate increases, indicating that reconstruction requires a relatively high sampling rate.

We provided a more detailed discussion in Supplementary Note 7. Most critically, we demonstrate that the complete set of Fourier bases constitutes a linear space closed under translations and rotations. This property makes the Fourier basis the optimal SPI sampling mode for motion-compensated imaging of two-dimensional composite motion targets.

## Experiment setup

Our experimental setup is shown in Fig. 3a. We use a passive DMD-based SPI system to image targets undergoing composite translational and rotational motion. At the receiving end of the signal, two dichroic mirrors are used to split the signal at wavelengths of 490 nm and 605 nm into three channels. These signals are then collected by three bucket detectors, thereby enabling multispectral information of the target to be collected from multiple channels. The spectral bands correspond to the RGB channels to achieve colored imaging of the target. In addition, the internal structure of each measurement frame consists of 6 localization patterns and imaging patterns that yield a single imaging coefficient, which defines the minimum temporal resolution for motion sensing in MC3-SPI.

Detailed experimental instrumentation and pattern setup is provided in Supplementary Notes 3 and 4.

## Motion capture and image reconstruction

The complete motion state of a rigid-body target moving in two-dimensional space can be determined by considering multiple points attached to the target. For common rigid targets, the intensity distributions of signals in different spectral bands vary, allowing the localization of multiple centroids on a single target. Based on this universal prior, in our experiment, we use Fourier localization patterns with a frequency of $1/2M$ (M denotes the size of the pattern) to achieve the most precise localization, allowing us to distinguish subtle differences among the RGB channels to determine the absolute coordinates of three centroids. The motion of a rigid body can be decomposed into translation and rotation based on an arbitrary point. As shown in Fig. 3b, the geometric center of the three centroids from the target's RGB channels as the reference point for the target, the directional angle of the line connecting any two centroids (R and B channels as an example) is defined as the rotation angle. Thus, the complete two-dimensional motion information of the target can be obtained.

After obtaining the two-dimensional rigid body motion information, we sequentially apply inverse translation and rotation on the measurement patterns to compensate for the target's translation and rotation. As shown in Fig. 3c, mimicking the inverse transformation method, the motion-compensated patterns are weighted by the corresponding spectral

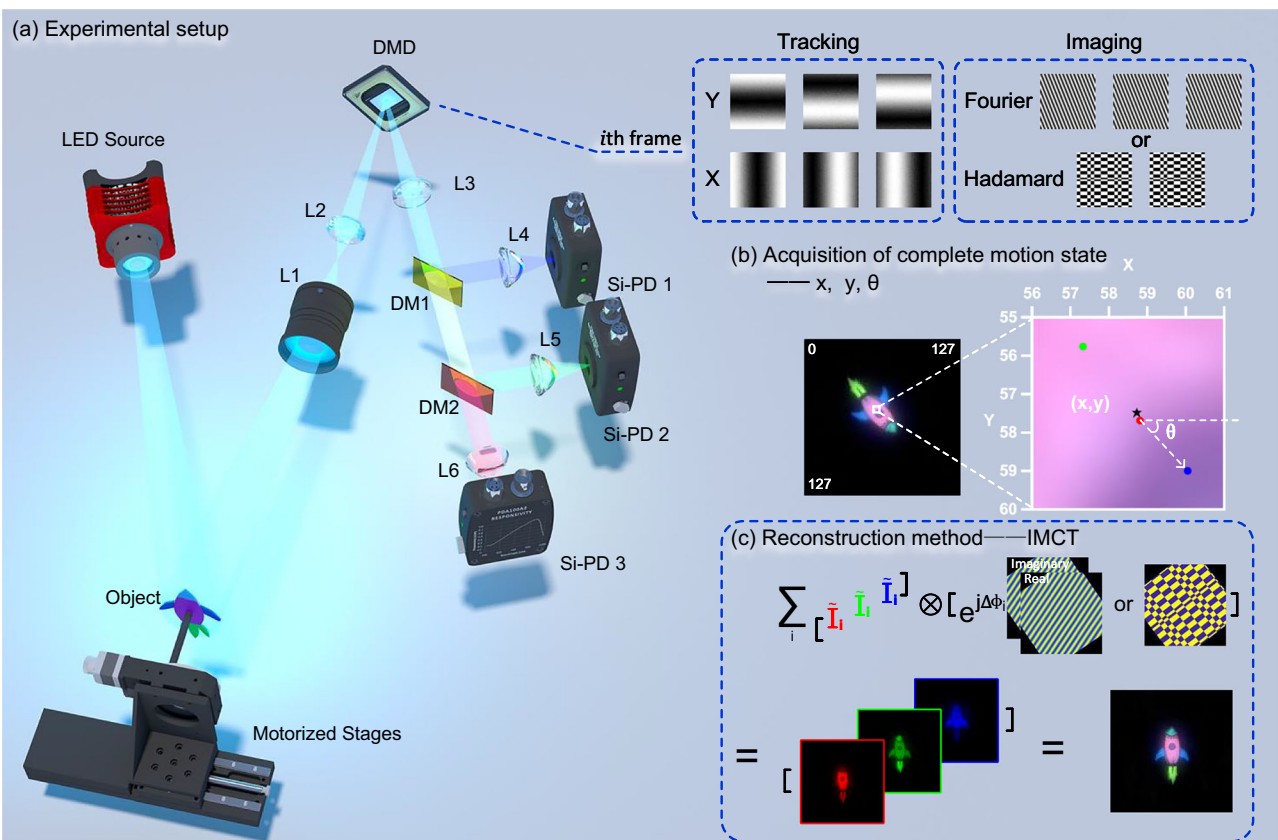

**Fig. 3 | Experimental setup, acquisition of complete motion state and principle of the IMCT (inverse motion-compensated transform) reconstruction method.**
**a** White light from an LED illuminates the target, and a passive SPI (single-pixel imaging) system based on a DMD (digital micromirror device) performs structured sensing. The signal is then split by two dichroic mirrors into three spectral components corresponding to the RGB channels, and each signal is measured by a single-pixel detector. Six Fourier localization patterns together with either Fourier or Hadamard imaging patterns constitute a measurement frame, and multiple frames are sequenced to alternately perform localization and imaging, ultimately yielding the complete pattern set. **b** The left image displays the real SPI image of the target at the current moment within a 128 × 128 pixel field of view, while the right image illustrates the method for obtaining the target's complete motion information. In the right image, the red, green, and blue points represent the multicentroid localization results from the RGB channels. The pentagram marks their geometric center, acting as the reference point for the target, and the angle between the blue and red points is used to determine the target's direction angle. **c** The motion-compensated basis patterns are weighted by their corresponding measured coefficients $I_i$ and summed to reconstruct the target image, and the RGB image is synthesized from the results of the three wavelength channels.

coefficients and summed to directly reconstruct the target image. However, the patterns are not standard basis patterns; we refer to this simple and effective approach as the inverse motion-compensated transform (IMCT) operation. Accordingly, the method based on Fourier-SPI is called the inverse motion-compensated Fourier transform (IMCFT), while the method based on Hadamard-SPI is referred to as the inverse motion-compensated Hadamard transform (IMCHT).

**Tracking and imaging of a motion multicentroid target**
In our first experiments, we used the colored letters "BIT" as a multi-centroid target, assigning them to the red, blue, and green wavelength channels for tracking and imaging. In the first trajectory, the target performs torsional oscillation about a reciprocating axis, with the maximum average angular speed and the maximum average linear speed of 45° s$^{-1}$ and 20mm s$^{-1}$, and rotation and translation amplitudes of 60° and 15mm, respectively. To increase the target's velocity relative to the measurement process, we reduced the DMD flipping rate to 150 Hz, corresponding to relative motions of 2.70° and 1.20 mm per measurement frame, and deliberately introduced motion blur in conventional SPI for validating the MC3-SPI.

Figure 4a shows the tracking of the three centroids together with the recovered rigid-body motion, including both translation and rotation. Figure 4b presents the motion-compensated reconstructions using Fourier

patterns with optimized sampling as described in the Supplementary Note 2 and Hadamard patterns. A 128 × 128 full-color image is obtained after sequential translation and rotation compensation. By comparing reconstructions without compensation, with translation-only compensation, and with full compensation, the improvement offered by MC3-SPI is demonstrated. TVAL3 is further applied to enhance the image quality.

The second trajectory, shown in Fig. 4c, is a random off-centroid rotation with randomly varying amplitude and angular velocity. The DMD flipping rate is 4500 Hz. As shown in Fig. 4d, MC3-SPI accurately tracks the large angular variations and yields a full-color reconstruction of "BIT" under this more challenging random motion.

Real-world objects are often more complex, so we further evaluated MC3-SPI on a Mario figurine and a custom cartoon rocket undergoing compound motion. For the Mario toy, the motion trajectory consisted of torsional oscillation and reciprocation with an average angular velocity of 30° s$^{-1}$ and an average linear velocity of 20mm s$^{-1}$, with amplitudes of 60° (rotation) and 15mm (translation). The DMD flip rate was set to 250 Hz, corresponding to relative speeds of 1.08° and 0.72 mm per measurement frame. For the rocket, the motion follows a random reciprocating oscillation, and the DMD flip rate was set to 2000 Hz. The results of the tracking and reconstruction are shown in Fig. 5.

Compared with the letters "BIT", the Mario and rocket images are more structurally complex and yield multicentroid localization results that

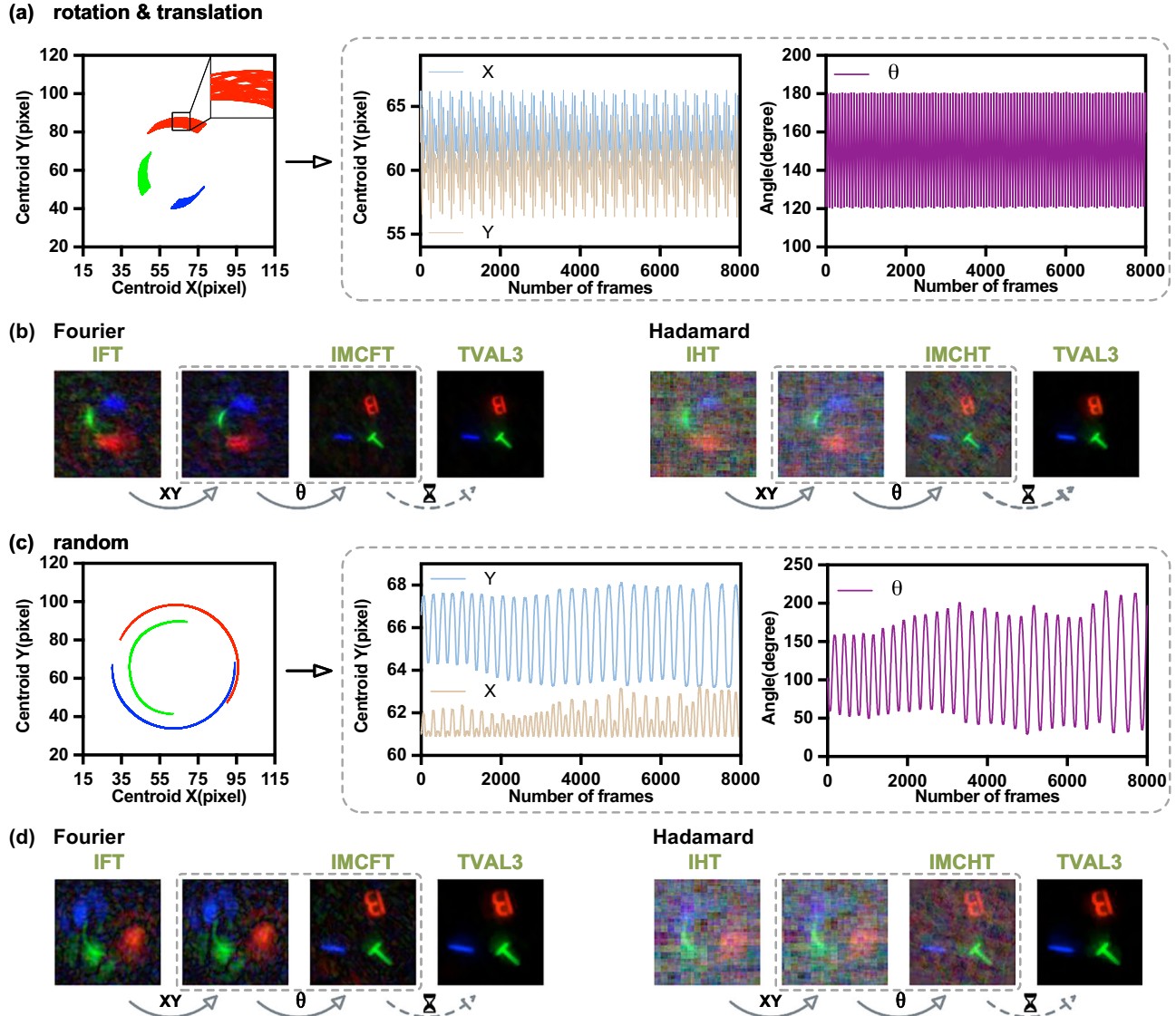

**Fig. 4 | Tracking and motion-compensated imaging of the color letters "BIT".** Tracking of the three centroids of the color letters "BIT" undergoing **a** composite motion and **c** random eccentric rotation, together with the recovered rigid-body motion parameters (absolute position and rotation angle). Motion-compensated imaging results for "BIT" under **b** composite motion and **d** random eccentric rotation using Fourier and Hadamard patterns, reconstructed with the IMCT (inverse motion-compensated transform) and TVAL3 (Total Variation Minimization by Augmented Lagrangian and Alternating Direction Algorithms) methods. All images are 128 × 128 pixels and reconstructed at 50% sampling.

are more densely clustered with smaller spacing, which places stricter demands on localization accuracy and imaging efficiency. However, our optimized localization method can still reliably distinguish the centroids across different channels for complex targets, enabling accurate estimation of the target's motion state and, in turn, guiding motion-compensated reconstruction with appropriately chosen sampling patterns to generate a clear, full-color image.

Across all four experiments, the IMCFT method significantly outperformed IMCHT for targets with two-dimensional composite motion. As seen in Fig. 4b, d and Fig. 5b, d, IMCFT provides more efficient information acquisition, cleaner backgrounds, and superior preservation of fine details. In particular, TVAL3-enhanced IMCFT reconstructions exhibit sharper edges and better retention of intricate features, supporting our previous insight that Fourier basis patterns are optimal for SPI involving both translation and rotation.

Because the motion-compensated basis patterns are no longer strictly orthogonal, IMCT reconstructions exhibit low-frequency blotchy background artifacts. TVAL3 effectively suppresses these artifacts while enhancing edge structures. In our experiments, IMCFT required only ~ 0.042 s to reconstruct a 128 × 128 full-color image, whereas TVAL3 required ~50 s. Thus, TVAL3 is preferable when reconstruction quality is the primary consideration, while IMCFT provides a better balance between speed and image quality for real-time applications. The motion processes across the four experiments are also provided as a video in Supplementary Movie 1.

**Extend-FOV tracking and imaging of a boundary-motion target**
A rigid target in 2D space has only three degrees of freedom: two translational degrees of freedom and one rotational degree of freedom, which can be completely determined by the two centroids bound to the target. However, MC3-SPI can locate multiple centroid positions, providing redundancy in localization. This redundancy enables our system to operate in extreme motion scenarios. For a target moving across the field-of-view boundary, the SPI-acquired coefficients remain valid within an appropriately defined discrete domain[64], and we design an extended field-of-view scenario accordingly.

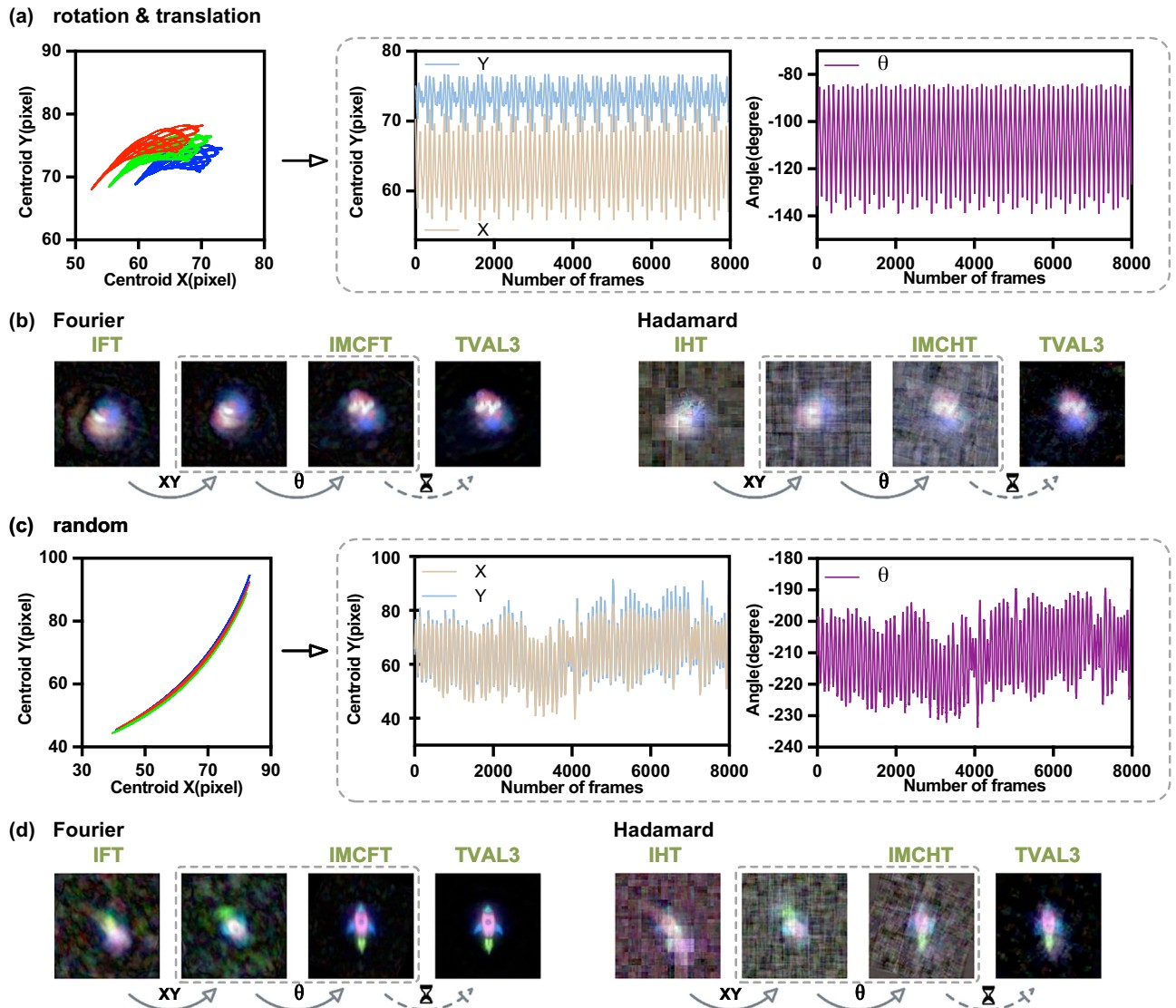

**Fig. 5 | Tracking and motion-compensated imaging of real objects. a, c** Tracking of the Mario toy (composite motion) and the rocket toy (random reciprocating motion), including three centroids and recovered rigid-body motion parameters. **b, d** Corresponding motion-compensated reconstructions using Fourier and Hadamard patterns with IMCT (inverse motion-compensated transform) and TVAL3 (Total Variation Minimization by Augmented Lagrangian and Alternating Direction Algorithms). All images are 128 × 128 pixels and reconstructed at 50% sampling.

To test the system under extreme motion, we imaged a boundary-motion target where the color "BIT" rotated randomly along the edge of the field of view (FOV), so that the three letters never fully entered the FOV simultaneously. The first trajectory plot in Fig. 6a shows the results obtained by directly applying the localization method, and distorted and offset trajectories can be observed in the upper part of the plot. because once the target moves beyond the FOV boundary, only the centroid of the visible part is recorded, and after the target leaves completely, only noise is captured. Since these distorted segments cannot reliably guide the motion compensation, they must be removed.

Figure 6a summarizes the trajectory processing. We first apply intensity and spatial filters to remove invalid segments, and then use the relative geometry of the three centroids to recover paths outside the FOV. For intensity filtering, we sum the three measurements from the three-step phase-shifting method and reject samples below half of the maximum signal. Spatial filtering then discards trajectories in the top one-fifth of the FOV, where distortion occurs. From the filtered trajectories, we generate a "condition met" map to gate valid imaging measurements and guide the extraction of the pairwise relative distances between centroids. Using these relative distances and the two valid coordinates (the two centroids inside the

FOV), we calculate the true coordinates of the centroid that lies outside the FOV.

By following this approach, we obtained the effective part of the target motion state and performed motion-compensated imaging to directly reconstruct an image of the target, as shown in Fig. 6b. It can be clearly observed that we recovered the target's trajectory outside the FOV (the portion where the y-axis coordinate exceeds 128). Furthermore, leveraging the inherently cumulative characteristics of SPI reconstruction, we directly reconstructed the image of the target with IMCFT, even though the target never fully entered the FOV. Moreover, owing to the aforementioned filtering process, the overall translational and rotational results derived from analyzing the trajectories of the three-channel centroids are discontinuous.

In terms of the imaging strategy, we optimized the Fourier sampling scheme to accommodate the filtering of the target trajectory without any prior information by repeatedly sampling 10% of the low-frequency components, ensuring that even after filtering, sufficient low-frequency sampling is maintained to guarantee imaging stability. Meanwhile, to achieve denser temporal coverage for the sampling 10% case, we slightly increased the DMD flipping rate to 3096 Hz, enabling completion of the sampling

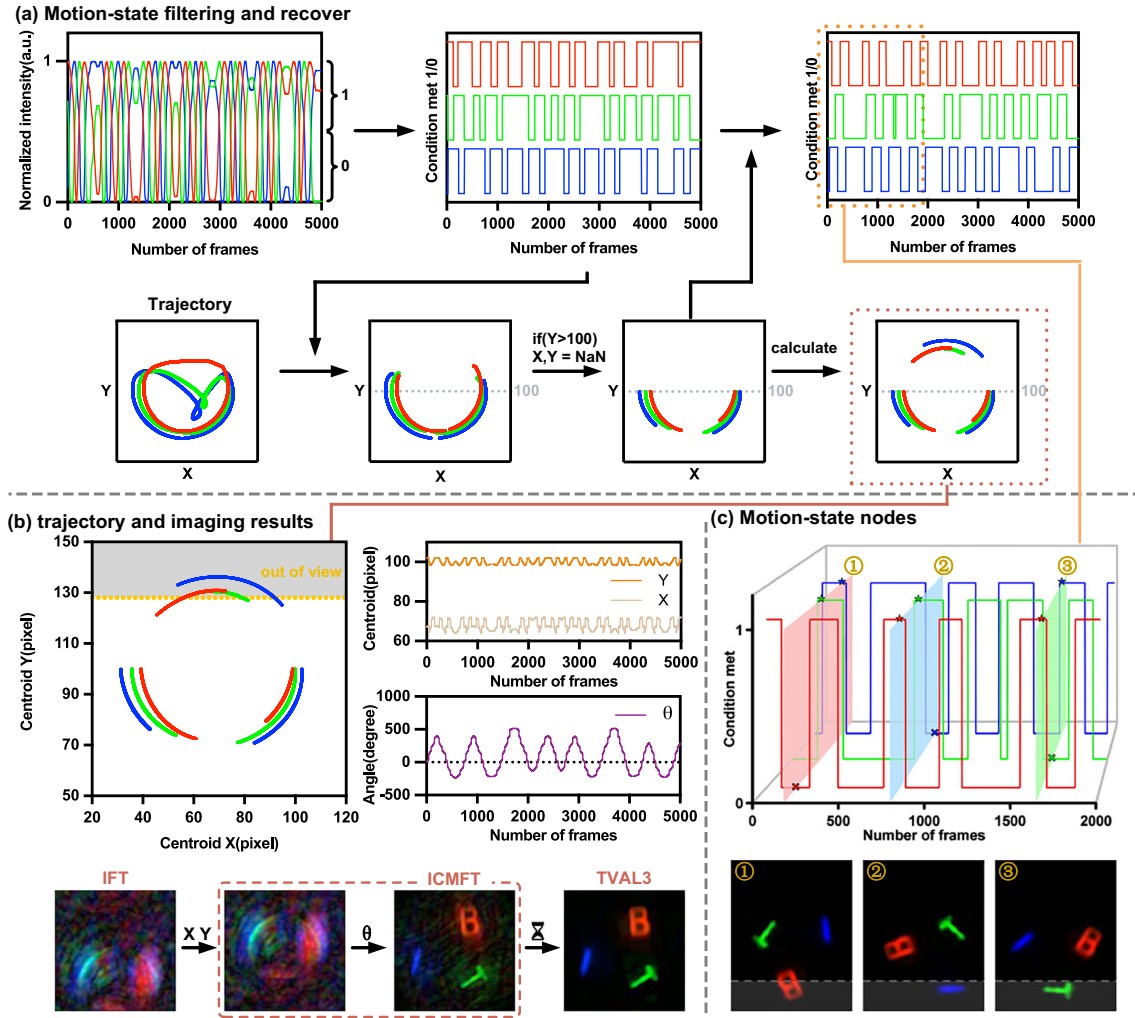

**Fig. 6 | Principles and results of a boundary-motion target imaging. a** Trajectory processing: By using the signal strength and spatial filtering, we remove the invalid portions of the trajectory caused by the target moving out of the field of view. Further, by using the relative positions of the three centroids, we accurately determine the positions of the centroids outside the field of view, allowing us to obtain the target's real motion state information. **b** Effective motion trajectory and imaging results: This part shows the reconstructed valid motion trajectory and the corresponding imaging results. Fourier patterns are used for imaging, and the motion compensation and reconstruction methods are the same as those described in the previous section. Reconstruction of a 128 × 128 pixel image with repeatedly sampling 10% of the low-frequency components. **c** Target motion state with condition met: Three important nodes of the target's motion state are selected and correlated with the target's status at those points.

10% within 4.76 s. This strategy is described in detail in the second part of Supplementary Note 2.

To further illustrate the details of our method, we show the correspondence between the condition met states and the target's motion state in Fig. 6c. We selected three representative moments and reconstructed the target's motion state at those times. The condition met states align perfectly with the motion of the target. At these moments, blue "I" and green "T" have completely exited the FOV, whereas red "B" is partially still within the FOV. This behavior matches the characteristics of the trajectory shown in the first trajectory in Fig. 6a.

Similarly, on the basis of Fig. 6b, we can reconstruct part of the target's motion state and output it as a video; the results are provided in Supplementary Movie 2.

## Discussion

We propose an effective SPI method based on RGB channel detection for attitude sensing and color imaging of rigid-body targets undergoing 2D motion. Compared with existing motion estimation schemes, MC3-SPI does not require any prior information about the target's motion or additional angle detection patterns, enabling the detection of arbitrary random

motion without sacrificing the temporal resolution. In fact, the complete motion state of the target (absolute position and rotation angle) can be obtained using only six Fourier localization patterns. Consequently, when the DMD operates at the maximum modulation frequency of 22kHz, the complete localization of the target can be performed at 3666Hz, and even when image patterns are considered, the target's motion state can be captured at 2444Hz. In particular, MC3-SPI is compatible with any SPI system, which means that it can be seamlessly integrated with existing techniques to increase SPI sampling speed and thus achieve higher temporal resolution.

For rotational compensation, even very small angular errors can cause displacements of several pixels in regions far from the rotation center, thereby degrading the quality of the reconstructed image. Therefore, a high localization accuracy is crucial for precise angle calculation. As illustrated in Supplementary Fig.7c, noise first causes rotational ghosting in the target image, confirming the importance of precise localization. Two centroid localization methods are used in the SPI system: the geometric moment (GM) localization method and the Fourier localization method (the method used in this paper). Although the GM method can obtain the absolute coordinates of the target, it lacks complementary measurements and exhibits a relatively poor robustness to noise (GM section of Supplementary

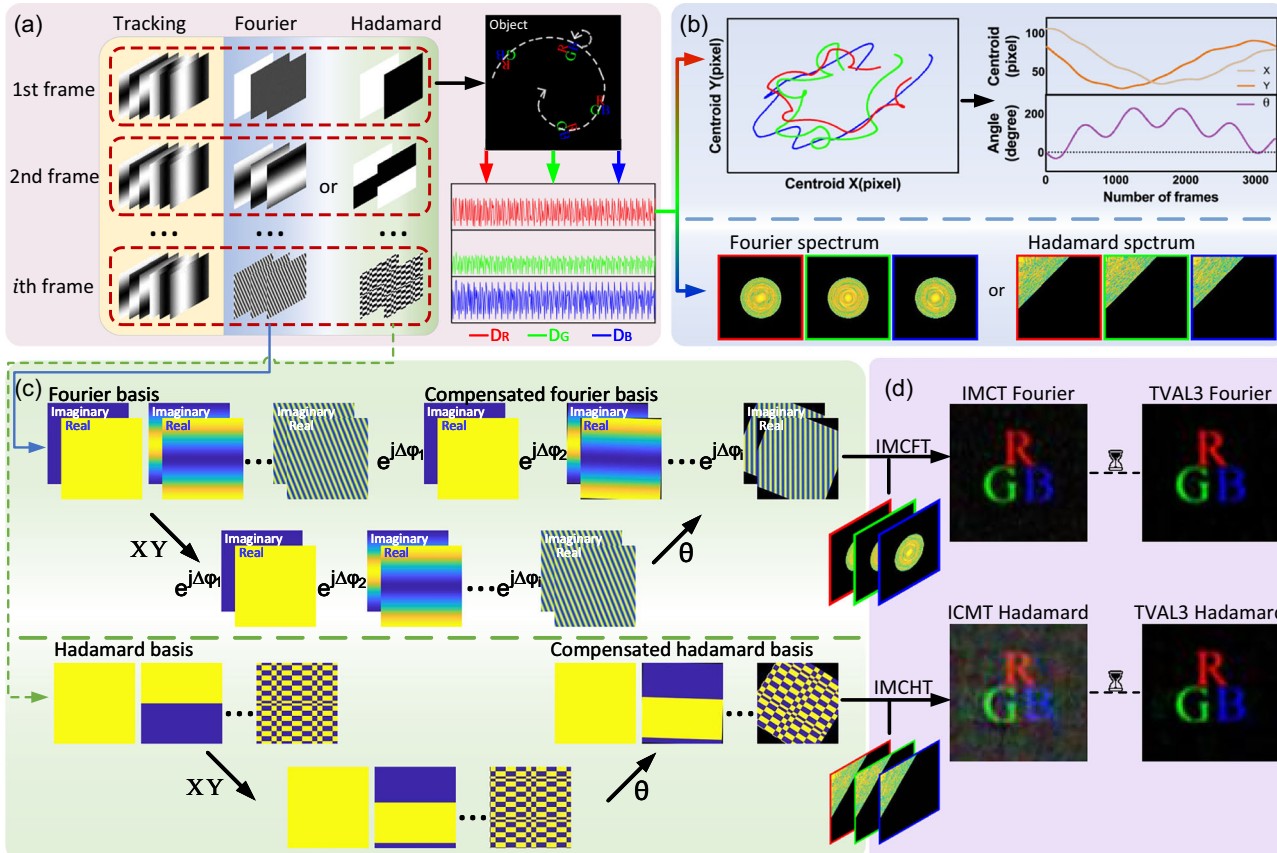

**Fig. 7 | Schematic for tracking and imaging a target undergoing 2D composite motion. a** Measurement process: Each measurement frame consists of six Fourier localization patterns and three Fourier or two Hadamard imaging patterns to capture both the target's motion information and image information, with data acquired via the RGB channels. **b** Data process: The localization and imaging data are separated and processed independently. The trajectories of the three color centroids are extracted from the localization data to determine the target's motion state (absolute coordinates and rotation angle), while the corresponding spectral information is obtained from the imaging data. **c** Compensation process: On the basis of the motion information, the basis patterns are inversely shifted to compensate for the motion. For the Fourier method, translation is compensated by multiplying by a motion-related coefficient. **d** Reconstruction process: The target image is reconstructed using the IMCT (inverse motion-compensated transform) and TVAL3 (Total Variation Minimization by Augmented Lagrangian and Alternating Direction Algorithms) methods (with TVAL3 requiring a longer processing time). Only the target region is displayed, and this region is magnified to highlight the reconstruction details.

Note 6). In contrast, the Fourier localization method is much more robust against noise, although strictly speaking, it can only provide relative coordinates mathematically[65]. However, with an appropriate configuration, the Fourier localization method can approximate the absolute position of the target[45]. Building upon this work, we conducted further research that improved the accuracy of the Fourier localization method. In this work, we employ Fourier basis functions with spatial frequencies of $1/2M$ and $1/2N$, and apply the Serria-Lite method with a serpentine diffusion path, achieving an optimal localization accuracy of approximately $1/3$ pixels when the target size accounts for approximately $1/9 \sim 1/4$ of the FOV. It is worth noting that sub-pixel errors can still induce reconstruction model mismatch. We provide a quantitative analysis in Supplementary Fig.18, which confirms that the impact of sub-pixel errors on Fourier-SPI reconstruction is limited and can generally be neglected. Moreover, as shown in Supplementary Fig.6a, Supplementary Fig.6b, and Supplementary Fig.7a, our localization method performs robustly in an environment with 40dB intrinsic noise (dark noise).

In SPI systems, single-pixel detectors capture the total intensity of modulated target signals, which improves signal utilization and allows for signal detection even at a low signal-to-noise ratio (SNR). To verify the robustness of the proposed approach, we conducted numerical simulations by directly adding different levels of noise to the acquired signal to simulate detector noise. As detailed in Supplementary Note 6, even in the case of severe noise, with an intrinsic noise level of 86.30dB and an ambient noise level of 0dB, our method still reconstructs images with minimal degradation.

In addition, target tracking requires real-time optical alignment to keep the target within the FOV, but high-speed motion often leads to loss of the target. By localizing multiple centroids of the target along the wavelength dimension, we obtain redundant motion-related information enabling boundary-motion target imaging, so that the system can acquire motion and imaging data effectively under more extreme conditions. This approach improves the fault tolerance of the optical system, reduces the load on the mechanical system, and enhances the overall stability and robustness of the system in complex dynamic scenarios.

The IMCFT method is simple and effective: It requires only weighting and summing the motion-compensated basis patterns with their corresponding spectral coefficients, thereby delivering outstanding reconstruction speed. The experimental results show that the IMCFT method reconstructs images in only ~1/1000 of the time required by the TVAL3 algorithm, while still providing accurate reconstruction results. As detailed in Supplementary Note 8, the IFMCT method reconstructs a recognizable image with a sampling rate of only 5%. For $128 \times 128$ pixel SPI, there are 820 sampling frames. With the experimental device and the workstation (Supplementary Note 3), the SPI measurement process takes 0.34s, the compensation time is 0.19s, and the reconstruction time is 0.0044s. Since the sampling and reconstruction processes in SPI are independent of the individual frames, these processes can be highly parallelized; thus, the measurement process determines the target recognition delay. If DMD is replaced with a higher-speed SPI scheme, the latency bottleneck will be determined by the compensation and reconstruction times, which are

approximately 0.20s. By vectorizing all data and performing the full computation on the CPU in a single pass, the time could be reduced to approximately 0.05s, albeit at the cost of substantial memory requirements. Moreover, owing to the inherent cumulative character of the IMCFT method, each update could build on the previous result. Each incremental update incurs a computational latency of only ~ 0.0015s, which is sufficient to support real-time video streaming at ~666fps once the approximate shape of the target is obtained.

Although MC3-SPI can effectively sense arbitrary two-dimensional motion of rigid targets, its applicability remains limited by intrinsic methodology and system-level constraints. First, our motion-sensing approach imposes constraints because the target orientation is inferred from the relative positions of centroids across different channels. This essentially requires cross-channel intensity distributions to break central symmetry so that the centroids of the channels are measurably separated. In addition, the inherently global sensing nature of the SPI system limits precise target acquisition and imaging in interfering environments such as multiple-object scenes or non-negligible backgrounds.

Our method can be implemented using only the most basic multi-channel SPI and requires no additional special design to capture the full two-dimensional motion states of the target without any prior information. Our method can also be easily integrated with existing hyperspectral SPI systems[14–16], enabling hyperspectral imaging of moving targets. Furthermore, centroid information can be extracted from non-wavelength dimensions in multi-parameter optical imaging (such as polarization, photon time of flight, or combinations thereof), thereby enabling multi-dimensional optical information imaging. Based on three-dimensional light-field illumination[66], our method enables the three-dimensional tracking of small targets in microscopy and motion-compensated volumetric imaging.

## Methods

Each SPI measurement integrates information from the entire target image, and as the measurements accumulate (weighted summation of the basis patterns), the reconstructed image becomes progressively clearer. However, if the target moves during this process, the reconstruction of the image will be misaligned, leading to ghosting. Essentially, our method corrects the image's position to ensure that the SPI process accumulates correctly. Therefore, MC3-SPI comprises four stages: sampling process, data process, compensation process, and reconstruction process, corresponding to the four panels in Fig. 7. In the following, we discuss each step and its corresponding methodology in detail.

### Sampling process

The sampling process is shown in Fig. 7a. In the SPI system, the pattern modulates the target light field, and the measurement equation is given by the following:

$$D_P = k\langle O(x, y), P(x, y)\rangle + \epsilon, \tag{1}$$

where, $(x, y)$ represents the 2D Cartesian coordinates in the scene, $O$ is the intensity distribution of the object's image in the SPI, $P$ is the pattern of SPI. $k$ is system attenuation coefficient, $\epsilon$ represents the intensity of the background noise. Since the response of the single-pixel detector to light intensity is linear, we denote the SPI measurement results as $D_P$ for simplicity.

The signal is split into three parts using two dichroic mirrors (with wavelengths corresponding to the RGB channels), and the signals are collected by three single-pixel detectors. The three channels are independent and synchronized in time.

In order to obtain the projection coefficients of the target under the Fourier basis $F$ in a DMD-based SPI system, we employ the three-step phase-shifting method to separate the basis patterns. A 2D sinusoid pattern $P_\varphi$ is specified with its spatial frequency $(f_x, f_y)$ and initial phase $\varphi$, which are given by:

$$P_\varphi\left(x, y|f_x, f_y\right) = \frac{1}{2} + \frac{1}{2}\cos\left[2\pi\left(f_x x + f_y y\right) + \varphi\right], \tag{2}$$

where, $\varphi = 0, \frac{2\pi}{3}, \frac{4\pi}{3}$. As described in Supplementary Note 1, the optimal localization results can be obtained by using Fourier patterns with frequencies of $1/2M$ and $1/2N$ in the SPI localization system.

Then, we perform 0/1 binarization using the Sierra-Lite spatial dithering method with a serpentine error diffusion path to achieve the best localization accuracy.

In contrast, the Hadamard method is simpler because the Hadamard basis $H$ consists of $+1/-1$. Thus, it can be easily sampled with positive and negative patterns $p_{+/-}$ as shown below:

$$\begin{aligned} P_+ &= \frac{1 + H}{2}, \\ P_- &= \frac{1 - H}{2}. \end{aligned} \tag{3}$$

Our method combines the localization patterns and imaging patterns into a single frame, treating the target as stationary within that frame, thereby enabling the simultaneous acquisition of both the target's image information and its motion information. We discuss the ordering strategy of the imaging patterns in Supplementary Note 2. The specific configuration of the patterns is provided in Supplementary Note 4.

### Data process

The data process is shown in Fig. 7b. First, we need to extract the Fourier or Hadamard coefficients from the SPI measurement data. The Fourier coefficients $\widetilde{I}_F$ can be obtained from the results of the three-step phase shifting, as shown by the following equation:

$$\begin{aligned} \widetilde{I}_F\left(f_x, f_y\right) = &\left[2D_0\left(f_x, f_y\right) - D_{\frac{2\pi}{3}}\left(f_x, f_y\right) - D_{\frac{4\pi}{3}}\left(f_x, f_y\right)\right] \\ &+ \sqrt{3}j\left[D_{\frac{2\pi}{3}}\left(f_x, f_y\right) - D_{\frac{4\pi}{3}}\left(f_x, f_y\right)\right], \end{aligned} \tag{4}$$

where, $D_\varphi$ is the SPI measurement result of $p_\varphi$. In addition, the attenuation coefficient k is determined by the imaging system and is a constant, hence it is ignored.

Similarly, the Hadamard coefficient $\widetilde{I}_H$ is obtained by the following equation:

$$\widetilde{I}_H = D_+ - D_-, \tag{5}$$

where, $D_{+/-}$ is the SPI measurement result of $p_{+/-}$.

By observing the target through the three RGB channels, the motion and image information of the target are both acquired. According to the combination order of the patterns, the localization and imaging measurements are separated, and the localization results and image spectral information are calculated separately. Notably, the spectral information of the target image is distorted because of the motion of the target, which makes traditional inverse transformation methods ineffective.

Intuitively, when we use Fourier modes with frequencies of $1/2M$ and $1/2N$ to locate the target, the maximum phase of the Fourier coefficients is $\pi$, corresponding to the maximum FOV size. Ignoring the inherent phase of the Fourier coefficient, the phase of the measurement values and the absolute position of the object are linearly related, as shown by the following equations:

$$\begin{aligned} y &= \frac{M}{\pi}\arg\left(\widetilde{I}_F\left(0, \frac{1}{2M}\right)\right), \\ x &= \frac{N}{\pi}\arg\left(\widetilde{I}_F\left(\frac{1}{2N}, 0\right)\right), \end{aligned} \tag{6}$$

where, $arg()$ denotes the argument operation. $x$ and $y$ denote the absolute position coordinates of the target in the SPI system.

Thus, we can obtain the three RGB centroids of the target $(x_R, y_R)$, $(x_G, y_G)$, $(x_B, y_B)$, allowing us to determine the complete motion state of the target, as shown by the following equations:

$$
\begin{aligned}
\widetilde{x} &= \frac{(x_R + x_G + x_B)}{3}, \\
\widetilde{y} &= \frac{(y_R + y_G + y_B)}{3}, \\
\widetilde{\theta} &= atan2(y_R - y_G, x_R - x_G) \\
&\quad or\ atan2(y_G - y_B, x_G - x_B) \\
&\quad or\ atan2(y_B - y_R, x_B - x_R),
\end{aligned}
\tag{7}
$$

where, $atan2(y, x)$ is used to compute the angle between the point $(x, y)$ and the origin, returning a value in the range $-\pi$ to $\pi$. $\widetilde{x}$ and $\widetilde{y}$ represent the geometric mean coordinates of the centroids across the three RGB channels, which can be considered as the overall absolute coordinates of the target. $\widetilde{\theta}$ is the target rotation angle, which can be regarded as the direction angle of the target.

### Compensation process

The compensation process is shown in Fig. 7c. In the SPI system, the essence of motion compensation is the inverse translation and rotation of the modulation patterns. The specific procedures differ for Fourier and Hadamard basis patterns.

For Fourier SPI, the translation of a Fourier basis pattern can be described using the Fourier phase shift property. Specifically, the pattern is multiplied by a corresponding coefficient $e^{i\Delta\varphi}$[44,65]. On top of this, we apply the inverse rotation to the Fourier basis and the resulting compensated basis $F'$ is calculated as follows:

$$
\begin{aligned}
F'_i &= \mathcal{R}_{\theta_0 - \theta_i}\{\mathcal{T}_{(x_0 - x_i, y_0 - y_i)}\{conj(F_i)\}\} \\
&= \mathcal{R}_{\theta_0 - \theta_i}\{conj(F_i)\}e^{j2\pi\langle(f_x, f_y)_i, (x_i - x_0, y_i - y_0)\rangle},
\end{aligned}
\tag{8}
$$

represents the translation vector. This means that when elements are shifted past one end of the pattern, they wrap around and reappear at the opposite end. $\mathcal{R}_\theta\{\}$ denotes a counterclockwise rotation of $\theta$ about the origin $(x_0, y_0)$. $(x_0, y_0)$ is the coordinates of the origin (center position), which we set, and $(x_i, y_i)$ is the current absolute position coordinates of the target. $conj()$ denotes the conjugation operation, which originates from the IFT.

In Hadamard-SPI, we need to perform the same operations. The compensated basis $H'$ is shown as follows:

$$
H'_i = \mathcal{R}_{\theta_0 - \theta_i}\{\mathcal{T}_{(x_0 - x_i, y_0 - y_i)}\{H_i\}\},
\tag{9}
$$

Note that motion compensation methods are not unique; the method presented here is considered the simplest and most convenient for our purposes.

### Reconstruction process

The reconstruction process is shown in Fig. 7d. By utilizing the imaging coefficients $\widetilde{I}$ from the data process and the motion-compensated basis patterns $F'$ or $H'$ from the compensation process, either the IMCT method or the TVAL3 algorithm can be applied to reconstruct a clear image of the object.

The IMCT method reconstructs the target image $\widetilde{O}$ as follows:

$$
\widetilde{O} = \sum_i \widetilde{I}_i * \left[F'_i\ or\ H'_i\right].
\tag{10}
$$

In both Fourier and Hadamard imaging, the essence of the inverse Fourier transform (IFT) and inverse Hadamard transform (IHT) methods is the summation of a complete set of orthogonal bases weighted by their coefficients. When imaging objects undergoing composite motion, we perform the same operation. However, the basis patterns must be translated or rotated to compensate for the motion of the target, which weakens their orthogonality and affects the reconstruction. Note that the orthogonality of the Fourier basis patterns is better preserved after translation and rotation, making them more suitable for imaging objects undergoing composite motion (Supplementary Note 7).

Another common approach is to use the TVAL3 algorithm for motion-compensated target imaging[44,59,67]. The TVAL3 algorithm relies on an iterative solution using an augmented Lagrangian and alternating-direction method, effectively suppressing the chaotic speckles caused by the weakened orthogonality of the basis patterns, balancing image details and noise, enhancing image edges, and producing better reconstruction results[68]. However, using the TVAL3 algorithm for reconstruction requires more time and has poor real-time performance.

### Data availability

The MNIST dataset used for simulating the SPI localization accuracy data is publicly available at http://yann.lecun.com/exdb/mnist/.

### Code availability

The code for the reconstruction algorithms used in this study and the raw data are publicly available at https://github.com/ArthurWu1999/MC3-SPI.

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

## Acknowledgements
This work was supported by the Beijing Institute of Technology Research Fund Program for Young Scholars (Grant no.20212012).

## Author contributions
C.S. proposed the idea of the research work and the theoretical analysis. C.S. and Y.C. performed the numerical simulations. C.S., Y.C., X.Y., and S.L. carried out the experiment. C.S., Y.C., X.Y., S.L., and Q.Z. discussed the results. C.S. and Y.C. prepared the paper, which was revised by all authors. All authors reviewed and approved the final manuscript.

## Competing interests
The authors declare no competing interests. Y.C. is employed by the Chinese Academy of Agricultural Mechanization Sciences Group Co., Ltd.; this affiliation had no role in the work.
