## [Transparent Peer Review file · Communications Engineering]

Multichannel mult centroid motion-compensated single-pixel imaging of an 2D arbitrarily moving rigid-body target

Corresponding Author: Dr Xuri Yao

Version 0:

Reviewer comments:

Reviewer #1

(Remarks to the Author)

This paper introduces a novel single-pixel imaging (SPI) method called MC3-SPI, designed to simultaneously track and image objects undergoing arbitrary 2D planar motion. The method leverages six optimized Fourier patterns for localization and multichannel centroid tracking (RGB) to characterize motion dynamics. An inverse motion-compensated transformation (IMCT) is proposed for efficient reconstruction, achieving real-time performance (~0.042s for 128×128 images) while maintaining fidelity even when parts of the target exit the field of view. The system is validated through experiments. 2444Hz can be achieved for imaging.

The novelty of the work is that it leverage three waveband channels and difference between their localization to detect the rotation of the object. Therefore it can recognize the motion of target for both translation and rotation. To me this is an important contribution to the community of SPI.

The approach is demonstrated in experiment. Explained of the computation algorithms are in details for understanding. I suggest publication of the work.

Here are some detailed suggestions.

1. In the result section the authors want to add more quantitative details about the experiment, such as the imaging speed, et al.
2. Method for single channel localization should be briefly covered in the result section, even just the basic concept.
3. Line 360 "accumulated image" here accumulated may be replaced by "reconstructed" to avoid misunderstanding.
4. There are some existing SPI scheme that simulataneous locates and images target with random motion. The authors want to clarify the difference/improvement of their method compared to existing ones, ideally with more details.

Reviewer #3

(Remarks to the Author)

see attached PDF

Reviewer #4

(Remarks to the Author)

I co-reviewed this manuscript with one of the reviewers who provided the listed reports. This is part of the Communications Engineering initiative to facilitate training in peer review and to provide appropriate recognition for Early Career Researchers who co-review manuscripts.

Version 1:

Reviewer comments:

Reviewer #1

(Remarks to the Author)

The authors have answered my questions. The manuscript can be accepted for publication.

Reviewer #3

(Remarks to the Author)

see attached PDF

Reviewer #4

(Remarks to the Author)

I co-reviewed this manuscript with one of the reviewers who provided the listed reports. This is part of the Communications Engineering initiative to facilitate training in peer review and to provide appropriate recognition for Early Career Researchers who co-review manuscripts.N

Version 2:

Reviewer comments:

Reviewer #3

(Remarks to the Author)

Overall Comment

- **Length concern**: This concern is left to the Associate Editor to evaluate.
- **Readability**: The revised manuscript shows a improvement in overall readability.
- **Sub-pixel motion**: The issue has been addressed, but two concerns remain (see comment **Q3.7**).
- **Supplementary material referencing**: Several supplementary items are not referenced in the main manuscript (specifically Supp. 5, 9, 10, 12).

Point-by-Point Comments

Q2.1

OK.

Q3.4

Good, the authors are now more rigorous in their statements regarding the smallest sampling unit in motion-compensated SPI.

Q3.7

Sub-pixel motion

The authors have reflected on the sub-pixel problem and provided an interesting theoretical discussion for dynamic Fourier-SPI. However, regarding the numerical simulations:

1. The sub-pixel motion used in the simulations should be explicitly described.
2. This may be due to phrasing, but the statement **"PSNR and SSIM evaluations corroborate this finding and show that the impact increases at higher spatial frequencies, but remains small in magnitude"** appears overstated — especially for PSNR, where a **15 dB decrease is significant**. That said, the absolute PSNR remains high, which is positive.

Interpolation

The authors acknowledge that motion compensation requires interpolation ("e.g. bilinear or bicubic"), but do not state which method was used. This detail is essential for reproducibility and interpretation.

Cyclic translation

The authors take a different position, which could be acceptable. In that regard, it would help to clearly state that the target is assumed to be smaller than the FOV, and that the background is zero-filled. I'm not sure why the authors discussed the case of a purely translational motion. Neither did I understand why an image within the FOV satisfies periodic boundary conditions: if the target leaves to the right, it does not mean that it appears to the left? I feel like I am missing something.

Orthogonality

As suggested, the authors now assess orthogonality over the **entire pattern set**, which is considerably more interpretable.

Even though the authors agree that evaluating the full pattern set is more rigorous and added those results to the supplementary material, the main paper (and part of the supplementary) still present results based on **random subsets**. In particular, the improvement of Hadamard pattern orthogonality at higher sampling ratios may arise **from interpolation artifacts** that flatten high-frequency patterns into near-zero patterns. If so, the conclusion that Hadamard-SPI "requires higher sampling ratios" may be partly influenced by the metric rather than the underlying reconstruction performance.

In my opinion, it would strengthen the paper to:

- Move Supplementary Fig. 11 into the main document,
- Remove Fig. 2 (or optionally move it to the supplementary material, while being cautious about the results interpretation).

****Important Note****: It should be made explicit in the main manuscript that the orthogonality is evaluated between ****motion-compensated patterns and the static (original) patterns**** without the reader having to go through the supplemental materials.

I was surprised by the reported computation time ("about a week") and would be curious to know more about the hardware/software used, although this is mostly personal curiosity.

Q3.9

The labeling issue has been clarified in the response, but ****Figure 3b still shows "IFT" in the legend****, which should be corrected.

The authors also added an interesting analysis explaining the poor SSIM performance of IMCFT, showing that it is primarily due to the luminance term of SSIM. The filtering adjustment addresses this well.

Q5.3

The SPC FOV was not highlighted ****in the reconstructions****, but this is acceptable as it was only a suggestion.

Reviewer #4

(Remarks to the Author)

I co-reviewed this manuscript with one of the reviewers who provided the listed reports. This is part of the Communications Engineering initiative to facilitate training in peer review and to provide appropriate recognition for Early Career Researchers who co-review manuscripts.

Version 3:

Reviewer comments:

Reviewer #3

(Remarks to the Author)

We believe the manuscript is suitable for publication, with a few remaining minor concerns (listed below in order of decreasing importance):

1) Concerning the orthogonality analysis

Please justify why orthogonality is evaluated between motion-compensated patterns and the original static patterns, rather than between the motion-compensated patterns themselves.

2) Concerning cyclic translation

The authors state in their response that it is commonly assumed that the field of view fully covers the target and that the background is clean. However, their own "extend-FOV" method explicitly relaxes these assumptions by relying on partial capture of the target in at least two channels. For a broad-scope journal, the assumptions underlying each method should be stated precisely and consistently.

3) Concerning the description of sub-pixel motion

The description of sub-pixel motion as a "continuous random sub-pixel motion throughout the SPI sampling" remains vague. There are multiple ways to generate random sub-pixel motion, and the specific method used should be stated more explicitly.

4) Concerning the interpolation order

The manuscript still does not explicitly state which interpolation method is used, as the author claim an interested reader would refer to the provided code. It is acceptable, even though this information is important for interpreting the results and could easily be included in the manuscript.

Reviewer #4

(Remarks to the Author)

I co-reviewed this manuscript with one of the reviewers who provided the listed reports. This is part of the Communications Engineering initiative to facilitate training in peer review and to provide appropriate recognition for Early Career Researchers who co-review manuscripts.

Dear Editor,

We sincerely appreciate your efforts in coordinating the peer review of our manuscript (Manuscript ID: COMMS-25-0420A; Research Article). We are also grateful to the reviewers for their thoughtful comments and constructive suggestions. In response, we have carefully revised the manuscript. In the following, we provide our point-by-point responses to the reviewers' comments.

Reviewer # 1:

Q1: In the result section the authors want to add more quantitative details about the experiment, such as the imaging speed, et al.

Reply:

We sincerely appreciate the reviewer's careful evaluation and constructive comments, which have helped us improve the clarity and overall quality of the manuscript.

In the Results section, we have added information on the DMD flipping rate and the target speed relative to the measurement frame rate, so that the readers can understand the DMD operating speed in the experimental system and the level of the target's relative speed as perceived by our system.

On line 264 of the revised main manuscript:

To increase the target's velocity relative to the measurement process, we reduced the DMD flipping rate during the experiments, which introduced appreciable motion blur in conventional SPI reconstructions and thereby enabled a rigorous validation of the MC3-SPI motion-compensation method. In practice, we set the DMD flipping rate to 150 Hz, which corresponds to relative speeds of 2.70° per measurement frame and 1.20 mm per measurement frame.

On line 285 of the revised main manuscript:

The DMD flipping rate was set to 4500 Hz.

On line 295 of the revised main manuscript:

The DMD flip rate was set to 250 Hz, which corresponds to relative speeds of 1.08° per measurement frame and 0.72 mm per measurement frame.

On line 298 of the revised main manuscript:

The DMD flip rate was set to 2000 Hz.

In the real experiments, the occurrence and severity of motion blur are determined by the ratio of the target motion speed to the sensing speed of the system (the relative speed). A higher relative speed makes the motion more pronounced from the perspective of the system. To highlight the effectiveness of our MC3-SPI motion compensation, we reduced the DMD flipping rate as much as feasible to maximize the relative speed. In addition, because the random motion generated by the hand-cranked shaft is comparatively faster, we correspondingly increased the DMD flipping rate, although it remained at a relatively low level.

In the extend-FOV imaging experiment, the target undergoes random eccentric rotation at the field-of-view boundary. To handle motion screening under fully prior-free conditions, we repeatedly performed 10% low-frequency Fourier sampling. To achieve denser temporal coverage, we further increased the DMD flipping rate. Nevertheless, even in this extreme imaging scenario, the DMD flipping rate remained far below its maximum, indicating that the proposed method is not contingent on the DMD flipping rate.

On line 380 of the revised main manuscript:

Meanwhile, to achieve denser temporal coverage for the sampling 10% case, we slightly increased the DMD flipping rate to 3096 Hz, enabling completion of the sampling 10% within 4.76 s.

Moreover, in Supplementary Note 3 we provide the actual acquisition time and the theoretical minimum acquisition time for the Fourier sampling configuration, which offer readers more comprehensive and detailed data.

In our real experiments, we use a 50% sampling ratio and set the DMD to the lowest feasible flipping rate, for example 150 Hz. Under this setting, a single Fourier-SPI acquisition requires 492 s, thereby allowing the target ample time to undergo substantial translational and rotational motion. If the DMD operates at its maximum flipping rate, a single acquisition takes only 3.35 s.

Finally, in the Discussion section, we have provided a detailed analysis of the highest temporal resolution of the system (a DMD flipping rate of 22000 Hz), covering two cases, specifically motion sensing without image reconstruction and motion-compensated imaging. Building on a performance analysis of the IMCFT reconstruction method, we further discuss a target recognition scheme based on our approach and the corresponding recognition speed. We present these analyses to help readers assess the potential of the proposed method.

We appreciate the reviewer’s helpful guidance.

Q2: Method for single channel localization should be briefly covered in the result section, even just the basic concept.

Reply:

We thank the reviewer for pointing out this issue and for the helpful feedback that guided our revisions.

Our understanding and improvement of the centroid localization method for SPI system form the foundation of the manuscript and warrant a detailed description in the main text. We have therefore added a subsection titled “Absolute coordinate localization of the centroid” at the beginning of the Results section.

Absolute coordinate localization of the centroid

Each Fourier coefficient aggregates all pixels, making local spatial changes global in the spectrum. For a single rigid target, motion can be detected by tracking a small set of coefficients over time. Fourier-SPI recovers these coefficients from only a few measurements by using Fourier patterns and a single-pixel detector.

By the Spatial-Shift property of the Fourier transform, a translation $(\Delta x, \Delta y)$ induces phase shifts of $-2\pi f_x \Delta x$ and $-2\pi f_y \Delta y$ in the coefficients at $(f_x, 0)$ and $(0, f_y)$, respectively. This property yields only relative displacement. We extend this to absolute positioning by linearly mapping the measured phase directly onto the spatial coordinates.

We numerically evaluate the absolute localization error as a function of spatial frequency, using Fourier basis patterns $(f_x, f_y) = (f, 0)$ and $(0, f)$ for the x and y components respectively.

Figure 1: *Simulation of localization performance for Fourier location patterns at different spatial frequencies f . The pattern size is 256×256 with no up-sampling, i.e., $M = 256$. The targets comprise 500 handwritten digits randomly selected from the MNIST dataset. For each target size, each target is randomly positioned 100 times, the mean root-mean-square localization error (RMSE) is computed, and the curve of localization error as a function of target size is plotted. (a) Theoretical localization performance of Fourier basis patterns at different spatial frequencies f . (b) Empirical localization performance of Fourier location patterns at different spatial frequencies f under three-step phase shifting combined with Sierra-Lite dithering and a serpentine error-diffusion path.*

As shown in Fig. 1(a), we evaluate the absolute localization error as a function of target size and spatial frequency for Fourier basis patterns. Localization at frequencies $f > 1/M$ fails because

multiple periods appear within the field of view, making it impossible to identify which period contains the target. We therefore restrict absolute localization to $f \leq 1/M$. Within this range, lower frequencies yield higher accuracy. At any given frequency, larger targets produce larger errors. The bias arises because each non-DC Fourier coefficient carries an image-dependent intrinsic phase. This intrinsic phase decreases as the frequency decreases or as the target becomes smaller, thereby improving the accuracy of localization. In summary, Fourier basis patterns with lower spatial frequencies f generally yield more accurate absolute localization.

However, in DMD-based SPI, three-step phase shifting is required to separate the basis patterns, and spatial dithering is used to binarize grayscale patterns. Extending the analysis in Fig. 1(a), we ran simulations with Sierra-Lite dithering and a serpentine error-diffusion path, and the results are shown in Fig. 1(b). For typical target sizes of $80 \sim 120$ pixels (approximately $1/9 \sim 1/4$ of the field of view), the optimal Fourier localization pattern has a spatial frequency of $1/(2M)$ and yields a localization error of $\sim 1/3$ of a pixel.

Supplementary Note 1 provides a more detailed discussion of the error analysis of the Fourier localization method and the rationale for selecting the spatial frequency $1/(2M)$.

As noted above, we distilled from Supplementary Note 1 the most essential analyses, namely the ideal analysis of the Fourier localization method and the practical analysis based on DMD-SPI. The former guides the direction of method optimization, whereas the latter determines the final choice of the Fourier spatial frequency under practical constraints. More detailed optimizations at the application level, such as the optimization of spatial dithering, are presented directly as results, and the specific optimization procedures are retained in Supplementary Note 1.

Q3: Line 360 “accumulated image” here accumulated may be replaced by “reconstructed” to avoid misunderstanding.

Reply:

We thank the reviewer for pointing out this issue.

The term “accumulated image” is indeed prone to ambiguity and could be misunderstood as exposure in a conventional camera. We have adopted the suggestion and replaced “accumulated image” with “reconstruction of the image”.

However, if the target moves during this process, *the reconstruction of the image* will be misaligned, leading to ghosting.

We appreciate this helpful suggestion.

Q4: There are some existing SPI scheme that simultaneous locates and images target with random motion. The authors want to clarify the difference/improvement of their method compared to existing ones, ideally with more details.

Reply:

We appreciate the insightful comments that have helped improve the manuscript.

Among existing SPI approaches, the PCA method based on geometric moments aligns most closely with our technical approach, which, like ours, simultaneously captures translational and rotational information. Beyond what we state in the Introduction, namely that our method does not require additional patterns dedicated to orientation sensing and that the Fourier based localization shows stronger robustness to noise, we have added a dedicated subsection in the Supplementary Notes that provides a more detailed performance comparison.

Supplementary Note 10: Target angle estimation in SPI: Multichannel Multicentroid vs Principal Component Analysis

In SPI, an effective technique for characterizing a rotating object employs geometric moments and principal component analysis (PCA); the major axis of the object can be determined from just six low-order moments. PCA identifies the principal axis of the target by analyzing the variance of its image projections onto axes at multiple orientations. For a target image $I(x, y)$, the raw moments are shown as follows:

$$m_{pq} = \sum_x \sum_y x^p y^q I(x, y). \quad (1)$$

The centroid of the target (x_c, y_c) can be obtained from the zeroth-order and first-order moments:

$$x_c = \frac{m_{10}}{m_{00}}, \quad y_c = \frac{m_{01}}{m_{00}}. \quad (2)$$

Then calculate the second-order central moments, which measure the spread of the intensity distribution around the centroid:

$$\begin{aligned} \mu_{20} &= \sum_x \sum_y (x - x_c)^2 I(x, y) = m_{20} - x_c m_{10}, \\ \mu_{02} &= \sum_x \sum_y (y - y_c)^2 I(x, y) = m_{02} - y_c m_{01}, \\ \mu_{11} &= \sum_x \sum_y (x - x_c)(y - y_c) I(x, y) = m_{11} - x_c m_{01} = m_{11} - y_c m_{10}. \end{aligned} \quad (3)$$

Form the covariance matrix of these moments:

$$\mathbf{M} = \begin{pmatrix} \mu_{20} & \mu_{11} \\ \mu_{11} & \mu_{02} \end{pmatrix}. \quad (4)$$

The principal orientation θ (relative to the x -axis) is the angle of the major eigenvector of \mathbf{M} , which simplifies to:

$$\theta = \frac{1}{2} \arctan\left(\frac{2\mu_{11}}{\mu_{20} - \mu_{02}}\right). \quad (5)$$

This formula yields θ in radians; multiply by $180/\pi$ to express it in degrees.

To quantitatively compare the performance of our multichannel multi-centroid angle determination method with the PCA-based approach, we conducted data simulations, and the virtual trajectory of the target is presented in Supplementary Material 5. For an identical motion, selecting a different reference point yields a distinct translational trajectory, whereas the relative angular variation remains unchanged. Accordingly, this study evaluates only the angular accuracy. Simulations were performed for three distinct targets, and the corresponding results are presented in Supplementary Fig.2.

Figure 2: *Multichannel multi-centroid (MC2) vs principal component analysis (PCA): angular accuracy.* (a) The RMSE of MC2 is 0.2909° . The RMSE of PCA is 3.3903° . (b) The RMSE of MC2 is 0.5015° . The RMSE of PCA is 5.6262° . (c) The RMSE of MC2 is 0.4042° . The RMSE of PCA is 10.5520° .

The multichannel multi-centroid method achieves an angular estimation accuracy that is an order of magnitude higher than that of PCA. The classical definition of image moments is continuous, whereas in the SPI framework both the target images and the patterns are discretized. In the discrete domain, rotation is not rigorously defined, since rotating a discrete image typically yields an approximate and often irreversible transformation. Consequently, PCA founded on geometric moments can only approximate the target's orientation.

In addition, PCA based on geometric moments can, in theory, recover only the principal axis of the target, which means that it cannot determine its absolute orientation, therefore cannot distinguish a 180° flip. This ambiguity can be eliminated in engineering practice by invoking a continuity assumption. As long as the imaging system provides sufficiently high temporal resolution, the detected angles can be assumed to vary smoothly; consequently, when an abrupt change is observed, subsequent angles are corrected by adding or subtracting 180° . Doing so yields a continuous rotational trajectory and allows the full rotational motion of the target to be reconstructed. In Supplementary Fig.2, the approach is applied to correct the angles obtained from PCA, and the algorithmic procedure is presented as Algorithm 1.

Algorithm 1 *Angle Continuity Correction for θ_{GM}*

Require: A sequence of angles $\theta_{GM}[1 \dots N]$ in degrees

Ensure: A corrected sequence with no large jumps ($> 150^\circ$)

```

1: for  $i = 1$  to  $N - 1$  do
2:   if  $\theta_{GM}[i + 1] - \theta_{GM}[i] > 150$  then
3:      $\theta_{GM}[i + 1] \leftarrow \theta_{GM}[i + 1] - 180$ 
4:   end if
5:   if  $\theta_{GM}[i + 1] - \theta_{GM}[i] < -150$  then
6:      $\theta_{GM}[i + 1] \leftarrow \theta_{GM}[i + 1] + 180$ 
7:   end if
8: end for

```

As noted above, for orientation estimation within the SPI framework, the multichannel multicentroid method outperforms the PCA approach based on geometric moments. However, SPI is capable of directly retrieving image moments of arbitrary order, eliminating the need for secondary post-processing of images. Geometric moments, especially central moments, can be combined to yield the Hu invariants, which remain unchanged under translation, scaling, and rotation and therefore offer a basis for image analysis. Thus, SPI that exploits geometric moments enables target detection without images and capitalizes on the inherently high throughput and high sensitivity of SPI.

Beyond the approaches discussed above, there are additional viable routes. One option is to integrate mature motion sensors into an SPI system, where a simple calibration enables simultaneous localization and imaging. Similarly, a high-speed CMOS camera can first detect target motion and then guide the SPI system to perform motion-compensated reconstruction. Within a classical SPI framework, an

analogous idea is to reconstruct a low-resolution preview from a small number of patterns to sense translation and rotation. This approach requires a trade-off between the accuracy of motion sensing and the temporal resolution of the system, and the achievable temporal resolution is often lower than that of a non-imaging motion detection scheme that estimates motion from a small number of patterns. We have added and refined the corresponding discussion in the Introduction.

When targets exhibit both translational and rotational motion, *a direct and effective strategy is to use well-established motion sensors or a CMOS camera to detect the motion in real time and calibrate it with the SPI system. In a classical SPI architecture, an analogous approach is to reconstruct a low-resolution preview image to estimate the motion, which inevitably entails a trade-off between localization accuracy and temporal resolution, limiting adaptability and accuracy for high-speed targets.*

We appreciate the reviewer's patience and constructive guidance.

Reviewer # 3 and # 4:

1.# Validity: Does the manuscript have flaws which should prohibit its publication? If so, please provide details.

Q1.1: The paper is not positioned within the dynamic single-pixel imaging literature. As will be explained in the next section, the paper overlooks key approaches (e.g. similar camera designs and alternative motion estimation strategies).

Reply:

We thank the reviewers for pointing out this issue.

In the Introduction, we have added a discussion of the existing literature and methods in dynamic SPI and cited the relevant work.

When targets exhibit both translational and rotational motion, *a direct and effective strategy is to use well-established motion sensors or a CMOS camera to detect the motion in real time and calibrate it with the SPI system [9]. In a classical SPI architecture, an analogous approach is to reconstruct a low-resolution preview image to estimate the motion, which inevitably entails a trade-off between localization accuracy and temporal resolution, limiting adaptability and accuracy for high-speed targets [6,7,8].*

Although these custom hardware systems have pattern projection rates of more than 1Mfps, the trade-off between spatial resolution and measurement time in SPI remains unresolved, the imaging frame rate is significantly lower than the modulation frequency of the device, approximately 1/1000 of it. *At present, several approaches have been proposed to mitigate the above trade-off, including incorporating the spectral dimension to enhance sparsity and thereby increase the compression ratio [2]; employing customized patterns to enable efficient low-resolution previews together with convex optimization for high-resolution video reconstruction [7]; and dynamically adjusting the balance between temporal and spatial resolution according to frame complexity [10].*

We again appreciate the reviewers' patience and guidance.

Q1.2: The manuscript claims to handle arbitrary motion, including in the title ("...arbitrarily moving target"), despite relying on a planar rigid motion assumption.

Reply:

We sincerely thank the reviewers for the careful evaluation and constructive suggestions.

We agree with the comment. Our approach is indeed based on a two-dimensional rigid body motion model, and we apologize for the earlier lack of clarity. We have accordingly revised the title to state explicitly that the method is limited to rigid targets and addresses arbitrary two-dimensional motion.

Because any two dimensional motion can be decomposed into translation and rotation with respect to an arbitrary reference point, we retain the wording "arbitrary" to reflect this kinematic fact.

Multichannel multicentroid motion-compensated single-pixel imaging of *an 2D* arbitrarily moving *rigid-body* target

We sincerely appreciate the reviewers' guidance and suggestions.

2.# Originality and significance: If the conclusions are not original, please provide relevant references. On a more subjective note, do you feel that the results presented are of immediate interest to many people in your own discipline, and/or to people from several disciplines?

Q2.1: In [0], the authors pioneered the use of Fourier localization patterns to estimate the center of mass of a scene. It is imperative that the manuscript explicitly states whether the work builds on or extends [0], thereby clarifying the novelty of the present contribution.

Reply:

We thank the reviewers for the valuable suggestion.

Ref. [0] is a 2024 publication by our group. In [0], we first confirmed that Fourier patterns at different spatial frequencies yield different localization errors, and that selecting an appropriate spatial frequency enables absolute localization. Specifically, provided that periodic structures are avoided in the pattern design, Fourier basis patterns can, to some extent, enable absolute position localization of the target object. However, we also observed a decrease in performance for larger targets. In the present work, the estimation of the orientation angle relies on the relative positions of centroids across channels, which requires absolute-coordinate localization of the centroids. For real objects, the inter-channel intensity distributions are more complex and often yield more clustered centroids, which demands higher localization accuracy to ensure reliable angle estimation. These considerations motivated us to further improve the performance of the Fourier localization method. The Fourier location pattern used in this paper is an optimization based on [0]. In particular, for larger targets ($1/9 \sim 1/4$ of the FOV), Fourier patterns with spatial frequencies of $1/2M$ and $1/2N$ (different with [0]) further improve the localization accuracy, reaching an error of about $1/3$ of a pixel.

Following the reviewers' suggestion, we explicitly state in the Discussion that our localization method is based on [0] with further performance optimization, achieving sub-pixel accuracy for larger targets ($1/9 \sim 1/4$ of the field of view, a common target size) .

However, with an appropriate configuration, the Fourier localization method can approximate the absolute position of the target [0]. *Building upon this work, we conducted further research that improved the accuracy of the Fourier localization method.* In this work, we employ Fourier basis functions with spatial frequencies of $1/2M$ and $1/2N$, and apply the Serria-Lite method with a serpentine diffusion path, achieving an optimal localization accuracy of approximately $1/3$ pixels when the target size accounts for approximately $1/9 \sim 1/4$ of the FOV.

We appreciate the reviewers' helpful guidance.

Q2.2: In general terms, the proposed approach is not situated within the domain of single-pixel imaging techniques. It is imperative that the manuscript explicitly relates the proposed approach to prior work in hyperspectral single-pixel imaging, stereo single-pixel techniques, and other multichannel designs (see references [1–4]).

Reply:

We thank the reviewers for the thoughtful observation.

We agree that strictly speaking, the proposed method is a multi-channel single-pixel imaging system. In this work, the rotational information is inferred from multiple centroids, each acquired through a specific channel. Nevertheless, each centroid is obtained within the SPI framework, where the optical field is modulated by a DMD and measured with a single pixel detector.

The scheme can also be implemented with temporal multiplexing. Using a high-speed filter wheel to realize time domain multispectral channels, a single single-pixel detector suffices (Jia, Tong, Dongyue Chen, Ji Wang, and Dong Xu. 2018. "Single-Pixel Color Imaging Method with a Compressive Sensing Measurement Matrix" Applied Sciences 8, no. 8: 1293.). Moreover, centroid acquisition is not limited to one optical dimension. Multiple dimensions of optics can be combined to define distinct centroids; for example: intensity, polarization, and photon time of flight may provide three separate centroids, respectively.

Our intention is to emphasize the core contribution, namely retrieving the rotational information of a moving target and addressing the challenge of imaging a target undergoing random two-dimensional

motion with a single-pixel camera. For this reason, we frame the work within the SPI domain rather than presenting it solely in the context of spectral imaging.

We thank the reviewers for the helpful reminder. Our approach serves as a general framework that can be readily combined with existing hyperspectral techniques to enable hyperspectral imaging of moving targets. We discuss this in the Discussion section and cite [1–3].

Our method can be implemented using only the most basic multichannel SPI and requires no additional special design to capture the full two-dimensional motion states of the target without any prior information. Our method can also be easily integrated with existing hyperspectral SPI systems [1–3], enabling hyperspectral imaging of moving targets.

As the reviewers pointed out, our method relies on a two-dimensional rigid body motion model. Extending it to three dimensions would require a three-dimensional rigid body model and the estimation of multiple centroids of a 3D target, which typically calls for spatial light field modulation techniques. Reference [4] reconstructs the 3D surface using detectors at multiple viewing angles, which is not directly compatible with our framework. We therefore cite this work in the Introduction.

Additionally, with SPI, multidimensional information can be acquired using a single-pixel detector [1–3] and allowing 3D imaging [4].

We appreciate the reviewers’ guidance.

Q2.3: Furthermore, the discourse on motion estimation is narrowly centred on methodologies employing localization patterns, while disregarding numerous well-established strategies in dynamic single-pixel imaging.

It is notable that the following approaches are not discussed : previewbased methods that estimate motion from low-resolution reconstructions [6–8], hybrid systems incorporating conventional CMOS cameras [9], and relevant video compressed sensing techniques [2, 7, 10]. Furthermore, related work that addresses the reconstruction of the scene beyond the FOV, such as [5], could be mentioned to clarify the distinction between motion estimation and out-of-FOV reconstruction.

Reply:

We thank the reviewers for the helpful reminder and suggestions.

We apologize for the oversight that led us to omit several established dynamic SPI approaches. In line with the comment, we have added the relevant citations and included a concise and appropriate discussion in the revised manuscript.

When targets exhibit both translational and rotational motion, a direct and effective strategy is to use well-established motion sensors or a CMOS camera to detect the motion in real time and calibrate it with the SPI system [9]. In a classical SPI architecture, an analogous approach is to reconstruct a low-resolution preview image to estimate the motion, which inevitably entails a trade-off between localization accuracy and temporal resolution, limiting adaptability and accuracy for high-speed targets [6–8].

Although these custom hardware systems have pattern projection rates of more than 1Mfps, the trade-off between spatial resolution and measurement time in SPI remains unresolved, the imaging frame rate is significantly lower than the modulation frequency of the device, approximately 1/1000 of it. At present, several approaches have been proposed to mitigate the above trade-off, including incorporating the spectral dimension to enhance sparsity and thereby increase the compression ratio [2]; employing customized patterns to enable efficient low-resolution previews together with convex optimization for high-resolution video reconstruction [7]; and dynamically adjusting the balance between temporal and spatial resolution according to frame complexity [10].

Reference [5] considers a continuous object larger than the field of view, such as the cerebral cortex, and reports that when the object is moving, discretizing in physical space and reconstructing under an extended field of view yields the lowest spectral bias. This is related to our setting, where we image a boundary-motion target whose size is smaller than the field of view and that is never fully

observed within the field. The Fourier coefficients acquired with conventional Fourier basis patterns can be regarded as approximations to the target Fourier coefficients defined on the extended field of view. The reconstruction then employs a more empirical IMCFT scheme, which can be interpreted as motion compensating the original field of view patterns to their correct locations within the extended field while relocating the reconstructed target to the center of the field of view. To demonstrate the connection, we cite [5] in the relevant paragraph.

For a target moving across the field-of-view boundary, the SPI-acquired coefficients remain valid within an appropriately defined discrete domain [5], and we design an extended field-of-view scenario accordingly.

We sincerely appreciate the reviewers' patience and guidance.

Q2.4 From my perspective, the methodology under discussion is relevant and of interest to researchers working in single-pixel imaging, computational imaging, and motion estimation. The discussion thus initiated is of particular pertinence with regard to the utilisation of multichannel information for motion tracking. Nevertheless, it is my conviction that the results and their presentation still require further development and clarification before the full potential and significance of the contribution can be appreciated by a broader audience.

Reply:

We thank the reviewers for the positive assessment.

Following this suggestion, we revised the closing paragraph of the Discussion to better convey the potential and significance of the approach. Building on its flexibility and broad compatibility, we outline the application prospects to clarify its anticipated impact.

Our method can be implemented using only the most basic multichannel SPI and requires no additional special design to capture the full two-dimensional motion states of the target without any prior information. Our method can also be easily integrated with existing hyperspectral SPI systems [1–3], enabling hyperspectral imaging of moving targets. Furthermore, centroid information can be extracted from non-wavelength dimensions in multi-parameter optical imaging (such as polarization, photon time of flight, or combinations thereof), thereby enabling multidimensional optical information imaging. Based on three-dimensional light-field illumination, our method enables the three-dimensional tracking of small targets in microscopy and motion-compensated volumetric imaging.

We appreciate the reviewers' patience and guidance.

3.# Data & methodology: Please comment on the validity of the approach, quality of the data and quality of presentation. Please note that we expect our reviewers to review all data, including any extended data and supplementary information. Is the reporting of data and methodology sufficiently detailed and transparent to enable reproducing the results?

Validity of the Approach:

The proposed approach is conceptually sound and addresses the challenge of motion estimation in single-pixel imaging using a novel three-detector system. The utilisation of colour-channel separation for motion inference represents a novel and well-justified approach.

Q3.1: The assumption of rigid motion is a reasonable starting point and is common in the field (especially for short acquisitions), but the authors should more explicitly discuss its limitations in the context of their system. For instance, in symmetric object configurations (e.g., two identical features rotating around a centre), the centre of mass may remain unchanged, resulting in erroneous angle estimations. Moreover, the method appears to be applicable only in circumstances where no additional objects enter the field of view. This is due to the fact that the extended FoV capability is reliant upon intensity filtering in relation to a fixed reference frame. Furthermore, the efficacy of the method in real-world scenarios with non-zero backgrounds remains to be elucidated.

Reply:

We thank the reviewers for the thoughtful observation.

We agree with this point. Our orientation estimation method may be ineffective for centrosymmetric targets, and handling strong interference such as multiple objects in the field of view or a non-negligible background remains challenging. To clarify the scope and limitations, we have added a dedicated paragraph in the Discussion that explicitly addresses these cases.

Although MC3-SPI can effectively sense arbitrary two-dimensional motion of rigid targets, its applicability remains limited by intrinsic methodology and system-level constraints. First, our motion-sensing approach imposes constraints because the target orientation is inferred from the relative positions of centroids across different channels. This essentially requires cross-channel intensity distributions to break central symmetry so that the centroids of the channel are measurably separated. In addition, the inherently global sensing nature of the SPI system limits precise target acquisition and imaging in interfering environments such as multiple-object scenes or non-negligible backgrounds.

To further analyze the issue and suggest possible solutions, we partition it into two categories. First, the inability to handle centrosymmetric targets is a limitation of our method. Second, the difficulty in operating under multiple objects or strong backgrounds stems more from the inherent constraints of the SPI.

Based on this separation, for the first category we propose an extension of channel selection that is not confined to a single optical dimension. Instead, we combine multiple optical dimensions to measure the same target and thereby extract latent non-centrosymmetric features. For example, when the target is centrosymmetric in the wavelength dimension, one may incorporate polarization information or jointly use intensity and polarization to break the symmetry. For the second category, we develop two solutions that address a static strong interfering object and a static strong interfering background, respectively, and we validate them with numerical simulations. These solutions rely on task-specific optimizations of SPI patterns tailored to the corresponding scenarios. We have added this section in the supplementary Note 11.

Supplementary Note 11: Discussion and analysis of the limitations of MC3-SPI

Although MC3-SPI can efficiently sense target motion and orientation, its applicability is constrained by several fundamental assumptions. On the one hand, some limitations arise from our method itself, which inherently requires the intensity distributions across channels to break central symmetry so that the channel centroids are measurably separated for rotation sensing; on the other hand, intrinsic constraints of the SPI system preclude accurate target capture and imaging in interfering environments such as multi-object scenes or strong backgrounds.

However, owing to the flexibility of MC3-SPI, customized designs can be incorporated within its framework to remain effective in the aforementioned specific scenarios.

Limitations for centrosymmetric targets

Our method relies on multichannel observations: the target exhibits different intensity distributions across channels, resulting in measurable offsets between the channel centroids and the global centroid. By analyzing the directional angles between each channel centroid and the global centroid, the instantaneous orientation of the target can be inferred. However, if the target's signal distribution is

centro-symmetric across all channels, the channel centroids coincide with the global centroid, providing no separable centroid information and rendering orientation estimation infeasible. Consequently, MC3-SPI functions reliably only when the signal distribution in at least one channel breaks central symmetry.

However, perfectly centrosymmetric objects are rare, and finer channel configurations can be adopted, for example extending from RGB to multispectral or even hyperspectral channels, to reveal object asymmetries. Moreover, even when an object is perfectly centrosymmetric in a single optical dimension, multiple optical dimensions can be jointly exploited, such as wavelength and polarization, to determine the wavelength centroid and the polarization centroid and thereby infer the target’s rotational information. Therefore, our method is theoretically effective across a wide range of objects.

When the object is completely symmetrical, it is worth mentioning that the localization of the target position remains accurate. This suggests that, after applying translational compensation, the angular variation of the target during SPI acquisition may be estimated from the quality of the reconstructed images. Furthermore, the proposed IMCFT provides a fast and effective reconstruction approach to support this estimation. Despite, the introduction of optimization algorithms incurs a prohibitive computational cost, causing real-time performance to be lost entirely.

In summary, our target angle sensing method does have limitations, yet we maintain that it nonetheless retains appreciable value.

Limitations when multiple objects are present in the field of view

Conventional SPI exhibits a typical global sensing mechanism: Each measurement integrates information from the entire field of view, and without prior knowledge of target location, it cannot focus on specific objects. Existing SPI-based localization methods typically extract only a “global centroid”, which is an aggregate quantity rather than the spatial coordinates of an individual target. When multiple targets or strong background interference are present in the field, the global centroid is influenced by all signal sources, rendering localization of the target of interest ineffective. Let pixels (x_i, y_i) for $i = 1, \dots, n$ have weights (masses) m_i . The global centroid (\bar{x}, \bar{y}) is defined as follows:

$$\bar{x} = \sum_{i=1}^n w_i x_i \quad \text{and} \quad \bar{y} = \sum_{i=1}^n w_i y_i, \quad w_i = \frac{m_i}{\sum_{j=1}^n m_j}. \quad (6)$$

When there is only a single target in the field of view, the global centroid defined by the above equation is evidently equivalent to the centroid of that target. However, when multiple targets are present in the field of view, the global centroid becomes equivalent to the intensity-weighted average of the individual target centroids.

Assume that n objects are present within the field of view. Let the centroids of objects (\bar{x}_i, \bar{y}_i) for $i = 1, \dots, n$ have weights (masses) M_i . The global centroid output by the SPI location method (\bar{x}, \bar{y}) is defined as follows:

$$\bar{x} = \sum_{i=1}^n w_i \bar{x}_i \quad \text{and} \quad \bar{y} = \sum_{i=1}^n w_i \bar{y}_i, \quad w_i = \frac{M_i}{\sum_{j=1}^n M_j}. \quad (7)$$

It follows that any non-target object present in the field of view “pulls” the SPI localization result, displacing it from the desired position. When such interfering objects are stationary, their influence can be entirely eliminated by a simple procedure. With prior knowledge of the precise positions and sizes of these static objects (a reasonable assumption since the objects are static), we set the corresponding regions of the SPI measurement patterns to zero, thereby rendering the interferences invisible to the SPI system. A straightforward numerical simulation was carried out to validate this idea, as shown in the Supplementary Fig.3.

Figure 3: Numerical simulation results of MC3-SPI in the presence of a strongly interfering object that flickers randomly at the center of the field of view, together with the pattern optimization strategy used to suppress the interferer and the corresponding MC3-SPI results. (a) Direct motion sensing and imaging results in the presence of a flickering strong interferer. (b) Motion sensing and imaging results obtained with optimized patterns in the presence of a strongly interfering flickering object. (c) Pattern optimization method that sets the regions corresponding to interfering object in the patterns to zero, thereby eliminating its influence entirely.

The initial intensity of the interfering object is set to twice that of the target, and in each simulated measurement, each of its color channels is individually multiplied by a random factor to emulate random flickering. The interfering object is placed at the center of the field of view. This constitutes severe interference and leads to complete failure of the method, as shown in Supplementary Fig.3 (a).

To eliminate the influence of non-target objects, we set the corresponding regions of the patterns to zero, as shown in the Supplementary Fig.3 (c). In this way, SPI completely ignores any variations in those regions without requiring additional modifications. As a result, our method accurately captures the target’s motion information and reconstructs a high-quality image of the target, as illustrated in the Supplementary Fig.3 (b).

However, when the interfering objects in the scene also move unpredictably, the above strategy becomes ineffective. More sophisticated approaches are required to isolate the target signal, for example, exploiting its distinctive emission wavelength or polarization characteristics.

Limitations in the presence of a nonzero background

Another scenario involves a target moving against a substantial background. Although this situation resembles the multi-object case, the crucial difference is that the target obscures a small region of the background. Consequently, the global centroid determined by the SPI system becomes an intensity-weighted average of the unoccluded background and the target centroid. When addressing this problem, the occluded portion of the background can be modeled as an object with negative weight, thereby reducing the background–occlusion scenario to the multi-object case.

Let the background centroid be denoted (\bar{x}_b, \bar{y}_b) with weight M_b , the target centroid (\bar{x}_t, \bar{y}_t) with weight m_t , and the centroid of the background region occluded by the target (\bar{x}'_t, \bar{y}'_t) with weight m'_t . The global centroid output by the SPI location method (\bar{x}, \bar{y}) is defined as follows:

$$\begin{aligned} \bar{x} &= \frac{m_t}{m_t + M_b - m'_t} \bar{x}_t + \frac{M_b}{m_t + M_b - m'_t} \bar{x}_b + \frac{-m'_t}{m_t + M_b - m'_t} \bar{x}'_t \\ \text{and } \bar{y} &= \frac{m_t}{m_t + M_b - m'_t} \bar{y}_t + \frac{M_b}{m_t + M_b - m'_t} \bar{y}_b + \frac{-m'_t}{m_t + M_b - m'_t} \bar{y}'_t. \end{aligned} \tag{8}$$

We now concentrate on the case of a static, invariant background. Adopting the reasonable prior that the background has been fully characterized, the initial step is to subtract its contribution from every SPI measurement. It is important to note that the occluded part of the background is never captured by SPI measurements; therefore, subtracting the complete background effectively introduces a negative artifact that coincides with the target. This negatively weighted artifact is effectively equivalent to a positively weighted one located at a negative position, and its presence exerts a pronounced influence on the system. Numerical simulations were conducted for this scenario and the results are presented in Supplementary Fig.4.

Figure 4: Numerical simulation in the presence of a non-negligible static background, with motion sensing and imaging results at three representative stages of the MC3-SPI optimization process. (a) Motion sensing and imaging results in the presence of background. (b) Motion sensing and imaging results with direct subtraction of the complete background. (c) Motion sensing and imaging results with iterative application of Algorithm 2 to appropriately eliminate background effects.

In the simulation, we set the total background intensity equal to that of the target. Although the background intensity remained relatively low, it still caused our method to fail, as shown in the Supplementary Fig.4 (a).

Supplementary Fig.4 (b) shows the result obtained by subtracting the full background contribution from the SPI measurements. The background intensity is dispersed through the field of view; the region masked by the target should have only a minor weight. Its negative weight amplifies its effect, producing severe distortions in the reconstructed target image.

Accordingly, we use the IMCFT result in Fig.4 (b) as a reference, generate a mask by thresholding, and apply it to the background to avoid the negatively weighted artifact introduced by subtracting

the full background from the SPI measurements. The newly corrected SPI measurements are then used for the IMCFT reconstruction. By iterating these steps an appropriate number of times, an undistorted target image can be obtained, as shown in Fig.4 (c). The algorithmic procedure is presented as Algorithm 2.

Algorithm 2 *Iterative background correction*

Require:

Location patterns P_l , imaging patterns P_{im} ,
Raw measurements $\{R_l, R_{im}\}$,
Fullcolour background matrix *background*.

Ensure: Get the corrected measurements $\{R'_l, R'_{im}\}$

- 1: Initialize *mask* to all zeros
- 2: Initialize $\Delta x, \Delta y, \Delta \theta$ to all zeros
- 3: **for** iteration $\ell = 1$ **to** 3 **do**
- 4: $mask \leftarrow 1 - mask$
- 5: **for** frame number $i = 1$ **to** *num* **do**
- 6: $mask_i \leftarrow \text{cirshift}(\text{imrotating}(mask, -\Delta\theta(i)), [\Delta y(i), \Delta x(i)])$
- 7: $R'_l(\cdot) \leftarrow R_l(\cdot) - P_l(\cdot)(mask_i \odot background)$
- 8: $R'_{im}(\cdot) \leftarrow R_{im}(\cdot) - P_{im}(\cdot)(mask_i \odot background)$
- 9: **end for**
- 10: $\Delta x, \Delta y, \Delta \theta \leftarrow \text{MC2}(R'_l)$
- 11: $U \leftarrow \text{IMCFT}(R'_{im}, [\Delta x, \Delta y, \Delta \theta])$
- 12: $mask \leftarrow \text{im2double}(\text{im2gray}(U))$
- 13: $mask(mask < 0.2) \leftarrow 0$
- 14: $mask \leftarrow mask \otimes \text{ones}(13 - 2\ell)$
- 15: $mask(mask \neq 0) \leftarrow 1$
- 16: **end for**

Notably, the proposed IMCF method enables rapid reconstruction of two-dimensional targets undergoing compound motion and is particularly well suited for loop-structured implementations.

Additionally, in the Algorithm 2 we convolve the image with all-ones kernels of different sizes to blur and thereby expand the object, ensuring that the regions set to one in the mask completely cover the target. In this case, we prefer leaving residual background to over-subtracting it; the former introduces peripheral noise around the target, whereas the latter yields a negatively weighted artifact with a strong impact.

However, our method is effective only when the background remains static. It will inevitably fail if the background undergoes complex variations or if multiple targets enter the scene simultaneously. We require an efficient approach to completely eliminate background contributions. Axial discrimination using the photon time of flight is a promising option. Once the target distance is determined, signals originating from other ranges can be rejected in full. Because the background is often not coplanar with the target, this strategy removes background interference regardless of how the background varies.

We appreciate the reviewers' guidance.

Quality of the Data and Presentation:

Q3.2: I suggest explaining the target's movement in "the real experiments". Also, the sentence online 189 ("To further validate... we conducted... with real targets") is confusing. It makes it seem like the earlier experiment was synthetic, but this doesn't seem to be true.

Reply:

We sincerely thank the reviewer for the careful reading and constructive suggestion.

We respectfully note that the motion in the real experiments has already been described in the main text. Both experimental sets adopted the same design strategy. Each set comprises two motion modes, one emphasizing rotational motion with concurrent translation and the other emphasizing random motion. We apologize for the earlier lack of clarity and have optimized the description of the target motion.

On line 260 of the revised main manuscript:

In the first trajectory, the target *performs torsional oscillation about a reciprocating axis* with the maximum average angular speed and the maximum average linear speed of the two-axis translation stage, which are 45° s^{-1} and 20mm s^{-1} , respectively, with amplitudes of 60° (rotation) and 15mm (translation).

On line 283 of the revised main manuscript:

the second trajectory is a random eccentric (*off-centroid*) rotation, with rotation amplitude and angular velocity randomly varying.

On line 292 of the revised main manuscript:

For the Mario toy, the motion trajectory consisted of *torsional oscillation and reciprocation* with an average angular velocity of 30° s^{-1} and an average linear velocity of 20mm s^{-1} , with amplitudes of 60° (rotation) and 15mm (translation).

On line 297 of the revised main manuscript:

For the rocket, the motion follows a random reciprocating *oscillation*.

We recognize that our wording at the transition between the two experimental sets could be misunderstood. To help readers more clearly understand the differences between the two experiments and the purpose of the second setup, we have refined and clarified the relevant statements.

Real-world objects are often more complex. To validate the effectiveness of MC3-SPI on complex targets, we track and image complex objects undergoing compound motion, using a Mario figurine and a custom cartoon rocket as targets.

We appreciate the reviewers' guidance.

Q3.3: Figure 2 is similar to Figure 3, so it can be left out of the main paper to make space for more valuable content from the extra material (e.g., some details on the improved Fourier localization method).

Reply:

We sincerely thank the reviewers for the careful evaluation and constructive suggestions.

In response to this comment, we have incorporated the improved Fourier localization method and the orthogonality analysis of different basis patterns into the main text, and we have streamlined the exposition to highlight the most salient technical content.

Figure 2 presents a more idealized experiment that uses the RGB “BIT” letters as the target, where the three-channel centroids are well separated, with large inter-channel distances. This ideal setting focuses on the localization–reconstruction pipeline and provides an intuitive validation, enabling readers to understand the experimental structure quickly, without confounding factors. The results agree with expectations and validate the method, allowing readers to develop an intuitive understanding of our approach in a simple, interpretable case.

By contrast, Figure 3 examines more complex real-world objects whose colors, shapes, and transitions between patches are more intricate, and the cross-channel centroids are more tightly clustered. This imposes stricter localization accuracy requirements to resolve nearby centroids. Under realistic conditions, the results more clearly reveal differences in image fidelity across reconstruction methods, which motivates differentiated comparisons across basis patterns and leads to improved reconstruction

strategies for challenging scenarios.

This progression from a simple, interpretable target to complex real scenes enhances readability and builds the argument step by step. In the main text, our discussion of the two figures is organized in a progressive manner rather than repeating the same content in different formats. As space permits in the revised manuscript, we respectfully retain Figure 2, which we believe improves readability by providing a controlled baseline.

We again thank the reviewers for the careful evaluation of the manuscript.

The supplementary data useful, but it also causes problems:

Q3.4: Supp. Fig. 1: It is important to understand the proposed localization method, so Supp. Fig. 1 should be moved into the main text. We don't know if the localization was checked using just one channel or the full three-channel device. This needs to be explained, especially the performance gain from using multiple channels. The authors currently check the position of the centre of mass, but because the goal is to estimate how much something has moved, it may be better to check the errors in the movements between frames. This would show how well the system is working.

Reply:

Thank you for the careful review and constructive comments.

Following this advice, we added at the beginning of the Results section a subsection titled "Absolute coordinate localization of the centroid" that analyzes the centroid localization method in an SPI system together with our optimization process. We selected the most essential components of Supplementary Figure 1, namely panels (a) and (b), refined the simulations, and placed them in the main text. The first analysis evaluates the localization performance of Fourier basis patterns with different spatial frequencies. The second analysis evaluates Fourier location patterns generated by three-step phase shifting and spatial dithering binarization based on the Sierra-Lite method with serpentine error diffusion path. The former is an idealized theoretical model that guides the initial choice of the spatial frequency of Fourier patterns. The latter is a practical simulation that incorporates DMD-induced pattern constraints to determine the final settings. We also added the necessary logical links and indicate that the spatial dithering method used corresponds to the results shown in Supplementary Figure 1(c) and 1(d).

Absolute coordinate localization of the centroid

Each Fourier coefficient aggregates all pixels, making local spatial changes global in the spectrum. For a single rigid target, motion can be detected by tracking a small set of coefficients over time. Fourier-SPI recovers these coefficients from only a few measurements by using Fourier patterns and a single-pixel detector.

By the Spatial-Shift property of the Fourier transform, a translation $(\Delta x, \Delta y)$ induces phase shifts of $-2\pi f_x \Delta x$ and $-2\pi f_y \Delta y$ in the coefficients at $(f_x, 0)$ and $(0, f_y)$, respectively. This property yields only relative displacement. We extend this to absolute positioning by linearly mapping the measured phase directly onto the spatial coordinates.

We numerically evaluate the absolute localization error as a function of spatial frequency, using Fourier basis patterns $(f_x, f_y) = (f, 0)$ and $(0, f)$ for the x and y components respectively.

As shown in Fig. 5(a), we evaluate the absolute localization error as a function of target size and spatial frequency for Fourier basis patterns. Localization at frequencies $f > 1/M$ fails because multiple periods appear within the field of view, making it impossible to identify which period contains the target. We therefore restrict absolute localization to $f \leq 1/M$. Within this range, lower frequencies yield higher accuracy. At any given frequency, larger targets produce larger errors. The bias arises because each non-DC Fourier coefficient carries an image-dependent intrinsic phase. This intrinsic phase decreases as the frequency decreases or as the target becomes smaller, thereby improving the accuracy of localization. In summary, Fourier basis patterns with lower spatial frequencies f generally yield more accurate absolute localization.

Figure 5: *Simulation of localization performance for Fourier location patterns at different spatial frequencies f . The pattern size is 256×256 with no up-sampling, i.e., $M = 256$. The targets comprise 500 handwritten digits randomly selected from the MNIST dataset. For each target size, each target is randomly positioned 100 times, the mean root-mean-square localization error (RMSE) is computed, and the curve of localization error as a function of target size is plotted. (a) Theoretical localization performance of Fourier basis patterns at different spatial frequencies f . (b) Empirical localization performance of Fourier location patterns at different spatial frequencies f under three-step phase shifting combined with Sierra-Lite dithering and a serpentine error-diffusion path.*

However, in DMD-based SPI, three-step phase shifting is required to separate the basis patterns, and spatial dithering is used to binarize grayscale patterns. Extending the analysis in Fig. 5(a), we ran simulations with Sierra-Lite dithering and a serpentine error-diffusion path, and the results are shown in Fig. 5(b). For typical target sizes of $80 \sim 120$ pixels (approximately $1/9 \sim 1/4$ of the field of view), the optimal Fourier localization pattern has a spatial frequency of $1/(2M)$ and yields a localization error of $\sim 1/3$ of a pixel.

Supplementary Note 1 provides a more detailed discussion of the error analysis of the Fourier localization method and the rationale for selecting the spatial frequency $1/(2M)$.

We respectfully note that our localization method operates with a single channel. The multichannel scheme is intended to estimate the target orientation angle from the relative positions of centroids across channels, thereby extending SPI based motion sensing to full two-dimensional rigid body motion. Therefore, the Fourier localization method forms the foundation of MC3-SPI, and we have placed the subsection titled “Absolute coordinate localization of the centroid” as the first subsection of the Results section.

We fully agree that motion compensation can be performed using relative displacement alone. One of the key contributions of MC3-SPI is that it retrieves the target’s rotational information, which requires the absolute coordinates of the target. This motivated us to focus on improving the localization performance of the absolute centroid coordinates. We understand the concern and therefore provide a mathematical proof of the translation property of the Fourier transform. This proof shows that, provided the displacement is smaller than one period of the pattern, any Fourier basis pattern yields an exact estimate of the relative displacement. We hope this clarification addresses the concern.

Specifically, we exploit the Spatial-Shift property of the Fourier transform, whereby a spatial translation corresponds to a phase modulation in the frequency domain. As shown in the following:

Let $\mathcal{F}\{f(x)\}(\xi) = F(\xi) = \int_{-\infty}^{\infty} f(x) e^{-j2\pi\xi x} dx$. For any $x_0 \in \mathbb{R}$,

$$\mathcal{F}\{f(x - x_0)\}(\xi) = \int_{-\infty}^{\infty} f(x - x_0) e^{-j2\pi\xi x} dx = e^{-j2\pi\xi x_0} \int_{-\infty}^{\infty} f(t) e^{-j2\pi\xi t} dt = e^{-j2\pi\xi x_0} F(\xi). \quad (9)$$

The 2D and DFT cases follow analogously, for any $x_0, y_0 \in \mathbb{N}$,

$$\mathcal{F}\{f(x - x_0, y - y_0)\}(u, v) = \sum_{x=0}^{N-1} \sum_{y=0}^{M-1} f(x - x_0, y - y_0) e^{-j2\pi(ux+vy)} \quad (10)$$

For a target smaller than the field of view, its translation can be regarded as a circular shift of the entire array. With the cyclic substitution $x' = x - x_0$, $y' = y - y_0$, the mapping is a bijection on $\{0, \dots, M - 1\} \times \{0, \dots, N - 1\}$, therefore, any discrete sum over this index set remains invariant.

$$\mathcal{F}\{f(x - x_0, y - y_0)\}(u, v) = \sum_{x'=0}^{N-1} \sum_{y'=0}^{M-1} f(x', y') e^{-j2\pi(u(x'+x_0)+v(y'+y_0))} = e^{-j2\pi(ux_0+vy_0)} F(u, v). \quad (11)$$

It is noteworthy that, since the smallest sampling unit in SPI is a single pixel, translational compensation does not need to account for sub-pixel displacements. As a result, with the Fourier basis pattern:

$$P(x, y | f_x, f_y) = \exp[-2\pi j \cdot (f_x x + f_y y)], \quad (12)$$

a displacement $(\Delta x, \Delta y)$ in the spatial domain therefore induces a phase shift $(-2\pi f_x \Delta x, -2\pi f_y \Delta y)$ in the Fourier domain of the pattern. In practice, we employ two sets of patterns, setting $f_x = 0$ and $f_y = 0$, respectively, to determine the relative displacement Δy and Δx of the target.

We appreciate the reviewers' constructive advices.

Q3.5: Supp. Fig. 4: Please explain how the motion was simulated

Reply:

Thank you for your attention and careful reading.

This comment is very helpful for improving the manuscript. In the MATLAB simulation, we use function handles to construct the equations of the motion parameters x , y , and θ with respect to time t . Within the loop, we determine the time of the current iteration from the loop index and the DMD flipping rate, and we place the target image at the appropriate position in the field-of-view matrix. We have added at the beginning of the Supplementary Notes the explicit equations for x , y , and θ as functions of t , which are consistent with the settings used in the MATLAB simulation code.

In the simulation, at virtual time t , the target's absolute position $(x(t), y(t))$ and rotation angle $\theta(t)$ are given by:

$$\begin{aligned} x(t) &= \text{round}((N - K - 2)\cos(2\pi vt + 1.2)/2/(1 + vt) + (M - K)/2), \\ y(t) &= \text{round}((M - K - 2)\cos(2\pi vt)/2/(1 + vt) + (M - K)/2), \\ \theta(t) &= 0.93 * 360 * ((1 + \sin(2\pi\omega t))/2 + (1 + 10\cos(2\pi * 0.513t + 1.02))/11 \\ &\quad + (2 + \cos(2\pi * 0.232t + 0.52))/3)/3, \end{aligned} \quad (13)$$

where, $t = (n - 1)\Delta t$, with n indexing the current frame, corresponding to the n -th SPI pattern. And Δt denotes the minimum temporal resolution of the system. In this simulation, the DMD operates at 20 kHz, which corresponds to a temporal resolution of 1/20 000 s. M and N denote the pixel dimensions of the SPI field of view, with both set to 768. K represents the target pixel size in the SPI perspective and is set to 280. The parameter v characterizes the translational motion of the target and is fixed at 0.618 in this simulation, while ω represents the rotational motion of the target and is fixed at 3.14.

We again thank the reviewers for the constructive input.

Q3.6: Supp. Fig. 5: The figure looks at different levels of noise, but it doesn't explain what kind of noise model is being used (e.g. is it Gaussian, Poisson or Skellam?). The figure also assumes that 40dB SNR is the lowest possible range, but this seems quite high (please explain how the signal powers given in the extra material are defined). What we can see is that the angle estimation is more accurate than the centre-of-mass estimation. This is because the displacement estimate is based on relative positioning, not absolute positioning.

Reply:

We thank the reviewers for raising this point.

The discussion of noise merits further clarification and refinement. In the numerical simulations we add Gaussian white noise using the MATLAB function `awgn()`. We state this explicitly in the relevant part of the Supplementary Notes.

We employ MATLAB's `awgn()` function to inject additive white Gaussian noise into the signal, setting the "signalpower" option to "measured" so that the routine first computes the signal level and then determines the required noise level.

Our definition of SNR follows the standard one. It is defined as the ratio of signal power to noise power and is expressed in decibels (dB).

We use the signal-to-noise ratio (SNR) to represent the noise level, as given by the following equation:

$$\text{SNR}_{\text{dB}} = 10 \log_{10} \left(\frac{P_{\text{signal}}}{P_{\text{noise}}} \right) \text{ or } 20 \log_{10} \left(\frac{A_{\text{signal}}}{A_{\text{noise}}} \right), \quad (14)$$

where, P is the power and A is the root mean square (RMS) amplitude.

As you noted, the level of 40 dB used in the simulations is relatively high. We believe that two factors contribute to this outcome. First, when adding noise, we used the "measured" option of `awgn()`, which automatically estimates the signal power and adds Gaussian white noise at the requested SNR. Because our simulations operate on 768×768 double-precision images, the measured signal power can be relatively high, which may in turn introduce noise with higher power. The "measured" option of `awgn()` may also lead to variations between simulations. Second, we add noise directly to the measurements, so the injected noise represents intrinsic system noise or dark noise, whose impact on localization and reconstruction is more direct and stronger than that of ambient background noise.

We respectfully take a different view and believe that angle estimation is more susceptible to noise than centroid localization. In Supplementary Figure 5a and 5b the dispersion of the angle is larger than that of the centroid coordinates under the same noise level, indicating a higher sensitivity of the angle to noise. In Supplementary Figure 5c at 30 dB the reconstructed image first exhibits motion artifacts along the rotational direction, which further suggests that the angle is more affected by noise. To improve readability in subsequent background noise simulations, we revised the plotting format and combined the RMSE of the angle and the coordinates into a single figure shown as the new panel a, which makes the difference in noise sensitivity more apparent. We again thank the reviewers for the careful comments.

Figure 6: *Numerical simulation results under different Gaussian noise levels (background noise). (a) RMSE of the tracking results under various noise levels. (b) PSNR and SSIM of the Fourier method's image reconstruction results under different noise levels. (c) Image reconstruction results of the Fourier method at signal-to-noise ratios of 10 dB, 0 dB, -10 dB, and -20 dB.*

Q3.7: Supp. Fig. 7 is lacking in thorough analysis: a single translation or rotation is not sufficient to establish a statistical analysis, in particular because the results should differ significantly for integer or sub-pixel pattern shifts. It is imperative that the probability distributions presented incorporate kernel definitions. Furthermore, the colormaps depicted in panels c) and d) are challenging to discern and are not consistent with those observed in panels a) and b). The authors also provide a mathematical explanation of the advantage of the Fourier basis, and this discussion could benefit from referencing recent work [5] that addresses similar ideas (interpolation of sharp Hadamard patterns leads to issues).

Reply:

We thank the reviewers for the careful evaluation.

We apologize for any lack of clarity in the Supplementary Notes. For each pattern, we repeated the orthogonality assessment after motion compensation 100 times to enable proper statistical analysis. We have added the corresponding clarification in the Supplementary Notes.

We sorted the basis patterns according to the SPI sampling strategy and selected the top 5%, 15%, 30%, 50% of the patterns. From each set of basis patterns, 800 patterns are randomly selected. For each selected pattern, random motion compensation is applied and the mean inner product is computed, with each pattern repeated 100 times. For each basis pattern, an arbitrary translation or rotation or translation and rotation is applied, after which its average Frobenius inner product with the remaining patterns is calculated. *This procedure was repeated 100 times per basis pattern.*

We understand the concern that subpixel pattern shifts may lead to significant differences. We respectfully note that, in the SPI framework, a pixel is the minimal unit of spatial resolution, so subpixel compensation is of limited practical value. On the one hand, SPI inherently tolerates subpixel jitter of the target without producing visible motion artifacts. On the other hand, when Fourier-SPI employs spatial dithering to binarize Fourier patterns, the reconstructed image inherently exhibits subpixel offsets. For these reasons, we do not introduce subpixel compensation in practice and instead adopt a nearest integer pixel compensation scheme.

We apologize for any ambiguity in the plotting of Supplementary Figure 7. We have redrawn Supplementary Figure 7 to improve readability. The results are now separated into three cases, namely translation only, rotation only, and translation with rotation, and we present distributions at sampling rates of 5%, 15%, 30%, and 50%. The caption of the figure explicitly states the kernel function used.

Figure 7: Orthogonality analysis of basis patterns *at different sampling rates* after (a) translation or (b) rotation *or* (c) *translation and rotation* is described using the average Frobenius inner product. The greater the amount of data close to zero, the better the orthogonality of the basis patterns after motion compensation. *Basis patterns of size 128×128 are used, with sampling ratios set to 5%, 15%, 30%, and 50%. In each subplot, the up panel shows the statistical results of the data, and the down panel displays the probability density function estimated with a normal kernel function and a bandwidth of 2. Random ± 1 patterns are used as a control to benchmark the orthogonality level.*

In the discussion of the morphological degradation of Hadamard patterns under rotation, we cite [5] to

better motivate and support our perspective.

This results in sharply defined Hadamard patterns becoming blurred at the interfaces between +1 and -1 after rotation, leading to a loss of structural features [5].

We again thank the reviewers for these helpful comments.

Q3.8: Supp. Fig. 9: The caption doesn't explain the differences between the first two rows clearly.

Reply:

We thank the reviewers for the attention to the legend details and for the helpful suggestions, and we apologize for the oversight.

We have corrected the insufficient description in the first two rows of Supplementary Figure 9.

Sampling rate	0.01	0.02	0.03	0.04	0.05	0.07	0.1	0.2	0.3	0.4	0.5
BIT Translation and rotation											
BIT random											
Mario											
Rocket											

Figure 8: IMCFT reconstruction results under different sampling rates.

We thank the reviewers for the helpful suggestion again.

Q3.9: Supp. Fig. 10 presents quantitative results under low sampling rates. Since no ground truth is available from experimental data, the authors use TVAL3 with 50% measurements as a reference. However, a simulated experiment with known ground truth could deliver a more compelling and interpretable evaluation.

Reply:

We thank the reviewers for the careful reading of the Supplementary Notes and for the constructive suggestions.

Following this advice, we added numerical simulations with known ground truth and present intuitive reconstruction results together with the corresponding SSIM and PSNR curves. To accurately assess the performance of the proposed IMCFT method, we include static IFT reconstructions as a reference. We present reconstructed images at the two key sampling rates of 5% and 10% across different methods to provide a more intuitive assessment of IMCFT performance. To specifically analyze how well IMCFT preserves the target structure, we introduce and apply a lightweight linear contrast stretch to suppress low-intensity clutter in the image background and highlight structural details. Considering the visual results after the linear contrast stretch together with the SSIM metrics, we believe that IMCFT effectively captures target structure and supports target recognition at low sampling rates.

To more precisely quantify the reconstruction efficiency of IMCFT, particularly in comparison with static Fourier-SPI with IFT reconstruction, we conducted numerical simulations based on the

framework described in Supplementary Note 5. We present IMCFT reconstructions at multiple sampling ratios and plot detailed PSNR and SSIM curves. For visual comparison, we also show static IFT reconstructions at the same sampling ratios together with their PSNR and SSIM curves. The results are shown in Supplementary Fig.9.

Figure 9: Numerical simulations of IMCFT reconstruction efficiency include reconstructed images at different sampling rates, as well as PSNR and SSIM values. The PSNR and SSIM are benchmarked against the indicators of static Fourier-SPI IFT reconstruction results. The linear contrast stretch (LCS) is applied to eliminate the influence of low-intensity background noise without altering the structure, allowing for more accurate quantification of the target structure’s reconstruction.

Compared with the previous experimental results, when using true original image as references in the numerical simulations, the PSNR and SSIM of the IMCFT reconstructions are lower. This outcome is expected because the basis patterns are deliberately rotated, which degrades the IMCFT reconstruction performance. Specifically, when the sampling rate exceeds 10%, the PSNR improvement of IMCFT is inferior to that of static IFT, and the SSIM of IMCFT almost stops increasing.

However, when visually comparing the reconstructions of IMCFT and static IFT, we do not observe noticeable differences. In a visual comparison between IMCFT and static IFT at a sampling rate 5%, both reconstructions display the characteristic ringing artifacts of Fourier-SPI at low sampling rates. Aside from the relatively more pronounced cluttered background in IMCFT, the reconstruction quality of the target itself is nearly identical. At a sampling ratio 10%, the reconstructed target images from IMCFT and static IFT are nearly indistinguishable when the background is disregarded. Even at a sampling ratio 50%, the two reconstructions differ only marginally and their structural content is almost identical.

We apply a simple linear contrast stretch to smooth high-intensity regions and suppress low-intensity

clutter in the background, thereby highlighting the structure of the reconstructed target. Visual inspection of IMCFT reconstructions before and after processing indicates that the linear contrast stretch does not alter the structural content of the target image, yet it significantly improves PSNR and SSIM, even outperforming static IFT. Therefore, we believe that IMCFT is effective in capturing the target structure and thereby enabling target recognition. Moreover, linear contrast stretch is computationally lightweight and requires only ~ 0.0002 seconds per application. The linear contrast stretch algorithm is detailed in Algorithm 3.

Algorithm 3 *Linear contrast stretch*

Require: uint8 RGB matrix `image`, stretch factor $\alpha = 1.2$

Ensure: Matrix `image` stretched to $[0, 255]$

- 1: `image` $\leftarrow (\text{image} - 128) \times \alpha + 128$
 - 2: Set values below 0 to 0: `image`(`image` < 0) = 0
 - 3: Set values above 255 to 255: `image`(`image` > 255) = 255
 - 4: Convert to unsigned 8-bit integers: `image` = `uint8`(`image`)
 - 5: **return** `image`
-

SSIM places greater emphasis on comparing structural information between reference and reconstructed images, making it more aligned with perceived visual quality. Based on the SSIM results and the direct visual perception of the reconstructed images, at approximately 5% sampling, the IMCFT method can reconstruct images sufficient for reliable target recognition, enabling MC3-SPI to rapidly identify high-speed targets.

We again thank the reviewers for the valuable suggestions, which have improved the manuscript.

Reproducibility:

Q3.10: Aside from the issues noted earlier, the methodology is described insufficient detail to follow the general approach. The authors kindly state that reconstruction code is available upon reasonable request. However, it is not entirely clear whether the code or implementation for motion estimation is included. If we were to consider providing access to this part of the pipeline, it could potentially lead to a significant improvement in reproducibility. Additionally, it may be beneficial to consider providing access to the experimental data, or alternatively using publicly available datasets (such as those from [9, 11]), as this could significantly strengthen the reproducibility and accessibility of the work.

Reply:

We thank the reviewer for the helpful suggestion.

We have made the MC3-SPI reconstruction code publicly available on GitHub. It includes a complete implementation of motion sensing and image reconstruction. The raw data corresponding to the main text are provided in the respective folders.

The code for the reconstruction algorithms used in this study and the raw data are publicly available at <https://github.com/ArthurWu1999/MC3-SPI>.

4.# Appropriate use of statistics and treatment of uncertainties: All error bars should be defined in the corresponding figure legends; please comment if that's not the case. Please include in your report a specific comment on the appropriateness of any statistical tests, and the accuracy of the description of any error bars and probability values.

Q4.1 See the comment made in the previous section on supp. Fig. 7.

5.# Conclusions: Do you find that the conclusions and data interpretation are robust, valid and reliable?

Q5.1: The conclusions in the main manuscript feel relatively limited in scope. While the paper effectively demonstrates that the proposed method yields visually compelling results and that Fourier patterns appear more suitable than Hadamard patterns for motion-compensated imaging, several crucial insights are relegated to the supplementary material. Specifically, the analysis of localization error with Fourier patterns and the orthogonality of motion-compensated patterns are vital. Integrating and discussing these points within the main document would significantly strengthen the overall narrative and impact of the research.

Reply:

We thank the reviewers for the positive assessment.

We agree that the analysis of localization error for Fourier patterns and the discussion of the orthogonality of motion-compensated patterns are essential. We have streamlined these materials and integrated them into the first two subsections at the beginning of the Results section as the theoretical foundation of the manuscript so that readers can access these key ideas more directly.

We distilled from Supplementary Note 1 the core analyses, namely the theoretical analysis of the Fourier localization method and the practical analysis based on DMD-SPI. The former guides the direction of method optimization, whereas the latter determines the final Fourier spatial frequency under practical constraints. For more detailed application-level optimizations such as spatial dithering, we present the results directly and retain the step-by-step procedures in Supplementary Note 1.

Absolute coordinate localization of the centroid

Each Fourier coefficient aggregates all pixels, making local spatial changes global in the spectrum. For a single rigid target, motion can be detected by tracking a small set of coefficients over time. Fourier-SPI recovers these coefficients from only a few measurements by using Fourier patterns and a single-pixel detector.

By the Spatial-Shift property of the Fourier transform, a translation $(\Delta x, \Delta y)$ induces phase shifts of $-2\pi f_x \Delta x$ and $-2\pi f_y \Delta y$ in the coefficients at $(f_x, 0)$ and $(0, f_y)$, respectively. This property yields only relative displacement. We extend this to absolute positioning by linearly mapping the measured phase directly onto the spatial coordinates.

We numerically evaluate the absolute localization error as a function of spatial frequency, using Fourier basis patterns $(f_x, f_y) = (f, 0)$ and $(0, f)$ for the x and y components respectively.

Figure 10: Simulation of localization performance for Fourier location patterns at different spatial frequencies f . The pattern size is 256×256 with no up-sampling, i.e., $M = 256$. The targets comprise 500 handwritten digits randomly selected from the MNIST dataset. For each target size, each target is randomly positioned 100 times, the mean localization RMSE is computed, and the curve of localization error as a function of target size is plotted. (a) Theoretical localization performance of Fourier basis patterns at different spatial frequencies f . (b) Empirical localization performance of Fourier location patterns at different spatial frequencies f under three-step phase shifting combined with Sierra-Lite dithering and a serpentine error-diffusion path.

As shown in Fig. 10(a), we evaluate absolute localization error versus target size and spatial frequency for Fourier basis patterns. Localization with frequencies $f > 1/M$ fails because multiple periods appear across the field of view, making the target’s period index ambiguous. We therefore restrict absolute localization to $f \leq 1/M$. Within this range, lower frequencies yield higher accuracy. For any frequency, larger targets produce larger errors. The bias arises because each non-DC Fourier coefficient carries an image-dependent intrinsic phase. This intrinsic phase decreases as the frequency decreases or as the target becomes smaller, which improves localization accuracy. In summary, Fourier basis patterns with lower spatial frequencies f generally yield more accurate absolute localization.

However, it is necessary to employ the three-step phase-shifting method to separate the basis patterns and spatial dithering to binarize grayscale patterns in DMD-based SPI. Building on Fig. 10(a), we ran simulations with Sierra-Lite dithering and a serpentine error-diffusion path, and the results are shown in Fig. 10(b). For typical target sizes of $80 \sim 120$ pixels (approximately $1/9 \sim 1/4$ of the field of view), the optimal Fourier localization pattern has a spatial frequency of $1/(2M)$ and yields a localization error of $\sim 1/3$ of a pixel.

Supplementary Note 1 provides a more detailed discussion of the error analysis of the Fourier localization method and the rationale for selecting the spatial frequency $1/(2M)$.

We apologize for the oversight in the orthogonality analysis. Our previous use of the plain Frobenius inner product was not sufficiently rigorous. We now adopt the averaged normalized Frobenius inner product to remove the influence of matrix norms and to more accurately quantify the loss of orthogonality induced by motion compensation.

To quantitatively assess this orthogonality, for a set of orthogonal patterns $\{P_1, P_2, \dots, P_n\}$, we employ the normalized Frobenius inner product (matrix inner product) between the compensated basis pattern P'_i and the remaining basis patterns P_j ($j \neq i$), as shown in the following equation:

$$C_i = \frac{MN}{n-1} \sum_{j \neq i} \frac{|\langle P'_i, P_j \rangle_F|}{\|P'_i\|_F \|P_j\|_F}, \quad 0 \leq C_i \leq MN. \quad (15)$$

where, Frobenius inner product $\langle A, B \rangle_F = \text{tr}(A^H B)$, Frobenius norm $\|A\|_F = \sqrt{\langle A, A \rangle_F}$. n denotes the number of complete orthogonal bases, (M, N) denotes the size of the bases. A smaller C_i means the patterns are closer to orthogonal.

We repeated the numerical simulations and added the corresponding content to the second subsection

of the Results section. The simulations cover sampling rates from 5% to 50% with a step of 1%, apply the median with interquartile band, and the sample size was increased to enable a more detailed and comprehensive analysis. We plot the results as curves to present the overall trend in an intuitive manner. More detailed data distributions are provided in the Supplementary Note 7 for readers to examine the statistical results further.

Orthogonality analysis of motion-compensated patterns

When imaging moving targets within an SPI architecture, motion compensation is commonly implemented by applying translations or rotations to the patterns in opposition to the target motion. Such transformations often alter the intrinsic properties of the patterns, thereby affecting their performance during sampling and reconstruction. Given that Fourier basis patterns and Hadamard basis patterns enable efficient sampling and reconstruction in SPI owing to their orthogonality, we analyze the orthogonality of these bases under motion compensation.

In compressed sensing, the maximum normalized Frobenius inner product of a set of patterns is used to evaluate their cross correlation and thereby ensure successful image reconstruction. In this work, we aim only to quantify the loss of orthogonality of motion-compensated patterns and therefore adopt the mean normalized Frobenius inner product. A lower value also indicates weaker correlation and thus a smaller loss of orthogonality induced by motion compensation.

We conducted numerical simulations to evaluate the orthogonality of Fourier basis patterns and Hadamard basis patterns under motion compensation. For each sampling rate, we form a new subset by randomly selecting 800 basis patterns from the set used at that sampling rate. Each selected pattern is subjected to a random rotation or a random translation or a random rotation and translation, and its mean normalized Frobenius inner product with all remaining patterns in the subset (Eq.12 of Supplementary Note 7) is computed. This procedure is repeated 100 times for every pattern, and the resulting data were statistically analyzed. Since the data are non-Gaussian, we compute the median together with the $p_{25} \sim p_{75}$ interval, the interquartile range (IQR), as a robust uncertainty band. This interval covers the central 50% of the observations. In addition, we included random ± 1 patterns (without motion-compensation) as a control, and the results are shown in Fig. 11.

Figure 11: Orthogonality analysis of basis patterns after motion-compensation at sampling rates between 5% and 50%. The orthogonality of the basis patterns is quantified by the mean Frobenius inner product. Statistical analysis employs the median together with the $p_{25} \sim p_{75}$ interval (IQR). Values concentrated nearer to zero indicate stronger orthogonality. The random ± 1 patterns (without motion-compensation) serve as a reference. (a) The orthogonality analysis of Fourier and Hadamard basis patterns under random translation compensation. (b) The orthogonality analysis of Fourier and Hadamard basis patterns under random rotation compensation. (c) The orthogonality analysis of Fourier and Hadamard basis patterns under random translation and rotation compensation.

As shown in Fig. 11, after motion-compensation, the orthogonality of the Fourier basis patterns is markedly superior to that of Hadamard basis patterns across the sampling rates of 5% \sim 50%. As shown in Fig. 11(a), Fourier basis patterns are scarcely affected by translational compensation. As shown in Fig. 11(b), Fourier basis patterns retain substantial orthogonality after rotation compensation. As shown in Fig. 11(c), after joint translation-rotation compensation, the loss of orthogonality of Fourier basis patterns throughout the full sampling range remains at a very low

level, which makes them well suited for motion-compensation imaging.

In contrast, the orthogonality of Hadamard basis patterns degrades substantially after motion compensation. In particular, after rotational compensation, their orthogonality at low sampling ratios ($\leq 10\%$) is even inferior to that of random ± 1 patterns, as illustrated in Fig. 11(b). In addition, in Fig. 11(c), as the sampling ratio increases, the orthogonality of the Hadamard basis patterns improves, indicating that reconstruction requires a relatively high sampling ratio ($\geq 30\%$).

We selected data at four sampling rate points for a comprehensive statistical analysis and provided a more detailed discussion in Supplementary Note 7. Most critically, we demonstrate that the complete set of Fourier basis constitutes a linear space closed under translations and rotations. Therefore, we believe that the Fourier basis is the optimal SPI sampling mode for motion-compensated imaging of two-dimensional composite motion targets.

We again thank the reviewers for the thoughtful and constructive comments.

Q5.2: The manuscript repeatedly suggests that the method operates without assumptions on scene motion. This claim is misleading. The approach explicitly relies on a rigid planar motion model, which is a significant underlying assumption. This critical dependency should be clearly acknowledged and thoroughly discussed throughout the manuscript to provide an accurate representation of the method’s applicability and limitations.

Reply:

We thank the reviewers for the careful reading and constructive suggestions.

We agree with this point and have added an explicit rigid-body constraint on the target in the title of the article to clarify the scope of applicability of our method.

Multichannel multicentroid motion-compensated single-pixel imaging of *an 2D* arbitrarily moving *rigid-body* target

We apologize for any ambiguity caused by our oversight. In the Results subsection “Motion capture and image reconstruction” we introduce the motion acquisition method of MC3-SPI, and the opening sentence explicitly clarifies the scope of applicability of our method.

The complete motion state of a rigid-body target moving in two-dimensional space can be determined by considering multiple points attached to the target.

We are sorry for any inconvenience this may have caused and we appreciate your patience and guidance.

Q5.3: The authors state they were able to "perform motion-compensated imaging to directly reconstruct a complete image of the target" (line 263). This statement is misleading. While the method successfully estimates motion that extends beyond the conventional field of view, it does not actually reconstruct parts of the scene that lie outside the system’s observable area. The authors should rephrase this claim to accurately reflect the true scope of their contribution and avoid overstating the capabilities of their system.

Reply:

We are deeply grateful to the reviewers for the careful assessment and thoughtful recommendations that guided our revisions. We have updated the manuscript accordingly and believe these changes enhance methodological rigor.

As noted correctly, our method successfully estimates motion that extends beyond the conventional field of view. Although the target never fully enters the field, every part of the target appears within the field at some time, so we do not reconstruct regions outside the system’s observable area. To avoid misunderstanding and to delineate the scope of the method, we have removed the wording “complete” image from the description and clarified the capability boundaries accordingly.

By following this approach, we obtained *the effective part of the target motion state* and performed motion-compensated imaging to directly reconstruct *a image of the target*.

In addition, we have revised the subsection title to explicitly specify the imaging subject as “a boundary-motion target”, which clarifies and delineates the scope of applicability of our method.

extend-FOV tracking and imaging of a boundary-motion target

We appreciate this helpful comment.

Q5.4: One of the manuscript’s stated conclusions is that the method remains robust under noise conditions of 40 dB SNR. However, as highlighted in the discussion of Supplementary Figure 5, this may represent a relatively high SNR threshold, which raises concerns about the method’s applicability in lower-SNR, real- world scenarios. This suggests the method might be less robust than initially implied, particularly in more challenging practical conditions. That said, it is important to note that the method performed well on real experimental data, indicating that its practical utility may extend beyond what is suggested by the simulations alone. A more nuanced discussion of this apparent discrepancy between simulation results and real-data performance would significantly help clarify the method’s actual robustness and its limitations in varying noise environments.

Reply:

We are deeply grateful to the reviewers for the careful reading and thoughtful recommendations that guided our revisions.

We agree that 40 dB is a relatively high SNR threshold, which implies that the signal power is about ten thousand times the noise power. This understandably raises questions about effectiveness under real-world conditions. We therefore clarify the noise analysis in Supplementary Note 6. In the numerical simulations we add Gaussian white noise directly to the SPI measurements, which in the noise model corresponds to intrinsic system noise or dark noise. Compared with typical ambient background noise, this type of noise affects localization and imaging more directly and more strongly. We therefore respectfully consider it reasonable that the method remains robust at an SNR of 40 dB.

Figure 12: Numerical simulation results under different *Gaussian noise levels (intrinsic noise / dark noise)*. (a) Tracking results of the target’s composite motion at signal-to-noise ratios of 50 dB, 40 dB, 30 dB, and 20 dB. (b) RMSE of the tracking results under various noise levels. (c) Image reconstruction results of the Fourier method at signal-to-noise ratios of 50 dB, 40 dB, 30 dB, and 20 dB. (d) PSNR and SSIM of the Fourier method’s image reconstruction results under different noise levels.

To further substantiate the reasonableness of the robustness level, we examined the specifications of the silicon-amplified photodetectors used in our setup (PDA100A2, Thorlabs). Under our experimental settings, the intrinsic SNR is approximately 66 dB. The system also permits 90 readouts for a single pattern, which provides an additional gain of about 19 dB through averaging. In total, the intrinsic SNR of the system is about 85 dB, which is well above the 40 dB threshold adopted in our simulations.

In particular, the silicon-amplified photodetectors (PDA100A2, Thorlabs) used in our setup deliver a RMS dark output of $229\mu\text{V}$ at the gain mode of 40dB setting, while the signal level measured during the experiments is about 0.5V. Therefore, the SNR of the photodetector for a single readout is approximately 66.78dB ($\Delta\text{SNR}_{\text{dB}} = 20\log_{10}\frac{RO}{DCR}$, $RO = 0.5\text{V}$ $DCR = 229 \times 10^{-6}\text{V}$).

Beside, the data acquisition card (USB-6366, National Instruments) can record 2×10^6 samples per second. The DMD (DLP7000, Texas Instruments) has a maximum flipping rate of 22 kHz, which implies that each pattern can be sampled at least 90 times. Because the signal component is coherent whereas the noise is not, repeated measurements cause the signal to add linearly as N while the noise grows only as \sqrt{N} . The signal accumulates \sqrt{N} times faster than the noise. Ideally, this would provide an improvement in the SNR of approximately 19.50dB ($\Delta\text{SNR}_{\text{dB}} = 20\log_{10}\sqrt{N}$, $N = 90$).

After all, the resulting SNR for a single pattern acquisition is approximately 86.30dB. This SNR level enables the reconstruction of high-quality images of targets undergoing compound motion.

Finally, to further corroborate the experimental findings, we conducted an additional robustness assessment in which ambient noise was simulated on top of the intrinsic noise level of the system. The results indicate that our method remains robust when the ambient SNR is 0 dB. This condition only requires the signal power to match the ambient noise power one to one, whereas the SNR available in our laboratory is well above this level. Taken together, these analyses support the reasonableness of the experimental results and provide a relatively comprehensive discussion of the robustness of our approach.

In optical imaging systems, another common noise source arises from ambient background illumination. It is worth noting that SPI senses the image holistically, with each measurement records the sum of the signals from numerous pixels across the field of view. In Fourier-SPI, a single readout corresponds to approximately half of the total optical power of the scene. Due to this integrative acquisition principle, SPI is inherently highly robust to background noise. To quantify this robustness, we analyze the system response to background noise on top of its intrinsic noise floor of 86.30 dB. The result is shown in Supplementary Fig. 13.

Figure 13: *Numerical simulation results under different Gaussian noise levels (background noise). (a) RMSE of the tracking results under various noise levels. (b) PSNR and SSIM of the Fourier method's image reconstruction results under different noise levels. (c) Image reconstruction results of the Fourier method at signal-to-noise ratios of 10 dB, 0 dB, -10 dB, and -20 dB.*

As shown in Supplementary Figs. 13 (a) and (b), the SPI exhibits strong robustness to background noise. Even at a background noise level of -10 dB, it can still reconstruct an approximate image of the target. In Supplementary Figs. 13 (c), a clear image can be reconstructed when the background noise level is 0 dB. Consequently, the proposed method with our system can faithfully reconstruct the target image even at ambient noise levels of ~ 0 dB.

We sincerely appreciate the reviewers' helpful guidance, which has been invaluable in refining the manuscript.

6.# Suggested improvements: Please list additional experiments or data that could help strengthening the work in a revision.

Q6.1: The main document exhibits some redundancy, and unfortunately, much of the deeper technical insight is relegated to the supplementary material. A reorganization that integrates key analyses, such as the localization error and the orthogonality of basis functions, into the main text would significantly enhance clarity and strengthen the overall message.

Reply:

We thank the reviewers for the insightful and constructive comments.

In the main text, we added a clearer presentation of the localization method and an expanded discussion of the orthogonality of the basis patterns. To convey several deeper insights more directly to readers,

we streamlined some parts relative to the Supplementary Notes and expanded key analyses where appropriate.

For the localization error of the Fourier method, we extend the idealized analysis by evaluating the localization performance of Fourier basis patterns with spatial frequencies greater than $1/M$, thereby showing the absolute position error across spatial frequencies in an intuitive manner. In the application-level model, we directly simulate binarization via spatial dithering using Sierra-Lite with a serpentine error diffusion path, report the practically achievable localization accuracy, and explain why we choose Fourier patterns with spatial frequency $1/(2M)$ as the localization mode.

Absolute coordinate localization of the centroid

Each Fourier coefficient aggregates all pixels, making local spatial changes global in the spectrum. For a single rigid target, motion can be detected by tracking a small set of coefficients over time. Fourier-SPI recovers these coefficients from only a few measurements by using Fourier patterns and a single-pixel detector.

By the Spatial-Shift property of the Fourier transform, a translation $(\Delta x, \Delta y)$ induces phase shifts of $-2\pi f_x \Delta x$ and $-2\pi f_y \Delta y$ in the coefficients at $(f_x, 0)$ and $(0, f_y)$, respectively. This property yields only relative displacement. We extend this to absolute positioning by linearly mapping the measured phase directly onto the spatial coordinates.

We numerically evaluate the absolute localization error as a function of spatial frequency, using Fourier basis patterns $(f_x, f_y) = (f, 0)$ and $(0, f)$ for the x and y components respectively.

Figure 14: *Simulation of localization performance for Fourier location patterns at different spatial frequencies f . The pattern size is 256×256 with no up-sampling, i.e., $M = 256$. The targets comprise 500 handwritten digits randomly selected from the MNIST dataset. For each target size, each target is randomly positioned 100 times, the mean localization RMSE is computed, and the curve of localization error as a function of target size is plotted. (a) Theoretical localization performance of Fourier basis patterns at different spatial frequencies f . (b) Empirical localization performance of Fourier location patterns at different spatial frequencies f under three-step phase shifting combined with Sierra-Lite dithering and a serpentine error-diffusion path.*

As shown in Fig. 14(a), we evaluate absolute localization error versus target size and spatial frequency for Fourier basis patterns. Localization with frequencies $f > 1/M$ fails because multiple periods appear across the field of view, making the target’s period index ambiguous. We therefore restrict absolute localization to $f \leq 1/M$. Within this range, lower frequencies yield higher accuracy. For any frequency, larger targets produce larger errors. The bias arises because each non-DC Fourier coefficient carries an image-dependent intrinsic phase. This intrinsic phase decreases as the frequency decreases or as the target becomes smaller, which improves localization accuracy. In summary, Fourier basis patterns with lower spatial frequencies f generally yield more accurate absolute localization.

However, it is necessary to employ the three-step phase-shifting method to separate the basis patterns and spatial dithering to binarize grayscale patterns in DMD-based SPI. Building on Fig. 14(a), we ran simulations with Sierra-Lite dithering and a serpentine error-diffusion path, and the results are shown in Fig. 14(b). For typical target sizes of $80 \sim 120$ pixels (approximately $1/9 \sim 1/4$ of the field of view), the optimal Fourier localization pattern has a spatial frequency of $1/(2M)$ and yields a

localization error of $\sim 1/3$ of a pixel.

Supplementary Note 1 provides a more detailed discussion of the error analysis of the Fourier localization method and the rationale for selecting the spatial frequency $1/(2M)$.

For the orthogonality analysis, we first correct a methodological issue in our earlier assessment. We previously sampled the inner product rather than the normalized inner product, the former being affected by pattern norms while the latter directly measures the angle between two patterns. Fortunately, all motion-compensated patterns are derived from a complete orthogonal basis, so norm variations are small and do not affect the conclusion that Fourier patterns outperform Hadamard patterns for motion-compensated imaging. For readability, we plot the median with interquartile band over sampling rates from 5% to 50% and discuss three cases separately, namely translation only, rotation only, and rotation combined with translation, which makes the results easier to read. Moreover, in the Supplementary Note 7 we provide the detailed distributions at sampling rates of 5%, 15%, 30%, and 50% and we improve the figure readability.

Orthogonality analysis of motion-compensated patterns

When imaging moving targets within an SPI architecture, motion compensation is commonly implemented by applying translations or rotations to the patterns in opposition to the target motion. Such transformations often alter the intrinsic properties of the patterns, thereby affecting their performance during sampling and reconstruction. Given that Fourier basis patterns and Hadamard basis patterns enable efficient sampling and reconstruction in SPI owing to their orthogonality, we analyze the orthogonality of these bases under motion compensation.

In compressed sensing, the maximum normalized Frobenius inner product of a set of patterns is used to evaluate their cross correlation and thereby ensure successful image reconstruction. In this work, we aim only to quantify the loss of orthogonality of motion-compensated patterns and therefore adopt the mean normalized Frobenius inner product. A lower value also indicates weaker correlation and thus a smaller loss of orthogonality induced by motion compensation.

We conducted numerical simulations to evaluate the orthogonality of Fourier basis patterns and Hadamard basis patterns under motion compensation. For each sampling rate, we form a new subset by randomly selecting 800 basis patterns from the set used at that sampling rate. Each selected pattern is subjected to a random rotation or a random translation or a random rotation and translation, and its mean normalized Frobenius inner product with all remaining patterns in the subset (Eq.12 of Supplementary Note 7) is computed. This procedure is repeated 100 times for every pattern, and the resulting data were statistically analyzed. Since the data are non-Gaussian, we compute the median together with the $p_{25} \sim p_{75}$ interval, the interquartile range (IQR), as a robust uncertainty band. This interval covers the central 50% of the observations. In addition, we included random ± 1 patterns (without motion-compensation) as a control, and the results are shown in Fig. 15.

As shown in Fig. 15, after motion-compensation, the orthogonality of the Fourier basis patterns is markedly superior to that of Hadamard basis patterns across the sampling rates of 5% \sim 50%. As shown in Fig. 15(a), Fourier basis patterns are scarcely affected by translational compensation. As shown in Fig. 15(b), Fourier basis patterns retain substantial orthogonality after rotation compensation. As shown in Fig. 15(c), after joint translation–rotation compensation, the loss of orthogonality of Fourier basis patterns throughout the full sampling range remains at a very low level, which makes them well suited for motion-compensation imaging.

In contrast, the orthogonality of Hadamard basis patterns degrades substantially after motion compensation. In particular, after rotational compensation, their orthogonality at low sampling ratios ($\leq 10\%$) is even inferior to that of random ± 1 patterns, as illustrated in Fig. 15(b). In addition, in Fig. 15(c), as the sampling ratio increases, the orthogonality of the Hadamard basis patterns improves, indicating that reconstruction requires a relatively high sampling ratio ($\geq 30\%$).

We selected data at four sampling rate points for a comprehensive statistical analysis and provided a more detailed discussion in Supplementary Note 7. Most critically, we demonstrate that the complete set of Fourier basis constitutes a linear space closed under translations and rotations. Therefore, we believe that the Fourier basis is the optimal SPI sampling mode for motion-compensated imaging of

Figure 15: *Orthogonality analysis of basis patterns after motion-compensation at sampling rates between 5% and 50%. The orthogonality of the basis patterns is quantified by the mean Frobenius inner product. Statistical analysis employs the median together with the $p_{25} \sim p_{75}$ interval (IQR). Values concentrated nearer to zero indicate stronger orthogonality. The random ± 1 patterns (without motion-compensation) serve as a reference. (a) The orthogonality analysis of Fourier and Hadamard basis patterns under random translation compensation. (b) The orthogonality analysis of Fourier and Hadamard basis patterns under random rotation compensation. (c) The orthogonality analysis of Fourier and Hadamard basis patterns under random translation and rotation compensation.*

two-dimensional composite motion targets.

We appreciate the reviewers' helpful guidance.

Q6.2: To further solidify the contributions, consider including the following experiments:

Q6.2.1: Evaluate the relative displacement between frames (rather than absolute position) in Supplementary Figure 1. This would more accurately reflect the method's direct relevance to motion compensation.

Reply:

We sincerely thank the reviewers for the insightful comments.

We fully agree that relative displacement can directly guide motion compensation. In practice, if the target motion with respect to the initial time is known at any instant, including both translation and rotation, one can reconstruct a sharp image at the initial position through motion compensation. At the same time, an important contribution of our work is that we successfully sense rotational motion. In our approach, estimating rotation relies on the relative positions of centroids across channels and therefore requires absolute coordinate localization. Due to the aforementioned requirements, we performed the simulations reported in Supplementary Figure 1 to evaluate absolute coordinate localization accuracy of Fourier basis patterns across different spatial frequencies, spatial dithering schemes, and dithering paths, and we proposed a DMD-based optimal configuration of Fourier localization patterns to support the feasibility of the method.

In addition, to fully address the concern, we provide a detailed proof of the spatial shift property of the Fourier transform in Supplementary Note 1. This property shows that target translation maps linearly to the phase of the Fourier coefficients, which implies that Fourier basis patterns at any spatial frequency can in principle recover the relative displacement exactly provided that the displacement does not exceed one period of the pattern.

Specifically, we exploit the Spatial-Shift property of the Fourier transform, whereby a spatial translation corresponds to a phase modulation in the frequency domain. As shown in the following:

Let $\mathcal{F}\{f(x)\}(\xi) = F(\xi) = \int_{-\infty}^{\infty} f(x) e^{-j2\pi\xi x} dx$. For any $x_0 \in \mathbb{R}$,

$$\mathcal{F}\{f(x - x_0)\}(\xi) = \int_{-\infty}^{\infty} f(x - x_0) e^{-j2\pi\xi x} dx = e^{-j2\pi\xi x_0} \int_{-\infty}^{\infty} f(t) e^{-j2\pi\xi t} dt = e^{-j2\pi\xi x_0} F(\xi). \quad (16)$$

The 2D and DFT cases follow analogously, for any $x_0, y_0 \in \mathbb{N}$,

$$\mathcal{F}\{f(x - x_0, y - y_0)\}(u, v) = \sum_{x=0}^{N-1} \sum_{y=0}^{M-1} f(x - x_0, y - y_0) e^{-j2\pi(ux+vy)} \quad (17)$$

For a target smaller than the field of view, its translation can be regarded as a circular shift of the entire array. With the cyclic substitution $x' = x - x_0$, $y' = y - y_0$, the mapping is a bijection on $\{0, \dots, M - 1\} \times \{0, \dots, N - 1\}$, therefore, any discrete sum over this index set remains invariant.

$$\mathcal{F}\{f(x - x_0, y - y_0)\}(u, v) = \sum_{x'=0}^{N-1} \sum_{y'=0}^{M-1} f(x', y') e^{-j2\pi(u(x'+x_0)+v(y'+y_0))} = e^{-j2\pi(ux_0+vy_0)} F(u, v). \quad (18)$$

It is noteworthy that, since the smallest sampling unit in SPI is a single pixel, translational compensation does not need to account for sub-pixel displacements. As a result, with the Fourier basis pattern:

$$P(x, y | f_x, f_y) = \exp[-2\pi j \cdot (f_x x + f_y y)], \quad (19)$$

a displacement $(\Delta x, \Delta y)$ in the spatial domain therefore induces a phase shift $(-2\pi f_x \Delta x, -2\pi f_y \Delta y)$ in the Fourier domain of the pattern. In practice, we employ two sets of patterns, setting $f_x = 0$ and $f_y = 0$, respectively, to determine the relative displacement Δy and Δx of the target.

We would like to thank the reviewers for their helpful guidance, and we hope the revisions address the concerns raised.

Q6.2.2: Provide an example of reconstruction beyond the field of view. This would powerfully demonstrate the practical benefit of estimating motion even when it occurs outside the sensor’s direct range.

Reply:

We thank the reviewers for the helpful suggestion.

Following this advice, we added a numerical simulation as Supplementary Note 9. To present more intuitively that the practical benefit of our system can sense motion occurring outside the field of view, we introduced two constraints in the simulation. First, we restricted the actually used patterns to a size of 86×50 pixels, while the reconstructed image has a size of 128×128 pixels. Second, we set the object size to be larger than the pattern size. Using the motion estimation method described in the main text, we successfully estimated the motion of a target larger than the field of view and reconstructed the target image, which provides a more direct validation of the effectiveness and feasibility of our approach.

Supplementary Note 9: Extend-pixel imaging of a motion target

Our previous experiments successfully captured the motion state of a target moving at the edge of the field of view and reconstructed its complete image, thereby extending the effective optical field of view of the system to some extent. Building on this, we can realize extend-pixel SPI imaging, where the number of pixels in the reconstructed image exceeds that of the SPI patterns, enabling complete reconstruction of a target whose size exceeds the system’s field of view. To validate our claim, we conducted numerical simulations and the results are shown in the Supplementary Fig.16.

Figure 16: *Pattern design for extend-pixel imaging together with motion sensing and motion-compensated imaging results for a target larger than the field of view. In detail, a 128×128 image was reconstructed using 86×50 pixel patterns, with the target measuring approximately 80×80 pixels. (a) Original patterns and real DMD-patterns. (b) Preliminary determination of whether a target channel lies within the field of view based on its intensity. (c) The directly measured trajectory and the trajectory after correction and completion, together with the corresponding target motion information, namely the x and y coordinates and the orientation angle θ . (d) Direct reconstructions, IMCFT extend-pixel reconstructions, and results optimized using TVAL3.*

A key to extend-pixel imaging is how to expand the patterns. We adopt a top-down strategy that crops small patterns directly from larger ones, as shown in Supplementary Fig.16(a). Small patterns are used during SPI acquisition, whereas the original-size patterns are used for image reconstruction. Because Fourier patterns permit arbitrary pixel dimensions, an appropriate pixel size can be selected to accommodate targets of different sizes and reduce additional SPI measurements, making Fourier patterns a suitable choice.

Since the target is larger than the field of view, the complete target cannot be observed and it moves persistently along the field boundary, which matches the motion-sensing approach described in the subsection Extended-FOV tracking and imaging of a cross-boundary motion target in the main text. We conducted numerical simulations of extend-pixel imaging, reconstructing a 128×128 image using 86×50 -pixel patterns, and the results are presented in Supplementary Fig.16(b)–(d).

In Supplementary Fig.16(c), we normalize the SPI-measured target channel intensities and apply thresholding to determine whether the corresponding channel lies within the field of view, which guides the validation of the directly computed trajectory. Supplementary Fig.16(c) displays the raw trajectory together with the corrected and completed valid trajectory, and reports the two-dimensional motion information of the target, which we use to perform motion-compensated reconstruction.

Supplementary Fig.16(d) presents the reconstruction without motion compensation and without extend-pixel processing, as well as the reconstruction obtained with IMCFT and extend-pixel processing. These results show that we successfully reconstruct the complete image of a target larger than the field of view and that the conventional TVAL3 algorithm effectively refines the reconstruction.

We appreciate the reviewers' helpful guidance and constructive feedback.

Q6.3: Finally, the term "noise speckle" (lines 215 and 218), used to describe motion artifacts, seems imprecise and potentially confusing. It would be beneficial to re-evaluate this terminology for greater accuracy.

Reply:

We sincerely appreciate the reviewers' careful review and constructive suggestions. We have revised the manuscript accordingly and believe the changes improve readability and rigor.

We have replaced "noise speckle" with the more descriptive "low-frequency blotchy texture" and explicitly indicated that it occurs in the image background to improve terminological accuracy and avoid confusion.

Unlike traditional inverse transformation methods, the motion-compensated basis patterns are no longer strictly orthogonal, which introduces *low-frequency blotchy texture* in *the background* of the IMCT reconstruction image.

We appreciate this constructive comment.

7.# References: Does this manuscript reference previous literature appropriately? If not, what references should be included or excluded?

Reply:

We respectfully note that the relevant literature has already been appropriately cited in the manuscript. We thank the reviewers again for the helpful guidance.

References

- [0] Shijian Li, Xu-Ri Yao, Wei Zhang, Yeliang Wang, and Qing Zhao. Tracking and fast imaging of a moving object via fourier modulation. *Physical Review Applied*, 22(4):044007, 2024.
- [1] Guilherme Beneti Martins, Laurent Mahieu-Williams, Thomas Baudier, and Nicolas Ducros. Openspyrit: an ecosystem for open single-pixel hyperspectral imaging. *Optics Express*, 31(10):15599–15614, 2023.
- [2] Yibo Xu, Liyang Lu, Vishwanath Saragadam, and Kevin F Kelly. A compressive hyperspectral video imaging system using a single-pixel detector. *Nature Communications*, 15(1):1456, 2024.
- [3] Vincent Studer, Jérôme Bobin, Makhlad Chahid, Hamed Shams Mousavi, Emmanuel Candes, and Maxime Dahan. Compressive fluorescence microscopy for biological and hyperspectral imaging. *Proceedings of the National Academy of Sciences*, 109(26):E1679–E1687, 2012.
- [4] Yiwei Zhang, Matthew P Edgar, Baoqing Sun, Neal Radwell, Graham M Gibson, and Miles J Padgett. 3d single-pixel video. *Journal of Optics*, 18(3):035203, 2016.
- [5] Thomas Maitre, Elie Bretin, Romain Phan, Nicolas Ducros, and Michaël Sdika. Dynamic single-pixel imaging on an extended field of view without warping the patterns. In *International Conference on Medical Image Computing and Computer-Assisted Intervention*, pages 275–284. Springer, 2024.

- [6] Jingjing Wu, Lifa Hu, and Jicheng Wang. Fast tracking and imaging of a moving object with single-pixel imaging. *Optics Express*, 29(26):42589–42598, 2021.
- [7] Aswin C Sankaranarayanan, Lina Xu, Christoph Studer, Yun Li, Kevin F Kelly, and Richard G Baraniuk. Video compressive sensing for spatial multiplexing cameras using motion-flow models. *SIAM Journal on Imaging Sciences*, 8(3):1489–1518, 2015.
- [8] Sagi Monin, Evgeny Hahamovich, and Amir Rosenthal. Single-pixel imaging of dynamic objects using multi-frame motion estimation. *Scientific Reports*, 11(1):7712, 2021.
- [9] Thomas Maitre, Elie Bretin, Laurent Mahieu-Williame, Michaël Sdika, and Nicolas Ducros. Hybrid single-pixel camera for dynamic hyperspectral imaging. In *2024 IEEE International Symposium on Biomedical Imaging (ISBI)*, pages 1–5. IEEE, 2024.
- [10] Jae Young Park and Michael B Wakin. Multiscale algorithm for reconstructing videos from streaming compressive measurements. *Journal of Electronic Imaging*, 22(2):021001–021001, 2013.
- [11] Thomas Maitre, Elie Bretin, Laurent Mahieu-Williame, Romain Phan, Michaël Sdika, and Nicolas Ducros. Dual-arm motion-compensated single-pixel imaging. *hal.science*, 2025.

8.# Clarity and context: Is the abstract clear, accessible? Are abstract, introduction and conclusions appropriate?

Q8.1: The abstract is clear and accessible. As outlined in the previous section, the introduction lacks references to previous art.

Reply:

We sincerely thank the reviewer for the positive assessment. In line with the suggestion, we have updated the Introduction by adding citations to existing dynamic SPI techniques. We appreciate the reviewers' helpful guidance.

Dear Editor,

We sincerely appreciate your continued efforts in coordinating the second round of peer review of our manuscript (Manuscript ID COMMS-ENG-25-0420A; Research Article). We are grateful to the reviewers for their thoughtful follow up comments and constructive suggestions. In response, we have carefully revised the manuscript again and provided comprehensive, detailed responses to the reviewers' concerns.

Overall Response

We appreciate the reviewers' positive assessment of our previous response and the valuable constructive suggestions. We note that two main concerns remain. In the following, we provide a concise overview; specific details are addressed in the subsequent point by point responses.

Regarding the concern about manuscript length and readability. We clarify that the present work comprises an optimized Fourier localization method within SPI, a multichannel perspective for motion perception, and an intuitive IMCT reconstruction that enables fast dynamic SPI reconstruction. Together, these elements constitute the proposed MC3-SPI framework. We validated the compatibility of Fourier basis patterns with the framework. And we further validate the potential of MC3-SPI for extended FOV imaging. Given the breadth of the study, a somewhat longer manuscript is needed to present and reproduce the research in a comprehensive way. We believe that providing essential experimental details is our responsibility and will facilitate improved methods in related work. At the same time, we have made every effort to streamline the prose to enhance readability. And the main focus on condensing the Results section, and all revisions are highlighted in blue for ease of checking. On this basis, we respectfully maintain that a simple and direct validation experiment is necessary for the logical flow of the paper.

We agree that our earlier statement on the sub-pixel issue was not sufficiently rigorous, and in light of prior work following the same technical route, we recognize that this aspect has been relatively overlooked to our knowledge. We have quantitatively reassessed the impact of sub-pixel motion on the final image quality of Fourier-SPI, and the results are consistent with the reviewers' view; a sub-pixel shift does introduce model mismatch. Nevertheless, its impact on the final reconstruction is limited, because it predominantly affects high-frequency content, whereas low-frequency information carries greater weight for the final reconstruction quality in Fourier-SPI. In summary, under typical sampling rates, the sub-pixel error can be considered negligible in practice (at approximately 30% sampling, the SSIM decreases from 1.00 to 0.90).

We will respond to the questions point by point and have revised the manuscript accordingly; all changes are clearly marked in green throughout the text. We hope these clarifications address the reviewers' concerns and we appreciate the guidance.

Q2.1:

The authors specify in the Discussion that the present work builds upon their previous publication [0], with modified spatial frequencies of the Fourier patterns and the use of the Sierra-Lite method.

In my opinion, the narrative could be made clearer by highlighting the difference from prior work earlier in the paper: (1) in the Introduction (around line 114) to make the novelty more explicit, and (2) in the Results Section 2.1.

The narrative could also be improved by explaining why [0] needed to be improved, as the authors stated in their reply (but not clearly in the revised document, I believe): "For real objects, the inter-channel intensity distributions are more complex and often yield more clustered centroids, which demands higher localization accuracy to ensure reliable angle estimation."

Reply:

We appreciate the reviewers' constructive suggestions. We have revised the manuscript accordingly as detailed below.

On line 111 of the revised main manuscript:

We develop an optimized Fourier localization method that uses *spatial-frequency-optimized localization patterns* and *Sierra-Lite dithering with serpentine error diffusion* to precisely determine centroids of targets *with diverse sizes and morphologies*.

In Section 2.1 of the Results, we have added the following discussion.

Consequently, the optimized high precision localization method can correctly detect subtle variations across a broader range of target sizes, for example accurately distinguishing the centroid positions of different constituent components in complex targets.

We apologize for the earlier lack of clarity and have refined the wording to make clearer the advantages that improved localization accuracy brings to motion sensing of real-world complex targets.

On line 267 of the revised main manuscript, we further clarify the benefits of high-precision localization.

Compared with the letters “BIT”, the Mario and rocket images are more structurally complex and yield mult centroid localization results that are more densely clustered with smaller spacing, which places stricter demands on localization accuracy and imaging efficiency. However, our *optimized localization method* still *can reliably distinguish the centroids across different channels for complex target*, enabling accurate estimation of the target’s motion state and, *in turn, guiding* motion-compensated reconstruction *with appropriately chosen sampling patterns* to generate a clear, full-color image.

We sincerely appreciate this helpful suggestion, which has informed our revisions.

Q3.4:

The requested analysis (formerly Supplementary Fig.1) has been moved into the main text, and the localization method is now well described. However, the statement that sub-pixel motion can be ignored (“the smallest sampling unit...”) remains debatable, as sub-pixel displacements may still induce model mismatch. This should be indicated.

Reply:

We thank the reviewers for pointing this out. The suggestion is valuable and has helped us improve the clarity of the statement.

As noted, Our statement that sub-pixel motion can be neglected is not appropriate. We have revised this statement and added a brief discussion in the revised text to clarify the point.

It is noteworthy that the minimum sampling unit in SPI is a single pixel. Accordingly, a sub-pixel displacement of the target induces an energy redistribution across neighboring samples at the resolution of the SPI sampling grid, which should be regarded as a nonrigid apparent displacement rather than a strict translation. To robustly capture such minute displacements, increasing the SPI sampling-grid resolution is the most reliable strategy.

In the Discussion, when addressing localization accuracy, we explicitly note that sub-pixel displacements can induce model mismatch, and we provide a quantitative analysis in the Supplementary Note 11.

In this work, we employ Fourier basis functions with spatial frequencies of $1/2M$ and $1/2N$, and apply the Serria-Lite method with a serpentine diffusion path, achieving an optimal localization accuracy of approximately $1/3$ pixels when the target size accounts for approximately $1/9 \sim 1/4$ of the FOV. *It is worth noting that sub-pixel errors can still induce reconstruction model mismatch. We provide a quantitative analysis in Supplementary Fig.18, which confirms that the impact of sub-pixel errors on Fourier-SPI reconstruction is limited and can generally be neglected.*

We are grateful for this helpful suggestion and hope the revisions address the concern raised.

Q3.7:

Although the orthogonality tests have been expanded, they may still be problematic.

I am not fully convinced by the authors' assertion that a pixel constitutes the minimal unit of spatial resolution in motion-compensated imaging. A sub-pixel shift (e.g. 0.3 pixels) alters the measurement based on the dot product between the shifted scene and the pattern. Treating it as equivalent to the unshifted case introduces a model mismatch. Furthermore, if nearest-neighbour interpolation is used to warp binary patterns, interpolation artefacts ("shuffling") may arise, particularly for rotational motion. In [5], the authors emphasised the importance of avoiding warping Hadamard patterns and proposed a bias-free discretisation method, highlighting that the interpolation method used to compensate for pattern motion remains a critical factor.

Assuming cyclic boundary conditions that wrap patterns at their edges also seems debatable: this is acceptable here because no new objects appear during acquisition, but it would otherwise (1) introduce significant bias in the reconstruction and (2) degrade the orthogonality of the basis. Finally, it would be beneficial to assess the orthogonality of the entire pattern set rather than random subsets at different sampling rates to provide a more complete evaluation.

Reply:

We are deeply grateful to the reviewers for the careful reading and thoughtful recommendations that guided our revisions. In light of these comments, we have updated the relevant sections, and we believe that the changes enhance the presentation and strengthen the overall technical soundness.

Sub-pixel

We appreciate the reviewers' insightful comments on sub-pixel issues in motion compensation, which prompted us to reexamine this aspect. We note that, along the same technical route, prior studies did not, to our knowledge, examine sub-pixel errors in depth. We fully agree that a sub-pixel shift (e.g., 0.3 pixels) alters the measurement defined by the dot product between the shifted scene and the pattern and can therefore cause a model mismatch. Nevertheless, earlier works reported high-quality reconstructions, therefore we reassessed the impact of this mismatch on the final image quality. Our results indicate that the adverse impact of sub-pixel motion is limited overall and is nearly imperceptible for the reconstruction of the main target.

Supplementary Note 11: Discussion and analysis of the sub-pixel motion

A core component of MC3-SPI is motion compensation; however, in practice it cannot be perfect. In addition to the usual errors introduced by noise, systematic errors also arise; for example, our optimized localization method still exhibits approximately one-third pixel localization error, and within each measurement frame there is a small temporal offset between localization and imaging. Therefore, after compensating the dominant target motion, residual small motions often persist and remain imperfectly compensated. This residual motion is typically small in magnitude and sub-pixel at the SPI sampling resolution. We therefore analyze this residual sub-pixel motion within the Fourier-SPI framework, which is best suited to MC3-SPI.

In a continuous model, let the image be $f(\mathbf{r})$ and the small displacement be δ . Define the difference:

$$\Delta(\mathbf{r}) = f(\mathbf{r} - \delta) - f(\mathbf{r}). \tag{1}$$

Denote the Fourier transforms by $F(\omega)$ and $\widehat{\Delta}(\omega)$. Then,

$$\widehat{\Delta}(\omega) = (e^{-j\omega \cdot \delta} - 1) F(\omega) = -2j e^{-j\omega \cdot \delta/2} \sin\left(\frac{\omega \cdot \delta}{2}\right) F(\omega), \tag{2}$$

For the small displacement δ ,

$$\widehat{\Delta}(\omega) \approx -j(\omega \cdot \delta) F(\omega). \quad (3)$$

Therefore, the effect of a small displacement on the Fourier spectrum exhibits two characteristics. First, its magnitude scales linearly with the small parameter δ , which means that the effect is limited. Second, it acts as a first-order high-pass filter, since the associated factor is proportional to ω .

In summary, in dynamic Fourier-SPI the effect of the sub-pixel displacement of a target is expected to be modest and more pronounced at higher spatial frequencies. Because low-frequency components carry greater weight for perceived image quality, the theoretical impact of sub-pixel displacement on final image quality is limited. To further quantify this effect, we performed numerical simulations in which the target underwent continuous sub-pixel motion throughout SPI sampling; the results are shown in Supplementary Fig. 1.

Figure 1: Analysis of the impact of sub-pixel motion on the final imaging quality of Fourier-SPI across different sampling rates. The upper panel presents a visual comparison of the reconstructions; the lower panel reports PSNR and SSIM with respect to a static IFT reference at the corresponding sampling rate, using 20 repeats with standard deviations shown as error bands.

Numerical simulations show that sub-pixel motion has a nearly imperceptible effect on Fourier-SPI reconstructions, which agrees with our previous analysis. And from the standpoint of motion-induced ghosting, sub-pixel shifts contribute only weakly to the final weighted-sum reconstruction, particularly for the main body of the target. Quantitatively, PSNR and SSIM evaluations corroborate this finding and show that the impact increases at higher spatial frequencies, yet remains small in magnitude. We therefore conclude that in MC3-SPI the effect of sub-pixel errors is limited and can generally be neglected.

Finally, it should be noted that Hadamard-SPI shares the same distribution of information weights across spatial frequencies as Fourier-SPI, so the theoretical impact of sub-pixel errors should likewise be small.

Interpolation

We thank the reviewer for the thoughtful observation. We fully agree that the interpolation strategy used for motion compensation is a key factor that affects reconstruction quality. In our original code, we compensate rotation by inversely rotating the pattern. This step inevitably requires interpolation

(e.g., bilinear or bicubic), which degrades pattern morphology and thereby reduces the sharpness of the reconstructed image. Under all tested conditions in our experiments, IMCHT reconstructions with Hadamard patterns were severely blurred. These observations and our interpretation are consistent with [5].

To further analyze the rotation induced degradation, we performed an orthogonality analysis to partially quantify the extent of degradation. The results indicate that the Fourier basis patterns preserve their morphology better after rotation, whereas the Hadamard basis patterns degrade more noticeably, which is in line with [5].

After additionally establishing that the complete set of Fourier basis patterns forms a linear space that is closed under translations and rotations, these results provide further support for the superior performance of Fourier basis patterns in motion compensation.

cyclic translation

We appreciate the reviewer’s comment. We respectfully take a different view. Our MC3-SPI simultaneously performs localization and imaging, therefore when a new target appears in the field of view, the first component that is challenged is localization. Moreover, we believe that a purely translational treatment is unlikely to adequately address the scenario where a new target emerges in the FOV.

We respectfully note that our study assumes the target is smaller than the field of view, so the image within the field of view naturally satisfies periodic boundary conditions. We also limit the scope to a single target, as specified in the title. Under these conditions, the use of cyclic translation in this work does not introduce additional bias.

Moreover, multiplying Fourier basis patterns by a phase factor is mathematically equivalent to a cyclic translation and does not degrade the basis orthogonality. For methodological consistency when comparing with Hadamard basis patterns, we therefore apply the same cyclic translation operation.

orthogonality of the entire pattern set

We thank the reviewers for the constructive suggestion. We agree that using the complete pattern set across sampling ratios is more rigorous. Our initial use of random subsets was intended to control computational cost. On our workstation, the computations required to produce the data for Figure 2 took about a week. To better characterize basis-pattern orthogonality under motion compensation, we performed many additional motion-compensation trials and, therefore, relied on random subsets to keep the runtime manageable. Following the reviewers’ suggestion, we conducted an orthogonality evaluation using the entire pattern set, which yields a more comprehensive assessment.

To further substantiate our conclusions, we evaluated the orthogonality of the complete sets of Fourier basis patterns and Hadamard basis patterns under motion compensation. For each basis pattern, we applied random motion-compensation transforms of various types and computed the mean normalized Frobenius inner product with the remaining basis patterns. This procedure was repeated 100 times. The results in Supplementary Fig.2 indicate that the orthogonality of the Fourier basis patterns is more robust to motion compensation than that of the Hadamard basis patterns. Consequently, when performing motion-compensated imaging of moving targets, Fourier-SPI supports effective reconstruction over a wider range of sampling rates, whereas Hadamar-SPI requires higher sampling rates to maintain ensemble orthogonality and preserve image quality.

Figure 2: *Orthogonality analysis of the complete basis pattern sets after translation or rotation or translation and rotation is described using the mean normalized Frobenius inner product. The closer the data distribution is to zero, the better the orthogonality of the basis patterns after motion compensation. In plot, the left panel shows the statistical results of the data, and the right panel displays the probability density function estimated with a normal kernel function and a bandwidth of 2. Random ± 1 patterns are used as a control to benchmark the orthogonality level.*

We thank the reviewer for the constructive suggestions and continued guidance.

Q3.9:

The simulated experiment is informative, though the results confirm that current performance is comparable to classical methods unless additional filtering is used.

Reply:

We thank the reviewers for the helpful reminder and apologize for the earlier ambiguity.

In the simulations the classical method static IFT refers to Fourier single pixel imaging of a static target with reconstruction by IFT. The proposed IMCFT addresses a moving target and reconstructs the image by a weighted summation of motion compensated basis patterns. When the target is static, IMCFT naturally reduces to the classical IFT. Because motion is involved, some degradation in the reconstruction quality of IMCFT is expected. To avoid misunderstanding, we have added a clarification in the Supplementary note 8.

To more precisely quantify the reconstruction efficiency of IMCFT, particularly in comparison with static Fourier-SPI with IFT reconstruction, we conducted numerical simulations based on the framework described in Supplementary Note 5. We present IMCFT reconstructions at multiple

sampling rates and plot detailed PSNR and SSIM curves. *IMCFT performs weighted summation over motion-compensated basis patterns, and when the target is stationary IMCFT reduces to the IFT. Thus, we show IFT reconstructions of a static target at the same sampling rates together with their PSNR and SSIM curves.*

We respectfully clarify that the proposed filtering step is not required for IMCFT. In the simulations we observe a mild tension between PSNR and SSIM. PSNR indicates only a small gap between IMCFT and static IFT, whereas SSIM rises rapidly at low sampling rates and then saturates near 0.6, with little further improvement at higher sampling. To understand this behavior we examined the three components of SSIM, namely luminance, contrast, and structural similarity. The analysis shows that the discrepancy primarily stems from the luminance term, while the contrast and structural terms perform well. We realized that low intensity background noise in the IMCFT reconstructions, together with normalization, leads to an overestimation of luminance and thus to the observed mismatch.

To address this issue, we apply intensity stretching with thresholding to remove low intensity background noise and to improve the luminance evaluation. On mechanistic grounds, we rename the filtering algorithm as linear intensity stretch (LIS). Finally, we note that the visual improvement brought by LIS is barely perceptible to the human eye, so we do not regard it as a necessary component of IMCFT.

Figure 3: *Performance analysis of IMCFT is conducted via numerical simulations across different sampling rates. The IFT reconstruction of a static target (static IFT) serves as the reference, and a Linear intensity stretch (LIS) algorithm is applied for image enhancement. (a) Visual comparison of the reconstructions. (b) PSNR and SSIM results. (c) Results of the three SSIM components, namely luminance, contrast, and structure comparison.*

By contrast, in Fig. 3 (b) the visual inspection agrees reasonably well with PSNR, and the PSNR curves of IMCFT and static IFT differ only slightly across sampling rates, proving that IMCFT reconstructed a low-noise image. However, the visual impression diverges markedly from SSIM. After a rapid rise, the SSIM of IMCFT stabilizes at about 0.6 and shows no further increase, leaving a pronounced gap relative to static IFT.

Considering the definition of SSIM, suppose x and y are two nonnegative image signals, the SSIM is composed of three terms: luminance $l(x, y)$, contrast $c(x, y)$, and structure comparison $s(x, y)$, which is defined as follow:

$$\text{SSIM}(x, y) = [l(x, y)]^\alpha [c(x, y)]^\beta [s(x, y)]^\gamma, \quad (4)$$

with

$$l(x, y) = \frac{2\mu_x\mu_y + C_1}{\mu_x^2 + \mu_y^2 + C_1}, \quad c(x, y) = \frac{2\sigma_x\sigma_y + C_2}{\sigma_x^2 + \sigma_y^2 + C_2}, \quad s(x, y) = \frac{\sigma_{xy} + C_3}{\sigma_x\sigma_y + C_3}. \quad (5)$$

A common choice sets $\alpha = \beta = \gamma = 1$ and $C_3 = C_2/2$. Under this setting SSIM can simplify to Eq.21.

To investigate the causes of the significant decline in SSIM, we performed numerical simulations for each of its three components individually. The results are shown in Supplementary Fig. 3 (c).

We find that the primary cause of the suboptimal SSIM of IMCFT lies in the luminance term; the luminance of IMCFT reconstructions differs substantially from that of static IFT, whereas the contrast and structure comparison terms perform well, with the former stabilizing around 0.9 and the latter approaching 1. It should be noted that the IMCFT reconstructions exhibit a persistent low-intensity low-frequency blotchy background that disperses target energy; furthermore, we normalized the reconstructions to match the `imshow` display range setting `[]` in MATLAB (`imshow(img, [])`), which inflates the overall brightness of IMCFT images and thereby contributes to the aforementioned SSIM discrepancy. Nevertheless, the strong performance in contrast and structure comparison indicate that IMCFT has potential for rapid target recognition at low sampling rates.

We apply a simple linear intensity stretch which smooth high-intensity regions and suppress low-intensity low-frequency blotchy in the background, thereby mitigating the luminance inflation introduced by normalization and enhancing the image. Visual inspection of IMCFT reconstructions before and after processing indicates that the linear intensity stretch does not alter the structural content of the target image, yet it significantly improves PSNR and SSIM, even outperforming static IFT. Therefore, we believe that IMCFT is effective in capturing the target structure and thereby enabling target recognition. Moreover, linear contrast stretch is computationally lightweight and requires only ~ 0.0002 seconds per application. The linear intensity stretch algorithm is detailed in Algorithm. It should be noted that for IMCFT the linear intensity stretch algorithm is not necessary. Although applying this algorithm leads to a noticeable increase in SSIM, the improvement in visual appearance is minimal. In scenarios such as target recognition that focus on structural features, the algorithm offers no practical benefit. Linear intensity stretch becomes a reasonable choice when the aim is to enhance reconstruction quality while retaining a lightweight and simple procedure.

We appreciate the reviewer’s careful reading and helpful remarks.

Q5.3:

Explicit highlighting of the SPC FOV in the reconstructions from Fig.6 could significantly improve understanding.

Reply:

We thank the reviewers for the helpful suggestion.

In Fig.6 we highlight the boundary of the field of view in the two-dimensional trajectory plot and use shading to indicate regions outside the field of view. In the final reconstructed motion video, we mark the field of view boundary with gray dashed lines so that readers can more clearly understand the target motion.

Figure 4: Principles and results of a boundary-motion target imaging. (a) Trajectory processing: By using the signal strength and spatial filtering, we remove the invalid portions of the trajectory caused by the target moving out of the field of view. Further, by using the relative positions of the three centroids, we accurately determine the positions of the centroids outside the field of view, allowing us to obtain the target's real motion state information. (b) Effective motion trajectory and imaging results: This part shows the reconstructed valid motion trajectory and the corresponding imaging results. Fourier patterns are used for imaging, and the motion compensation and reconstruction methods are the same as those described in the previous section. Reconstruction of a 128×128 pixel image with repeatedly sampling 10% of the low-frequency components. (c) Target motion state with condition met: Three important nodes of the target's motion state are selected and correlated with the target's status at those points.

We are grateful for the reviewer's comments and for the opportunity to improve this work.

Dear Editor,

We sincerely appreciate your continued efforts in coordinating the third round of peer review of our manuscript (Manuscript ID COMMS-25-0420A; Research Article). We are also grateful to the reviewers for their careful follow-up comments. In response, we have revised the manuscript accordingly and prepared detailed, point-by-point responses to all concerns raised.

Sincerely,
Xuri Yao
on behalf of all authors
Beijing Institute of Technology

Overall Response

We sincerely appreciate the reviewers' positive assessment of our previous response and their constructive suggestions. We address the questions point by point and have revised the manuscript accordingly, with all modifications highlighted in orange throughout the text. We hope these clarifications address the concerns raised, and we are grateful for the continued guidance.

Q3.7:

Sub-pixel motion

The authors have reflected on the sub-pixel problem and provided an interesting theoretical discussion for dynamic Fourier-SPI. However, regarding the numerical simulations:

1. The sub-pixel motion used in the simulations should be explicitly described.

Reply:

We thank the reviewers for pointing this out, and we sincerely apologize for our oversight in omitting a key description of sub-pixel motion.

In this case, we introduce prior free random sub-pixel motion that persists throughout the SPI sampling process. Following the reviewer's suggestion, we have supplemented the relevant description.

To further quantify this effect, we performed numerical simulations in which the target underwent *continuous random sub-pixel motion throughout SPI sampling*; the results are shown in Supplementary Fig. 18.

We sincerely appreciate the reviewer's thorough guidance.

2. This may be due to phrasing, but the statement "PSNR and SSIM evaluations corroborate this finding and show that the impact increases at higher spatial frequencies, but remains small in magnitude" appears overstated—especially for PSNR, where a 15 dB decrease is significant. That said, the absolute PSNR remains high, which is positive.

Reply:

We appreciate the reviewer's detailed assessment.

We clarify that when quantifying the impact of target sub-pixel motion on the reconstruction, we used the reconstruction of a static target as the reference, which implies that the reference differs across sampling ratios. Therefore, the cross-sampling comparison of the metric decrease is not well defined. By contrast, the absolute value of the metric directly quantifies the loss in reconstruction quality caused by sub-pixel motion at a given sampling ratio, which has a clearer interpretation.

When comparing metric differences, PSNR and SSIM should be computed for the two reconstructions relative to the original image. The outcome should be similar to Fig. 2b (in Q3.9) which evaluates the reconstruction efficiency of IMCFT, where the PSNR difference between static IFT reconstruction and IMCFT reconstruction is about 5 dB. Given that the conditions in Fig. 2b are more stringent, the gap in the present case should be smaller.

We have revised our wording and placed the emphasis on the absolute values of PSNR and SSIM.

Quantitatively, PSNR and SSIM decrease when the sampling ratio increases, yet the absolute values remain high overall, which is consistent with our earlier analysis that the influence of sub-pixel motion is more concentrated in the high-frequency components and is limited overall.

We appreciate the reviewer's careful and constructive guidance.

Interpolation

The authors acknowledge that motion compensation requires interpolation (e.g. bilinear or bicubic), but do not state which method was used. This detail is essential for reproducibility and interpretation.

Reply:

We thank the reviewer for pointing this out.

We acknowledge that we did not make this explicit, since both bilinear and bicubic interpolation are feasible in our method, and we did not observe clear differences between these choices in our experiments. The data and source code that support the findings of this study are available at <https://github.com/ArthurWu1999/MC3-SPI>.

In the Code availability section of the main text, we have also open-sourced the code.

The code for the reconstruction algorithms used in this study and the raw data are publicly available at <https://github.com/ArthurWu1999/MC3-SPI>.

We hope this clarifies.

Cyclic translation

The authors take a different position, which could be acceptable. In that regard, it would help to clearly state that the target is assumed to be smaller than the FOV, and that the background is zero-filled. I'm not sure why the authors discussed the case of a purely translational motion. Neither did I understand why an image within the FOV satisfies periodic boundary conditions: if the target leaves to the right, it does not mean that it appears to the left? I feel like I am missing something.

Reply:

We appreciate the reviewers' careful assessment, and we would be pleased to discuss the details.

We note that, in SPI, when referring to imaging a specific target, it is commonly assumed that the FOV fully covers the target and that the background is clean and close to all-zero. Moreover, for SPI with moving targets, it is commonly assumed that a small target moves within a large FOV. In particular, for motion-sensing and motion-compensated SPI, the localization method intrinsically requires that the target be smaller than the FOV and that the background remain sufficiently clean. On this basis, we have not explicitly restated the target size and the near all-zero background in the main text.

Regarding why we only consider pure translation when discussing cyclic translation, we note that planar motion has two degrees of freedom, namely translation and rotation. The cyclic-translation operation compensates only the translational component of the target motion, so the rotational component can be neglected at this stage.

Regarding the periodic boundary conditions, we clarify that the compensation operates only on the target (or the visible portion of the target) within the FOV. On this base, a simple translation with zero padding is essentially indistinguishable from cyclic translation.

In the orthogonality analysis, we apply cyclic translations to the Hadamard patterns to correspond to the Fourier patterns. As shown in Eq. 8 of the main text, the translational compensation of the pattern is $\text{conj}(F_i) e^{j2\pi((f_x, f_y)_i, (x_i - x_0, y_i - y_0))}$. Owing to the periodic structure of the Fourier bases, this operation is equivalent to a cyclic translation.

We hope this clarifies, and we appreciate the reviewers' advice.

Orthogonality

As suggested, the authors now assess orthogonality over the entire pattern set, which is considerably more interpretable.

Even though the authors agree that evaluating the full pattern set is more rigorous and added those results to the supplementary material, the main paper (and part of the supplementary) still present results based on random subsets. In particular, the improvement of Hadamard pattern orthogonality at higher sampling ratios may arise from interpolation artifacts that flatten high-frequency patterns into near-zero patterns. If so, the conclusion that Hadamard-SPI “requires higher sampling ratios” may be partly influenced by the metric rather than the underlying reconstruction performance.

Reply:

We appreciate the reviewers’ comment.

We agree that using the complete sampling pattern set is the most rigorous approach, however we clarify that our analysis focuses on the statistical distribution of the normalized Frobenius inner product to assess the level of orthogonality of the patterns after motion compensation. We regard a random subset as a down-sampling of this distribution, and we report the median and the $p_{25} \sim p_{75}$ interquartile range (IQR) as a robust uncertainty band, which we believe is statistically reasonable.

We next provide a brief computational complexity analysis to clarify that the random subset scheme already entails a very high computational cost.

For each sampling rate of Fourier basis, m, n are the base size, k selected bases, and L operations per basis. The uncompensated subset matrix has an effective size of approximately $(k - 1) \times (mn)$, whereas each compensated basis is a vector of length mn . As a result, a single multiplication requires on the order of $O(kmn)$ complex multiply-accumulate operations. Consequently, with the total number of matrix-vector products kL , the overall time complexity can be expressed as $O(Lk^2mn)$.

Substituting the specific parameters $m = n = 128$ (hence $mn = 16384$), $k = 800$, $L = 100$, the aggregate computational load of about 1.04×10^{12} complex multiply-accumulate operations, which is on the order of 10^{12} . Moreover, this estimate does not yet account for the additional cost of the motion compensation, which are also invoked 80000 times for each sampling rate.

We appreciate the reviewer’s insight about the compensation patterns. We agree with the observation that high-frequency Hadamard basis patterns may, due to interpolation, be flattened into near-zero patterns. We noted a similar issue during the first round, namely that the raw Frobenius inner product can be influenced by the magnitude distribution of the matrices and therefore does not reliably assess orthogonality. Consequently, we use the normalized Frobenius inner product, that is, the Frobenius inner product divided by the product of the Frobenius norms of the two matrices, to more accurately evaluate their orthogonality.

We apologize for the lack of clarity in our description of the orthogonality metric. To avoid ambiguity, we have named it *sMNFI* (scaled mean normalized Frobenius inner product magnitude) and have replaced the term accordingly in the main text and the Supplementary Notes 7.

Each chosen pattern is randomly rotated, translated, or both, and its *sMNFI* (*scaled mean normalized Frobenius inner-product magnitude*) with the other *motion-uncompensated patterns* in the subset is computed (Eq. 12 in Supplementary Note 7).

We hope this clarifies, and we appreciate the reviewers’ advice.

In my opinion, it would strengthen the paper to: Move Supplementary Fig. 11 into the main document, Remove Fig. 2 (or optionally move it to the supplementary material, while being cautious about the results interpretation).

Reply:

We appreciate the reviewer’s helpful suggestion. After careful consideration, we would like to retain our current presentation, as we believe it better reflects typical Fourier-SPI practice and maintains an appropriate balance between completeness and readability.

First, in Fourier-SPI, full sampling is rarely adopted in practice because the required acquisition time can be prohibitively long. Moreover, high-frequency patterns tend to be more sensitive to noise, such that increasing the sampling rate does not necessarily translate into improved reconstructions and may, in some settings, even lead to reduced quality. Since the low-frequency components of the Fourier basis often carry the dominant image information, analyses at lower sampling ratios can be more practically informative. Second, we aim for figures in the main text to convey the key results with sufficient information density, while reserving more detailed distributions and supporting analyses for the Supplementary Materials, which we hope provides a clear and reader-friendly organization.

We thank the reviewer for the helpful suggestion. In the caption of Fig. 2, we have referenced Supplementary Fig. 10 and Fig. 11 so that readers interested in the full data distribution can quickly locate the corresponding material in the Supplementary Notes.

Figure 1: Orthogonality analysis of Fourier and Hadamard basis patterns after motion-compensation at sampling rates between 5% and 50%. The orthogonality of the basis patterns is quantified by the *sMNF* (*scaled mean normalized Frobenius inner-product magnitude*) between each compensated basis pattern and the other *uncompensated basis patterns*. Statistical analysis employs the median together with the $p_{25} \sim p_{75}$ interval (IQR). Values concentrated nearer to zero indicate stronger orthogonality. The random ± 1 patterns (without motion-compensation) serve as a reference. Orthogonality analysis under (a) random translation compensation, (b) random rotation compensation, and (c) random translation and rotation compensation. *The specific data distributions are shown in Supplementary Fig. 10 and Supplementary Fig. 11.*

We are sincerely appreciative of the reviewer’s thoughtful guidance.

Important Note: It should be made explicit in the main manuscript that the orthogonality is evaluated between motion-compensated patterns and the static (original) patterns without the reader having to go through the supplemental materials.

Reply:

We thank the reviewers for the careful assessment and the helpful suggestions.

Following your recommendation, we have added the key description.

For each sampling rate, we randomly select 800 basis patterns to form a subset. Each chosen pattern is randomly rotated, translated, or both, and its mean normalized Frobenius inner product with *the other motion-uncompensated patterns* in the subset (Eq. 12 in Supplementary Note 7) is computed.

Figure 1: Orthogonality analysis of Fourier and Hadamard basis patterns after motion-compensation at sampling rates between 5% and 50%. The orthogonality of the basis patterns is quantified by the *sMNF* (scaled mean normalized Frobenius inner-product magnitude) between each compensated basis pattern and the other *uncompensated basis patterns*. Statistical analysis employs the median together with the $p_{25} \sim p_{75}$ interval (IQR). Values concentrated nearer to zero indicate stronger orthogonality. The random ± 1 patterns (without motion-compensation) serve as a reference. Orthogonality analysis under (a) random translation compensation, (b) random rotation compensation, and (c) random translation and rotation compensation. *The specific data distributions are shown in Supplementary Fig. 10 and Supplementary Fig. 11.*

We are sincerely grateful for the reviewers' thoughtful guidance.

I was surprised by the reported computation time ("about a week") and would be curious to know more about the hardware/software used, although this is mostly personal curiosity.

Reply:

We thank the reviewer for the inquiry.

The workstation used in this study was configured as follows: CPU R9-9950X with PBO enabled; RAM 96 GB at 6000 MT/s, CL 28; a ZHITAI TiPro9000 SSD on PCIe 5.0 with 4 TB capacity. All numerical simulation experiments throughout the manuscript were implemented in MATLAB.

The computational complexity of the orthogonality analysis for Fourier basis patterns at a single sampling rate is as shown earlier. We set 46 nodes between sampling ratios of 5% and 50%, covering three types of basis patterns and three motion compensation settings, therefore a complete evaluation would increase the computational complexity by at least two orders of magnitude.

We hope this clarifies our statement regarding the computation time of approximately one week, and we appreciate your understanding.

Q3.9:

The labeling issue has been clarified in the response, but Figure 3b still shows “IFT” in the legend, which should be corrected.

Reply:

We appreciate the reviewer’s careful assessment, and we have corrected our oversight.

Figure 2: Performance analysis of IMCFT is conducted via numerical simulations across different sampling rates. The IFT reconstruction of a static target (static IFT) serves as the reference, and a Linear intensity stretch (LIS) algorithm is applied for image enhancement. (a) Visual comparison of the reconstructions. (b) PSNR and SSIM results. (c) Results of the three SSIM components, namely luminance, contrast, and structure comparison.

We sincerely appreciate the reviewer’s meticulous guidance.

Q5.3:

The SPC FOV was not highlighted in the reconstructions, but this is acceptable as it was only a suggestion.

Reply:

We thank the reviewer for the helpful suggestion, and we apologize that the highlighting in the trajectory plot did not meet the expectation.

In panel C, we have labeled the boundary of the FOV and blurred the region outside the FOV to provide a clear distinction.

Figure 3: Principles and results of a boundary-motion target imaging. (a) Trajectory processing: By using the signal strength and spatial filtering, we remove the invalid portions of the trajectory caused by the target moving out of the field of view. Further, by using the relative positions of the three centroids, we accurately determine the positions of the centroids outside the field of view, allowing us to obtain the target's real motion state information. (b) Effective motion trajectory and imaging results: This part shows the reconstructed valid motion trajectory and the corresponding imaging results. Fourier patterns are used for imaging, and the motion compensation and reconstruction methods are the same as those described in the previous section. Reconstruction of a 128×128 pixel image with repeatedly sampling 10% of the low-frequency components. (c) Target motion state with condition met: Three important nodes of the target's motion state are selected and correlated with the target's status at those points.

Dear Editor,

We sincerely appreciate your continued efforts in coordinating the fourth round of peer review of our manuscript (Manuscript ID COMMS-25-0420A; Research Article). We are also grateful to the reviewers for their careful follow-up comments. In response, we have revised the manuscript accordingly and prepared detailed, point-by-point responses to the remaining concerns raised.

We hope that our revisions have addressed all concerns and we look forward to receiving your final decision. Thank you for your time and consideration.

Sincerely,
Xuri Yao
on behalf of all authors
Beijing Institute of Technology

Q1: Concerning the orthogonality analysis

Please justify why orthogonality is evaluated between motion-compensated patterns and the original static patterns, rather than between the motion-compensated patterns themselves.

Reply:

We thank the reviewer for this insightful question. We respectfully clarify that the primary purpose of our orthogonality analysis is to quantify the loss of orthogonality induced by the motion compensation process itself, or more precisely, to evaluate the deviation of the compensated matrix relative to the original ideal matrix. This choice is justified by two key reasons:

First, matrices such as the Fourier and Hadamard matrices possess well-understood and optimal orthogonal properties, which have been extensively validated in variety SPI applications. Our objective is not to generate a new set of orthogonal bases with unknown properties through motion compensation, but rather to verify if the compensated patterns can still approximate the ideal orthogonal characteristics.

Second, evaluating orthogonality between compensated patterns mixes the effects of different motion operators and relative motion, which obscures the specific distortion introduced by the compensation method. Compared with the original static patterns, we can directly isolate the error introduced by the compensation process.

Therefore, we believe that our approach provides a clearer measure of the fidelity of the motion compensation method.

Q2: Concerning cyclic translation

The authors state in their response that it is commonly assumed that the field of view fully covers the target and that the background is clean. However, their own "extend-FOV" method explicitly relaxes these assumptions by relying on partial capture of the target in at least two channels. For a broad-scope journal, the assumptions underlying each method should be stated precisely and consistently.

Reply:

We take note of the reviewer's comment.

We provide a further clarification regarding the previous review round. When the target is smaller than the FOV, motion compensation operates on the entire target that appears fully within the FOV. In the extend-FOV experiment, although the target never appears fully within the FOV, each compensation step is applied to the portion of the target that lies inside the FOV at the corresponding measurement. Therefore these two experimental settings share the same theoretical basis.

We respectfully clarify that all experiments in this work implement motion compensation using circular translation. The source code supporting the findings of the extend-FOV method have already been made publicly available at <https://github.com/ArthurWu1999/MC3-SPI/tree/main/half2full>.

Q3: Concerning the description of sub-pixel motion

The description of sub-pixel motion as a "continuous random sub-pixel motion throughout the SPI sampling" remains vague. There are multiple ways to generate random sub-pixel motion, and the specific method used should be stated more explicitly.

Reply:

We appreciate the reviewer's comment.

In our simulations, we used the `rand() - 0.5` expression in MATLAB to generate random sub-pixel motion, which yields zero-mean random shifts that are uniformly distributed within $[-0.5, 0.5]$ pixels. The horizontal and vertical sub-pixel shifts were generated independently and applied throughout the SPI sampling sequence to emulate continuous random sub-pixel motion during acquisition.

To further quantify this effect, we performed numerical simulations in which the target underwent *continuous random sub-pixel motion throughout SPI sampling, where the sub-pixel shifts were uniformly distributed in $[-0.5, 0.5]$ pixels*; the results are shown in Supplementary Fig. 18.

Q4: Concerning the interpolation order

The manuscript still does not explicitly state which interpolation method is used, as the author claim an interested reader would refer to the provided code. It is acceptable, even though this information is important for interpreting the results and could easily be included in the manuscript.

Reply:

We thank the reviewer for providing the comment.

Although the code is publicly available, we acknowledge that explicitly stating the parameters improves clarity. Therefore, we have added a description of the interpolation method to the Supplementary Material to facilitate easier interpretation of the results, as recommended.

In practical applications, when rotating the basis patterns, interpolation methods (*bilinear and bicubic, as employed in this paper*) must be employed, as illustrated in the Supplementary Fig.12.

Key results: Please summarise what you consider to be the outstanding features of the work.

This work addresses the challenge of imaging moving scenes using single-pixel technology. The authors propose a novel motion estimation pipeline that can recover 2D rigid motions during acquisition using a three-detector single-pixel camera. Each detector corresponds to a distinct spectral channel. By projecting two additional Fourier light patterns per measurement, the method can estimate both the centre of mass and the rotation angle of the scene, provided there are distinct intensity distributions across spectral bands. Notably, the multichannel design enables the system to infer motion beyond the field of view when one channel leaves the observable region.

Validity: Does the manuscript have flaws which should prohibit its publication? If so, please provide details.

The manuscript presents findings that are of interest; however, two main issues should prohibit its publication.

1. The paper is not positioned within the dynamic single-pixel imaging literature. As will be explained in the next section, the paper overlooks key approaches (e.g. similar camera designs and alternative motion estimation strategies).
2. The manuscript claims to handle arbitrary motion, including in the title ("...arbitrarily moving target"), despite relying on a planar rigid motion assumption.

Originality and significance: If the conclusions are not original, please provide relevant references. On a more subjective note, do you feel that the results presented are of immediate interest to many people in your own discipline, and/or to people from several disciplines?

The manuscript requires a more comprehensive literature review.

In [0], the authors pioneered the use of Fourier localization patterns to estimate the center of mass of a scene. It is imperative that the manuscript explicitly states whether the work builds on or extends [0], thereby clarifying the novelty of the present contribution.

In general terms, the proposed approach is not situated within the domain of single-pixel imaging techniques. It is imperative that the manuscript explicitly relates the proposed approach to prior work in hyperspectral single-pixel imaging, stereo single-pixel techniques, and other multichannel designs (see references 1–4). Furthermore, the discourse on motion estimation is narrowly centred on methodologies employing localization patterns, while disregarding numerous well-established strategies in dynamic single-pixel imaging. It is notable that the following approaches are not discussed: preview-based methods that estimate motion from low-resolution reconstructions [6-8], hybrid systems incorporating conventional CMOS cameras [9], and relevant video compressed sensing techniques [2, 7, 10]. Furthermore, related work that addresses the reconstruction of the scene beyond the FOV, such as [5], could be mentioned to clarify the distinction between motion estimation and out-of-FOV reconstruction.

From my perspective, the methodology under discussion is relevant and of interest to researchers working in single-pixel imaging, computational imaging, and motion estimation. The discussion thus initiated is of particular pertinence with regard to the utilisation of multichannel information for motion tracking. Nevertheless, it is my conviction that the results and their presentation still require further

development and clarification before the full potential and significance of the contribution can be appreciated by a broader audience.

#Data & methodology: Please comment on the validity of the approach, quality of the data and quality of presentation. Please note that we expect our reviewers to review all data, including any extended data and supplementary information. Is the reporting of data and methodology sufficiently detailed and transparent to enable reproducing the results?

Validity of the Approach:

The proposed approach is conceptually sound and addresses the challenge of motion estimation in single-pixel imaging using a novel three-detector system. The utilisation of colour-channel separation for motion inference represents a novel and well-justified approach.

The assumption of rigid motion is a reasonable starting point and is common in the field (especially for short acquisitions), but the authors should more explicitly discuss its limitations in the context of their system. For instance, in symmetric object configurations (e.g., two identical features rotating around a centre), the centre of mass may remain unchanged, resulting in erroneous angle estimations. Moreover, the method appears to be applicable only in circumstances where no additional objects enter the field of view. This is due to the fact that the extended FoV capability is reliant upon intensity filtering in relation to a fixed reference frame. Furthermore, the efficacy of the method in real-world scenarios with non-zero backgrounds remains to be elucidated.

Quality of the Data and Presentation:

I suggest explaining the target's movement in "the real experiments. Also, the sentence on line 189 ("To further validate... we conducted... with real targets") is confusing. It makes it seem like the earlier experiment was synthetic, but this doesn't seem to be true.

Figure 2 is similar to Figure 3, so it can be left out of the main paper to make space for more valuable content from the extra material (e.g., some details on the improved Fourier localization method).

The supplementary data useful, but it also causes problems:

- Supp. Fig. 1: It is important to understand the proposed localization method, so Supp. Fig. 1 should be moved into the main text. We don't know if the localization was checked using just one channel or the full three-channel device. This needs to be explained, especially the performance gain from using multiple channels. The authors currently check the position of the centre of mass, but because the goal is to estimate how much something has moved, it may be better to check the errors in the movements between frames. This would show how well the system is working..

- Supp. Fig. 4: Please explain how the motion was simulated.

- Supp. Fig. 5: The figure looks at different levels of noise, but it doesn't explain what kind of noise model is being used (e.g. is it Gaussian, Poisson or Skellam?). The figure also assumes that 40dB SNR is the lowest possible range, but this seems quite high (please explain how the signal powers given in the extra material are defined). What we can see is that the angle estimation is more accurate than the centre-of-mass estimation. This is because the displacement estimate is based on relative positioning, not absolute positioning.

- Supp. Fig. 7 is lacking in thorough analysis: a single translation or rotation is not sufficient to establish a statistical analysis, in particular because the results should differ significantly for integer or sub-pixel pattern shifts. It is imperative that the probability distributions presented incorporate kernel definitions. Furthermore, the colormaps depicted in panels c) and d) are challenging to discern and are not consistent with those observed in panels a) and b). The authors also provide a mathematical explanation of the advantage of the Fourier basis, and this discussion could benefit from referencing recent work [5] that addresses similar ideas (interpolation of sharp Hadamard patterns leads to issues).

- Supp. Fig. 9: The caption doesn't explain the differences between the first two rows clearly.

- Supp. Fig. 10 presents quantitative results under low sampling rates. Since no ground truth is available from experimental data, the authors use TVAL3 with 50% measurements as a reference. However, a simulated experiment with known ground truth could deliver a more compelling and interpretable evaluation.

Reproducibility:

Aside from the issues noted earlier, the methodology is described in sufficient detail to follow the general approach.

The authors kindly state that reconstruction code is available upon reasonable request. However, it is not entirely clear whether the code or implementation for motion estimation is included. If we were to consider providing access to this part of the pipeline, it could potentially lead to a significant improvement in reproducibility.

Additionally, it may be beneficial to consider providing access to the experimental data, or alternatively using publicly available datasets (such as those from [9, 11]), as this could significantly strengthen the reproducibility and accessibility of the work.

Appropriate use of statistics and treatment of uncertainties: All error bars should be defined in the corresponding figure legends; please comment if that's not the case. Please include in your report a specific comment on the appropriateness of any statistical tests, and the accuracy of the description of any error bars and probability values.

See the comment made in the previous section on supp. Fig. 7.

Conclusions: Do you find that the conclusions and data interpretation are robust, valid and reliable?

The conclusions in the main manuscript feel relatively limited in scope. While the paper effectively demonstrates that the proposed method yields visually compelling results and that Fourier patterns appear more suitable than Hadamard patterns for motion-compensated imaging, several crucial insights are relegated to the supplementary material. Specifically, the analysis of localization error with Fourier patterns and the orthogonality of motion-compensated patterns are vital. Integrating and discussing these points within the main document would significantly strengthen the overall narrative and impact of the research.

The manuscript repeatedly suggests that the method operates without assumptions on scene motion. This claim is misleading. The approach explicitly relies on a rigid planar motion model, which is a significant underlying assumption. This critical dependency should be clearly acknowledged and

thoroughly discussed throughout the manuscript to provide an accurate representation of the method's applicability and limitations.

The authors state they were able to "perform motion-compensated imaging to directly reconstruct a complete image of the target" (line 263). This statement is misleading. While the method successfully estimates motion that extends beyond the conventional field of view, it does not actually reconstruct parts of the scene that lie outside the system's observable area. The authors should rephrase this claim to accurately reflect the true scope of their contribution and avoid overstating the capabilities of their system.

One of the manuscript's stated conclusions is that the method remains robust under noise conditions of 40 dB SNR. However, as highlighted in the discussion of Supplementary Figure 5, this may represent a relatively high SNR threshold, which raises concerns about the method's applicability in lower-SNR, real-world scenarios. This suggests the method might be less robust than initially implied, particularly in more challenging practical conditions. That said, it is important to note that the method performed well on real experimental data, indicating that its practical utility may extend beyond what is suggested by the simulations alone. A more nuanced discussion of this apparent discrepancy between simulation results and real-data performance would significantly help clarify the method's actual robustness and its limitations in varying noise environments.

Suggested improvements: Please list additional experiments or data that could help strengthening the work in a revision.

The main document exhibits some redundancy, and unfortunately, much of the deeper technical insight is relegated to the supplementary material. A reorganization that integrates key analyses, such as the localization error and the orthogonality of basis functions, into the main text would significantly enhance clarity and strengthen the overall message.

To further solidify the contributions, consider including the following experiments:

1. Evaluate the relative displacement between frames (rather than absolute position) in Supplementary Figure 1. This would more accurately reflect the method's direct relevance to motion compensation.
2. Provide an example of reconstruction beyond the field of view. This would powerfully demonstrate the practical benefit of estimating motion even when it occurs outside the sensor's direct range.

Finally, the term "noise speckle" (lines 215 and 218), used to describe motion artifacts, seems imprecise and potentially confusing. It would be beneficial to re-evaluate this terminology for greater accuracy.

References: Does this manuscript reference previous literature appropriately? If not, what references should be included or excluded?

[0] Li, S., Yao, X.-R., Zhang, W., Wang, Y., Zhao, Q.: Tracking and fast imaging of a moving object via fourier modulation. *Physical Review Applied* 22(4), 044007 (2024)

[1] G. Beneti Martins, L. Mahieu-Williams, T. Baudier, and N. Ducros, "Openspyrit: an ecosystem for open single-pixel hyperspectral imaging," *Optics Express*, vol. 31, no. 10, pp. 15 599–15 614, 2023

[2] Y. Xu, L. Lu, V. Saragadam, and K. F. Kelly, "A compressive hyperspectral video imaging system using a single-pixel detector," *Nature Communications*, vol. 15, no. 1, p. 1456, 2024

- [3] V. Studer, J. Bobin, M. Chahid, H. S. Mousavi, E. Candes, and M. Dahan, "Compressive fluorescence microscopy for biological and hyperspectral imaging," *Proceedings of the National Academy of Sciences*, vol. 109, no. 26, pp. E1679–E1687, 2012.
- [4] Y. Zhang, M. P. Edgar, B. Sun, N. Radwell, G. M. Gibson, and M. J. Padgett, "3d single-pixel video," *Journal of Optics*, vol. 18, no. 3, p. 035203, 2016.
- [5] T. Maitre, E. Bretin, R. Phan, N. Ducros, and M. Sdika, "Dynamic single-pixel imaging on an extended field of view without warping the patterns," in *International Conference on Medical Image Computing and Computer-Assisted Intervention*. Springer, 2024, pp. 275–284.
- [6] J. Wu, L. Hu, and J. Wang, "Fast tracking and imaging of a moving object with single-pixel imaging," *Optics Express*, vol. 29, no. 26, pp. 42 589–42 598, 2021.
- [7] A. C. Sankaranarayanan, L. Xu, C. Studer, Y. Li, K. F. Kelly, and R. G. Baraniuk, "Video compressive sensing for spatial multiplexing cameras using motion-flow models," *SIAM Journal on Imaging Sciences*, vol. 8, no. 3, pp. 1489–1518, 2015.
- [8] S. Monin, E. Hahamovich, and A. Rosenthal, "Single-pixel imaging of dynamic objects using multi-frame motion estimation," *Scientific Reports*, vol. 11, no. 1, p. 7712, 2021
- [9] T. Maitre, E. Bretin, L. Mahieu-Williame, M. Sdika, and N. Ducros, "Hybrid single-pixel camera for dynamic hyperspectral imaging," in *2024 IEEE International Symposium on Biomedical Imaging (ISBI)*. IEEE, 2024, pp. 1–5.
- [10] J. Y. Park and M. B. Wakin, "Multiscale algorithm for reconstructing videos from streaming compressive measurements," *Journal of Electronic Imaging*, vol. 22, no. 2, pp. 021 001–021 001, 2013
- [11] Maitre, T., Bretin, E., Mahieu-Williame, L., Phan, R., Sdika, M., & Ducros, N. (2025). Dual-arm motion-compensated single-pixel imaging. <https://hal.science/hal-05068181v1>

Clarity and context: Is the abstract clear, accessible? Are abstract, introduction and conclusions appropriate?

The abstract is clear and accessible. As outlined in the previous section, the introduction lacks references to previous art.

Inflammatory material: Does the manuscript contain any language that is inappropriate or potentially libelous?

No.

Springer Nature is committed to diversity, equity and inclusion; please raise any concerns that may in your view have an impact on this commitment.

None.

Please indicate any particular part of the manuscript, data, or analyses that you feel is outside the scope of your expertise, or that you were unable to assess fully.

Please address any other specific question asked by the editor via email.

Overall Comment

The manuscript has improved substantially and most of the previous concerns have been addressed by providing clearer explanations and more extensive analyses. The authors now reference the relevant literature and discuss potential extensions more effectively. The structure is also better organised, which improves the overall presentation.

However, I have two main concerns. Firstly, the readability of the manuscript could be improved further. There is redundancy in the explanations, and the length of both the main text and the supplementary materials makes the paper difficult to follow. Secondly, the treatment of sub-pixel motion and the orthogonality assumptions are still conceptually debatable (see comment Q3.7).

These should be discussed as limitations rather than being assumed to be negligible. Addressing these two points would significantly strengthen the manuscript.

I list all my comments about the authors' point-by-point response below.

Q1.1

The authors have added a discussion of dynamic SPI literature in the Introduction.

Q1.2

They have revised the title and clarified that the method applies to rigid two-dimensional motion only.

Q2.1

The authors specify in the Discussion that the present work builds upon their previous publication [0], with modified spatial frequencies of the Fourier patterns and the use of the Sierra-Lite method.

In my opinion, the narrative could be made clearer by highlighting the difference from prior work earlier in the paper: (1) in the Introduction (around line 114) to make the novelty more explicit, and (2) in the Results Section 2.1.

The narrative could also be improved by explaining why [0] needed to be improved, as the authors stated in their reply (but not clearly in the revised document, I believe): "For real objects, the inter-channel intensity distributions are more complex and often yield more clustered centroids, which demands higher localization accuracy to ensure reliable angle estimation."

Q2.2

The relationship to hyperspectral, stereo, and multichannel SPI techniques has been added with appropriate citations and an explanation of compatibility.

Q2.3

Relevant prior dynamic-SPI and hybrid-system references have been included, and distinctions with preview-based and CMOS-assisted approaches are discussed.

Q2.4

The closing discussion now emphasizes broader significance and possible extensions.

Q3.1

The authors added a discussion of the limitations of their rigid-motion assumption and proposed additional methods to address issues such as centrosymmetry, multiple objects, and background interference.

Q3.2

The description of target motion in real experiments has been clarified with quantitative details and consistent terminology

Q3.3

The authors argue that retaining Figure 2 improves readability. I remain somewhat concerned that the manuscript may be overly long, which could hinder understanding.

Q3.4

The requested analysis (formerly Supplementary Fig. 1) has been moved into the main text, and the localization method is now well described. However, the statement that sub-pixel motion can be ignored ("the smallest sampling unit...") remains debatable, as sub-pixel displacements may still induce model mismatch. This should be indicated.

Q3.5

The motion simulation process has been fully detailed with explicit equations.

Q3.6

The noise model and SNR definition are now clearly explained (Gaussian white noise using ``awgn()``), and results are clarified.

Q3.7

Although the orthogonality tests have been expanded, they may still be problematic.

I am not fully convinced by the authors' assertion that a pixel constitutes the minimal unit of spatial resolution in motion-compensated imaging. A sub-pixel shift (e.g. 0.3 pixels) alters the measurement based on the dot product between the shifted scene and the pattern. Treating it as equivalent to the unshifted case introduces a model mismatch. Furthermore, if nearest-neighbour interpolation is used to warp binary patterns, interpolation artefacts ('shuffling') may arise, particularly for rotational motion. In [5], the authors emphasised the importance of avoiding warping Hadamard patterns and proposed a bias-free discretisation method, highlighting that the interpolation method used to compensate for pattern motion remains a critical factor.

Assuming cyclic boundary conditions that wrap patterns at their edges also seems debatable: this is acceptable here because no new objects appear during acquisition, but it would otherwise (1) introduce significant bias in the reconstruction and (2) degrade the orthogonality of the basis. Finally, it would be beneficial to assess the orthogonality of the entire pattern set rather than random subsets at different sampling rates to provide a more complete evaluation.

Q3.8

Presentation and data organization have improved; no remaining issues.

Q3.9

The simulated experiment is informative, though the results confirm that current performance is comparable to classical methods unless additional filtering is used.

Q3.10

Code clarification noted; response is acceptable.

Q5.1

The authors discussed centroid localization and pattern orthogonality in the main document, aligning with previous comments.

Q5.2

Clarification on the motion assumption is acceptable.

Q5.3

Explicit highlighting of the SPC FOV in the reconstructions from Fig. 6 could significantly improve understanding.

Q5.4

Noise assumptions have been clarified.

Q6.1

Overall manuscript clarity and structure have improved, but the length and language could still benefit from polishing.

Q6.2.1

Technical corrections and explanations are acceptable.

Q6.2.2

The new experiment for the extended FOV is good.

Q6.3

The term “low-frequency blotchy texture” is satisfactory; no further issues.